# Stochastic Semi-Gradient Descent for Learning Mean Field Games with Population-Aware Function Approximation

**Chenyu Zhang**
MIT
Cambridge, MA 02139
zcysxy@mit.edu

**Xu Chen**
Columbia University
New York, NY 10025
xc2412@columbia.edu

**Xuan Di**
Columbia University
New York, NY 10025
sharon.di@columbia.edu

## Abstract

Mean field games (MFGs) model interactions in large-population multi-agent systems through population distributions. Traditional learning methods for MFGs are based on fixed-point iteration (FPI), where policy updates and induced population distributions are computed separately and sequentially. However, FPI-type methods may suffer from inefficiency and instability due to potential oscillations caused by this forward-backward procedure. In this work, we propose a novel perspective that treats the policy and population as a unified parameter controlling the game dynamics. By applying stochastic parameter approximation to this unified parameter, we develop SemiSGD, a simple stochastic gradient descent (SGD)-type method, where an agent updates its policy and population estimates simultaneously and fully asynchronously. Building on this perspective, we further apply linear function approximation (LFA) to the unified parameter, resulting in the first population-aware LFA (PA-LFA) for learning MFGs on continuous state-action spaces. A comprehensive finite-time convergence analysis is provided for SemiSGD with PA-LFA, including its convergence to the equilibrium for linear MFGs—a class of MFGs with a linear structure concerning the population—under the standard contractivity condition, and to a neighborhood of the equilibrium under a more practical condition. We also characterize the approximation error for non-linear MFGs. We validate our theoretical findings with six experiments on three MFGs.

## 1 Introduction

Mean field games (MFGs) (Huang et al., 2006; Lasry & Lions, 2007) offer a tractable framework for modeling multi-agent systems with a large homogeneous population. In MFGs, the impact of other agents on a particular agent is encapsulated by a *population mass* (or *mean field*), providing a reliable approximation of actual interactions among agents when the number of agents is large. Consequently, understanding the *population* is fundamental in MFGs, as learning these games entails considering both the agent's policy and the population dynamics.

Prior work on learning MFGs has mainly focused on fixed-point iteration (FPI) methods and their variations, which is characterized by a *forward-backward* procedure that *alternately* calculate the policy update w.r.t. a *fixed* population and the induced population distribution w.r.t. a *fixed* policy (Guo et al., 2019; Elie et al., 2020; Perrin et al., 2020; Xie et al., 2021; Cui & Koeppl, 2021; Laurière et al., 2022b; Anahtarci et al., 2023). However, FPI-type methods face several limitations: 1) The policy learning and population learning in FPI-type methods typically involves distinct iterative processes, and are implemented separately and executed sequentially, potentially increasing the overall *computational burden* (Mao et al., 2022; Zaman et al., 2023). 2) Vanilla FPI methods suffer from *instability*. As the policy or population is fixed while updating the other, the differences between updates in consecutive iterations can be drastic, causing oscillations in the learning process, a phenomenon commonly observed in practice (Cui & Koeppl, 2021). 3) Separating the forward and backward processes prevents us from applying abundant methods developed for policy learning, such as function approximation (Mao et al., 2022; Huang et al., 2024), to population learning.

This work delves into the rapidly growing field of online learning for MFGs (Mao et al., 2022; Angiuli et al., 2022; Zaman et al., 2023; Yardim et al., 2023; Zhang et al., 2024a; Zeng et al., 2025), where an online agent interacts with the environment and gathers observations on a Markov chain to update its policy and population estimate. A key aspect of our approach is recognizing that the Markov chain is jointly *parameterized* by both the policy and population distribution, which together define the transition kernel. From this perspective, finding the mean field equilibrium (MFE) becomes a problem of identifying the *optimal parameter*—the fixed point of the Bellman and transition operators. This viewpoint unlocks a large toolbox of stochastic parameter approximation methods for learning MFGs, including simple stochastic gradient descent (SGD)-type methods. Specifically, by treating the policy and population as a *unified parameter*, we update both estimates *fully asynchronously* using the same batch of samples, thereby eliminating the need for a forward-backward process.

Building on this perspective, we introduce another powerful technique: linear function approximation (LFA), a widely used approach in stochastic approximation methods for policy learning on large or continuous state-action spaces (Melo et al., 2008; Jin et al., 2020). By applying LFA to the unified parameter—encompassing both the policy and population—we develop the first *population-aware* LFA (PA-LFA) for learning MFGs on continuous state-action spaces. This is particularly significant, as many common MFGs, such as autonomous driving (Huang et al., 2021; Chen et al., 2023a;b; Mo et al., 2024), flocking (Perrin et al., 2021), and crowd modeling (Lachapelle & Wolfram, 2011; Burger et al., 2013), inherently involve continuous state-action spaces. With PA-LFA, our method enables a simple, fully online, model-free SGD-type method for learning MFGs in these settings, supported by strong theoretical guarantees.

While elegant, treating the policy and population as a unified parameter raises several questions: 1) Updating policy and population estimates fully asynchronously creates a *strong coupling* between the two, which is absent in forward-backward or multi-loop structures (Yardim et al., 2023; Zhang et al., 2024a) or is alleviated in multi-timescale approaches (Mao et al., 2022; Angiuli et al., 2023; Zaman et al., 2023). How can we analyze this coupling and the non-stationary Markov chain it generates to establish convergence guarantees? 2) Empirically, our method—a simple SGD-type method without additional stabilization mechanisms—outperforms vanilla FPI in *stability* and *efficiency* (Section 7). What theoretical explanation underpins this? 3) How can we design PA-LFA for continuous state-action spaces such that each population update maintains comparable *operational complexity* to that in the finite state-action case? 4) How does PA-LFA influence convergence? Under what conditions does it converge to an MFE, and if it doesn't, how large is the *approximation error*?

**Main results.** With the above questions in mind, we highlight the key contributions of this work:

- We propose SemiSGD, a simple online SGD-type method for learning MFGs. We innovatively treat the policy and the population as a unified parameter. Algorithmically, we update both simultaneously and asynchronously using the same online observations and learning rate, thus eliminating the forward-backward process typical of FPI methods. Theoretically, we show that the unified parameter in SemiSGD follows a *descent direction* towards the MFE, whereas neither the policy nor population alone is guaranteed to do so (Lemma 1). This gives a potential explanation for the stability and efficiency of SemiSGD over vanilla FPI (Section 7).

- We formulate a novel framework of linear MFGs, characterizing a class of MFGs with linear structure concerning the population measure. Linear MFGs accommodates continuous state-action spaces and includes MFGs on finite state-action spaces as a special case. We prove that linear MFGs enable linear parameterization of both the value function and the population measure.

- We extend the linear parameterization to develop a *population-aware linear function approximation (PA-LFA)* for general MFGs. SemiSGD equipped with PA-LFA is the first method to apply LFA to the population measure in MFGs. Notably, updates in SemiSGD with PA-LFA maintain the same operation complexity as in the finite state-action space case, highlighting the simplicity and efficiency of our method.

- Finite-time convergence analysis is provided for SemiSGD with PA-LFA. We novelly regard the learning process as a *stochastic approximation on a non-stationary Markov chain parameterized by the unified parameter*. This perspective enables a straightforward SGD-type analysis, elegantly handling the strong coupling between the policy and population and offering insights into the learning dynamics of SemiSGD. We prove that, under a contractivity condition no stronger than prior work, SemiSGD converges to the MFE. The contractivity condition can be hard to

verify in practice and potentially implies large regularization. In response, we propose a new condition that is more practical. This condition allows general non-regularized policies, under which SemiSGD converges to a neighborhood centered at the MFE. In both scenarios, SemiSGD enjoys state-of-the-art sample complexities.

- For non-linear MFGs, SemiSGD with PA-LFA converges to the *projected* MFE (or its neighborhood). We characterize the distance between this projected MFE and the actual MFE in terms of the intrinsic approximation error in the linear representation.

- We conduct six experiments on three MFGs to demonstrate the properties of our methods: 1) For continuous state-action spaces, PA-LFA is more efficient and accurate than discretization; 2) SemiSGD automatically stabilizes without additional mechanics, achieves a higher accuracy, and is faster by eliminating the forward-backward procedure.

**Related work.** Learning MFGs has garnered increasing attention, with recent comprehensive reviews by Laurière et al. (2022a); Cui et al. (2022). To address the instability issue of FPI, various stabilization mechanisms have been proposed, broadly categorized into: 1) regularization (Cui & Koeppl, 2021; Anahtarci et al., 2023); 2) fictitious play (Elie et al., 2020; Perrin et al., 2020; Cardaliaguet & Hadikhanloo, 2017); and 3) mirror descent (Perolat et al., 2021; Xie et al., 2021; Yardim et al., 2023; Laurière et al., 2022b). Recently, Angiuli et al. (2022); Zaman et al. (2023); Yardim et al. (2023); Zhang et al. (2024a) developed online oracle-free methods for the population learning in MFGs, where a population estimate is updated using online observations without requiring an oracle to manipulate the population or directly return the population measure. Angiuli et al. (2023); Zaman et al. (2023); Mao et al. (2022); Zeng et al. (2025) further proposed single-loop multi-timescale schemes to asynchronously update the policy and population estimates. On the policy learning side, Mao et al. (2022); Huang et al. (2024) considered function approximation, though they both assume access to the population measure rather than learning it. An extended literature review with detailed comparisons of existing setups and methods is provided in Appendix A.

**Notation.** A complete list of symbols and their meanings is provided in Table 3. We denote the set of probability measures on a space $\mathcal{X}$ as $\mathcal{D}(\mathcal{X})$, and the set of signed measures as $\mathcal{M}(\mathcal{X})$. When $\mathcal{X}$ is finite with $d$ elements, the probability simplex on it is denoted as $\Delta^d \subset \mathbb{R}^d$. $\delta_x$ is the Dirac delta measure at $x$. Without subscript, the inner product denotes the vector inner product $\langle x, y \rangle = x^T y$. $x \oplus y$ is the direct sum of elements in linear spaces, which reduces to their concatenation $(x; y)$ when $x$ and $y$ are vectors. The norm without subscript denotes the $\ell_2$ norm for vectors, and $\ell_2$ operator norm for matrices. For (vector-valued) functions on $\mathcal{S}$, the $L_2$ inner product is defined as $\langle f, g \rangle_{L_2} = \int_{\mathcal{S}} f(s)g(s)\mathrm{d}s$, and the $L_1$ norm is denoted as $\|f\|_{\mathrm{TV}} = \int_{\mathcal{S}} \|f(s)\|_1 \mathrm{d}s$.

## 2 STOCHASTIC SEMI-GRADIENT DESCENT FOR MFGS ON FINITE STATE-ACTION SPACES

### 2.1 REVISIT ONLINE LEARNING FOR MFGS ON FINITE STATE-ACTION SPACES

We consider an infinite-horizon Markov decision process (MDP) denoted by $(\mathcal{S}, \mathcal{A}, r, P, \gamma)$, with the state space $\mathcal{S}$ and action space $\mathcal{A}$ being finite. In MFGs, the reward function $r$ and transition kernel $P$ depend on the population distribution over the state space. Specifically, for a given state-action pair $(s, a) \in \mathcal{S} \times \mathcal{A}$ and population distribution $\mu \in \mathcal{D}(\mathcal{S})$, $r(s, a, \mu)$ and $P(s' \mid s, a, \mu)$ denote the reward received and the probability of transitioning to state $s' \in \mathcal{S}$. We consider a bounded reward function with $\|r\|_\infty \leq R$. $\gamma$ is the discount factor.

An agent in an MFG aims to find a policy $\pi$, which maps a state to a distribution on the action space determining the agent's actions, that maximizes its expected cumulative discounted reward while interacting with the population. We utilize a value-based approach to calculate policies and assume access to a policy operator $\Gamma_\pi$ (Zou et al., 2019; Zhang et al., 2024a) that returns a policy based on an (action-)value function $Q \colon \mathcal{S} \times \mathcal{A} \to \mathbb{R}$. We write $\pi_Q := \Gamma_\pi(Q)$. We define two operators for any value functions $Q, Q' \in \mathbb{R}^{|\mathcal{S}| \times |\mathcal{A}|}$ and population measures $M, M' \in \mathcal{D}(\mathcal{S})$:

$$\mathcal{T}_{(Q,M)}Q'(s,a) := \mathbb{E}_{(Q,M)}\left[r(s,a,M) + \gamma Q'(s',a')\right], \text{ with } a' \sim \pi_Q, s' \sim P(\cdot \mid s,a,M) \quad \text{(Bellman)}$$

$$\mathcal{P}_{(Q,M)}M'(s') := \mathbb{E}_{(M',Q)}\left[P(s' \mid s,a,M)\right], \text{ with } s \sim M', a \sim \pi_Q. \quad \text{(Transition)}$$

Then our learning goal, the mean field equilibrium (MFE), is defined as the fixed point of these two operators. With an argmax policy operator, the fixed point of the Bellman operator satisfies the Bellman optimality equation, leading to the standard MFE definition in the reward-maximizing setting (Laurière et al., 2022b; Angiuli et al., 2023); with a regularized policy operator, e.g., softmax, the fixed point corresponds to the regularized MFE (Cui & Koeppl, 2021; Zaman et al., 2023). See Appendix B.1 for more discussions on our MFE definition.

**Definition 1** (Mean field equilibrium). A value function-population distribution pair $(Q, M)$ is a *mean field equilibrium (MFE)* if it satisfies $Q = \mathcal{T}_{(Q,M)}Q$ and $M = \mathcal{P}_{(Q,M)}M$.

A typical FPI-type method (approximately) calculates the fixed point of the Bellman operator given the current population distribution, i.e., the best response, and the fixed point of the transition operator given the current value function, i.e., the induced population distribution, alternately. Under certain contractivity conditions, FPI converges to the MFE (Guo et al., 2019).

This work focuses on model-free online learning methods for MFGs. In such methods, an agent maintains estimates about the value function and population measure, and uses online observations to update its estimates. For the value function, temporal difference (TD) methods (Sutton & Barto, 2018) are widely used. Given an online observation tuple $O = (s, a, s', a')$ and step-size $\alpha$, on-policy TD (or SARSA (Sutton & Barto, 2018)) updates the Q-value function as follows:

$$Q(s,a) \leftarrow Q(s,a) - \alpha \mathfrak{g}(Q;O), \text{ with } \mathfrak{g}(Q;O) \coloneqq Q(s,a) - r(s,a,M) - \gamma Q(s',a'). \quad (1)$$

For the population distribution, Monte Carlo (MC) sampling (Łatuszyński et al., 2013) is a common choice. Given a new state sample $s'$ and step-size $\alpha$, MC updates the population measure as follows:

$$M \leftarrow M - \alpha \mathfrak{g}(M;O), \text{ with } \mathfrak{g}(M;O) \coloneqq M - \delta_{s'}, \quad (2)$$

where $\delta_{s'}$ is the Dirac delta measure and is a one-hot vector for finite state spaces.

TD and MC updates are widely used in online learning for MFGs (Angiuli et al., 2022; Zaman et al., 2023; Zhang et al., 2024a). Fixing the policy and population measure (i.e., fixing the Markov kernel), TD converges to the optimal value function, and MC converges to the induced population distribution.

## 2.2 STOCHASTIC SEMI-GRADIENT DESCENT

Combining (1) and (2) in a single pass gives a simple SGD-type method. $\mathfrak{g}$ in (1) and (2) is referred as semi-gradients, as it does not represent the actual gradient of any loss function but provides a similar descent direction to the *stationary point*, i.e., the zero point of $\mathfrak{g}$.[1] We term our method SemiSGD and present it in Algorithm 1, where $(Q_t; M_t)$ denotes the concatenated representation of the value function and population measure, as they are both vectors for finite state-action spaces. Similarly, $\mathfrak{g}$ represents the concatenation of the semi-gradients for the value function and population measure.

---

**Algorithm 1:** Stochastic semi-gradient descent (SemiSGD)

---

1 **input**: Initial value function $Q_0(= \langle \phi, \theta_0 \rangle)$, population estimate $M_0(= \langle \psi, \eta_0 \rangle)$, and state $s_0$.
2 **for** $t = 0, 1, \ldots, T$ **do**
3      Observe $a_t \sim \Gamma_\pi(Q_t)[s_t], r_t = r(s_t, a_t, M_t), s_{t+1} \sim P(\cdot \mid s_t, a_t, M_t), a_{t+1} \sim \Gamma_\pi(Q_t)[s_{t+1}]$.
4      Update for finite MFGs: $(Q_{t+1}; M_{t+1}) = (Q_t; M_t) - \alpha_t \mathfrak{g}((Q_t; M_t); s, a, s', a')$ ;
5         or with PA-LFA: $\xi_{t+1} = \Pi(\xi_t - \alpha_t \mathfrak{g}_t(\xi_t))$.
6 **return** $(Q_T, M_T)$ or $\xi_T$.

---

Algorithm 1 is an online, single-loop, uni-timescale method that is free from any forward-backward process. Not only does it update the value function and population measure estimates in an SGD-like manner, but it provides an actual descent direction through the *mean-path* semi-gradient, defined as $\bar{\mathfrak{g}}_{(Q,M)}(\cdot) \coloneqq \mathbb{E}_{(Q,M)}\mathfrak{g}(\cdot; O)$, where $O$ is the online observation tuple following the steady distribution induced by the policy and transition kernel determined by $Q$ and $M$.

---

[1] With a slight abuse of notation, we use a single operator $\mathfrak{g}$ to return semi-gradients throughout the paper, for both the action-value function and population measure; it should be clear from the context and its arguments which one it refers to.

**Lemma 1** (Descent direction; informal). *Suppose $(Q_*; M_*)$ is an MFE. Suppose the reward function, transition kernel, and policy operator are Lipschitz continuous with Lipschitz constant $L$. For any value function $Q$ and population measure $M$, with $\Delta Q := Q - Q_*$ and $\Delta M := M - M_*$, we have*

$$-\left\langle \Delta M, \bar{\mathfrak{g}}_{(Q,M)}(M)\right\rangle \lesssim -\|\Delta M\|^2 + L\|\Delta M\|(\|\Delta Q\| + \|\Delta M\|),$$

$$-\left\langle \Delta Q, \bar{\mathfrak{g}}_{(Q,M)}(Q)\right\rangle \lesssim -\|\Delta Q\|^2 + L\|\Delta Q\|(\|\Delta Q\| + \|\Delta M\|).$$

*Due to the coupling between $Q$ and $M$, neither $-\bar{\mathfrak{g}}_{(Q,M)}(M)$ nor $-\bar{\mathfrak{g}}_{(Q,M)}(Q)$ is guaranteed to be a descent direction, no matter how small the Lipschitz constant $L$ is. However, if $L \lesssim 1/2$, adding the two inequalities gives*

$$-\left\langle (Q; M) - (Q_*; M_*), \bar{\mathfrak{g}}_{(Q,M)}((Q; M))\right\rangle \lesssim -\|(Q; M) - (Q_*; M_*)\|^2/2.$$

A formal and generalized version of Lemma 1 for linear MFGs, which encompasses finite MFGs (Example 2), is proved in Appendix H.1. We now discuss some key insights from Lemma 1.

**Descent direction.** The concatenated semi-gradient points to a descent direction for the concatenated estimate $(Q; M)$, while neither the value function nor population measure alone is guaranteed to follow a descent direction, no matter how small the Lipschitz constant is when far from equilibrium (i.e., large $\|M - M_*\|$ and $\|Q - Q_*\|$). This strongly suggests treating the value function and population measure as a unified parameter and updating them simultaneously at the same rate. The small Lipschitz constant condition $L \lesssim 1/2$ in the lemma, formalized as Assumption 4 in Section 5, ensures contractivity. Thus, while both methods converge under the contractivity assumption, SemiSGD follows a descent direction but FPI may not, hinting the *sample efficiency* of SemiSGD. See Figure 1 for an illustration.

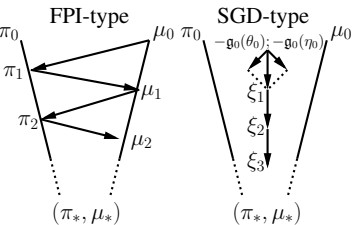

Figure 1: Learning dynamics.

**Incremental update by design.** Incrementally updating parameters or damping update steps, rather than switching to entirely new estimates, is a common stabilization technique in learning MFGs (Laurière et al., 2022a), used in methods like fictitious play and mirror descent (Laurière et al., 2022b). As an SGD-type method, updates in SemiSGD are incremental by design. This, combined with the previous point, suggests that SemiSGD enjoys *automatic stabilization* without needing additional stabilization mechanisms, as confirmed by our experiments (Section 7). These two insights also offer a potential explanation for the oscillation issues seen in FPI-type methods.

**Stochastic approximation on non-stationary Markov chains.** The lemma highlights that the value function and population measure jointly shape the landscape of learning MFGs and should be treated as a unified parameter. SemiSGD naturally fits as a *stochastic approximation* method on a *non-stationary* Markov chain for finding the equilibrium parameter, with non-stationarity arising from the changing transition kernel as the value function and population measure are updated. Two-timescale approaches are used to mitigate the non-stationarity and parameter coupling (Angiuli et al., 2023; Zaman et al., 2023; Mao et al., 2022). Using a unified parameter, our approach enables a simpler uni-timescale scheme with a straightforward SGD-type analysis that accounts for coupling and achieves better sample complexities (see Table 2). Moreover, for general policy operators, which may fail the small Lipschitz constant condition, the stochastic approximation is on a *rapidly changing* Markov chain (Zhang et al., 2023). Building on the results for this class of methods, we are able to, for the first time, characterize the finite-time convergence performance of learning MFGs without a contractivity or monotonicity condition.

## 3 LINEAR MEAN FIELD GAMES

We now extend our setup beyond finite state-action spaces. Viewing the value function and population measure as a unified parameter, we can naturally extend linear MDP (Jin et al., 2020) and LFA for the value function to handle population measures on large or continuous state spaces. Before extending LFA to population measures in the next section, we first introduce linear MFGs, a class of MFGs based on linear MDPs with a linear structure in the transition kernel w.r.t. the population measure.

**Definition 2** (Linear mean field games). An MDP $(\mathcal{S}, \mathcal{A}, P, r, \gamma)$ is a linear MDP (Jin et al., 2020) with feature map $\phi: \mathcal{S} \times \mathcal{A} \to \mathbb{R}^{d_1}$ if there exists $d_1$ (signed) population-dependent measures

$\omega_M = (\omega_M^{(i)})_{i=1}^{d_1} \in \mathcal{M}(\mathcal{S})^{d_1}$ and an *unknown* population-dependent vector $\nu_M \in \mathbb{R}^{d_1}$, such that for any state-action pair $(s,a) \in \mathcal{S} \times \mathcal{A}$ and population distribution $M \in \mathcal{D}(\mathcal{S})$, we have

$$P(s' \,|\, s, a, M) = \langle \phi(s,a), \omega_M(s') \rangle, \qquad r(s,a,M) = \langle \phi(s,a), \nu_M \rangle.$$

A linear MFG further assumes a measure basis $\psi \in \mathcal{D}(\mathcal{S})^{d_2}$ such that for any population $M$, there exists an *unknown* matrix $\Omega_M \in \mathbb{R}^{d_1 \times d_2}$ such that $\omega_M = \Omega_M \psi$, indicating that $P(s' \,|\, s, a, M) = \langle \phi(s,a), \Omega_M \psi(s') \rangle$. We require $\phi$ to be $L_\infty$ and $\psi$ to be $L_\infty$ and $L_2$ (thus its Gram matrix exists). Without loss of generality, we assume $\sup_{s,a} \|\phi(s,a)\|_2 \leq 1$ and $\sup_s \|\psi(s)\|_1 \leq F$.

A linear MFG assumes MDP components, the transition kernel and reward function, lie in the linear span of known basis functions, giving a linear structure to the value function and population measure.

**Proposition 1.** *For a linear MFG, for any population distribution $M$ and policy $\pi$, we denote $Q_M^\pi$ as the action-value function and $\mu_M^\pi$ as the induced population measure w.r.t. the MDP determined by $M$ and $\pi$. Then, there exist vector parameters $\theta \in \mathbb{R}^{d_1}$ and $\eta \in \mathbb{R}^{d_2}$ such that*

$$Q_M^\pi(s,a) = \langle \phi(s,a), \theta \rangle, \qquad \mu_M^\pi(s') = \langle \psi(s'), \eta \rangle, \qquad \forall (s,a,s') \in \mathcal{S} \times \mathcal{A} \times \mathcal{S}.$$

The proof of Proposition 1 is deferred to Appendix E. Remarkably, requiring that the transition kernel being linear w.r.t. a measure basis is essential for the linear structure of the population measure. In this paper, we reserve letters $\theta$ for the value function parameter and $\eta$ for the population measure parameter. And we denote the *concatenated parameter* as $\xi = (\eta; \theta)$. Additionally, we will use parameters $(\eta, \theta, \xi)$ and their corresponding functions $(Q, M, (Q, M))$ interchangeably in our analysis. For example, we say $\xi \in \mathbb{R}^{d_1 + d_2}$ is a mean field equilibrium (MFE) if its corresponding value function $Q = \langle \phi, \theta \rangle$ and population measure $M = \langle \psi, \eta \rangle$ satisfy Definition 1.

**Example 1** (Linear MDP plus population-independent transition kernel). MFGs with a linear MDP and population-independent transition kernel are a trivial example of linear MFGs, where $\Omega_M = I$ for all $M \in \mathcal{D}(\mathcal{S})$.

**Example 2** (Finite state-action space). For finite MFGs, let the feature map $\phi$ return the one-hot vector of each state-action pair, i.e., $\phi(s,a) = e_{(s,a)} \in \mathbb{R}^{|\mathcal{S}||\mathcal{A}|}$, and the measure basis $\psi$ return the Dirac delta measure at each state, i.e., $\psi(s') = \delta_{s'} \in \Delta^{|\mathcal{S}|}$. Construct the matrix $\Omega_M \in \mathbb{R}^{|\mathcal{S}||\mathcal{A}| \times |\mathcal{S}|}$ such that $(\Omega_M)_{(s,a),s'} = P(s' \,|\, s, a, M)$. Similarly, construct the vector $\nu_M \in \mathbb{R}^{|\mathcal{S}||\mathcal{A}|}$ such that $(\nu_M)_{(s,a)} = r(s,a,M)$. Then, for any $(s,a,s') \in \mathcal{S} \times \mathcal{A} \times \mathcal{S}$, we have

$$P(s' \,|\, s, a, M) = e_{(s,a)}^T \Omega_M \delta_{s'} = \langle e_{(s,a)}, \Omega_M \delta(s') \rangle, \quad r(s,a,M) = e_{(s,a)}^T \nu_M = \langle e_{(s,a)}, \nu_M \rangle.$$

Example 2 implies that all the analysis for linear MFGs applies to finite MFGs (see also Appendix K).

## 4 SEMISGD WITH POPULATION-AWARE LINEAR FUNCTION APPROXIMATION

Proposition 1 presents the first linear parameterization of the population measure in MFGs. Applying this to general MFGs leads to population-aware linear function approximation (PA-LFA). More discussions on motivations of PA-LFA can be found in Appendix B.2. PA-LFA necessitates a tailored stochastic update rule for the population measure estimate. Let $M_*$ be the objective population measure and define the loss function $\mathcal{L} := \frac{1}{2}\|M - M_*\|_{L_2}^2$. Then, the gradient of the loss is

$$\nabla_\eta \mathcal{L} = \langle \nabla_\eta M, M - M_* \rangle_{L_2} = \langle \psi, \langle \psi, \eta \rangle - M_* \rangle_{L_2}.$$

As the MFE population measure $M_*$ is unknown, we replace it with the empirical Delta distribution (bootstrapping), giving the semi-gradient

$$\mathfrak{g}(\eta; s') = \langle \psi, \langle \psi, \eta \rangle - \delta_{s'} \rangle_{L_2} = \int_{\mathcal{S}} \psi(s)\psi(s)^T \eta \, \mathrm{d}s - \int_{\mathcal{S}} \psi(s)\delta_{s'}(s) \, \mathrm{d}s =: G_\psi \eta - \psi(s'), \quad (3)$$

where $G_\psi := \int_{\mathcal{S}} \psi(s)\psi(s)^T \mathrm{d}s$ is the Gram matrix of measure basis $\psi$.

For finite MFGs, (2) retains the updated population measure as a probability vector, which is not necessarily the case when using (3) as the semi-gradient, due to the presence of general $G_\psi$ and $\psi(s')$ that may not be an identity matrix or a probability vector. Therefore, we need to apply a projection to the updated parameter to ensure that it remains a probability vector, giving our stochastic update rule:

$$\eta_{t+1} = \Pi_\Delta(\eta_t - \alpha_t \mathfrak{g}_t(\eta_t)), \quad (4)$$

where $\mathfrak{g}_t(\eta) := \mathfrak{g}(\eta; s_{t+1})$ and $\Pi_\Delta$ is the projection operator onto the probability simplex $\Delta^{d_2}$.

*Remark* 1 (Operation complexity). The simplex projection has a worst-case complexity of $O(d_2^2)$ (Condat, 2016). In semi-gradient evaluation, $G_\psi$ is a constant matrix, and thus we only need to evaluate $\psi$ at the next state $s'$ and multiply parameter with a fixed matrix, with the matrix-vector multiplication having a complexity of $O(d_2^2)$. Therefore, the total worst-case operation complexity of the update rule (4) is $O(d_2^2)$. Moreover, the simplex projection has an expected complexity of $O(d_2)$ (Condat, 2016). And if $G_\psi$ has precomputed properties and/or special structures, such as being sparse with $O(d_2)$ non-zero elements, the complexity of the semi-gradient evaluation is $O(d_2)$, giving a total expected operation complexity of $O(d_2)$, which is the same as that of (2) for finite MFGs with a state space of size $|\mathcal{S}| = O(d_2)$.

Similarly, we can get the TD update rule for the value function parameter $\theta$ (Sutton & Barto, 2018):

$$\mathfrak{g}(\theta; O) = \phi(s, a)\left(\langle\phi(s, a) - \gamma\phi(s', a'), \theta\rangle - r\right) =: G_\phi(O)\theta - \varphi(O),$$

where $O = (s, a, r, s', a')$, and $G_\phi(O) := \phi(s, a)(\phi(s, a) - \gamma\phi(s', a'))^T$ and $\varphi(O) := \phi(s, a)r$. The update rule for the action-value function parameter is

$$\theta_{t+1} = \Pi_D(\theta_t - \alpha_t \mathfrak{g}_t(\theta_t)), \tag{5}$$

where we denote $\mathfrak{g}_t(\theta) := \mathfrak{g}(\theta; O_t)$, $O_t = (s_t, a_t, r_t, s_{t+1}, a_{t+1})$, and $\Pi_D$ is the projection operator onto the Euclidean ball $B_D^{d_1} := \{\theta \in \mathbb{R}^{d_1} : \|\theta\| \le D\}$. Similar to the population measure update, the projection is commonly used with LFA to ensure that the value function parameter remains bounded (Bhandari et al., 2018; Zou et al., 2019), which is automatically satisfied in finite MFGs (Appendix K). Specifically, we need $D \ge \|\theta_*\|$, where $\theta_*$ is the MFE value function parameter.

Combining (4) and (5) gives the update rule for the unified parameter:

$$\xi_{t+1} = \Pi(\xi_t - \alpha_t \mathfrak{g}_t(\xi_t)) := \Pi_{B_D^{d_1} \times \Delta^{d_2}}\left(\xi_t - \alpha_t\left((G_\phi(O_t) \oplus G_\psi)\xi_t - (\varphi \oplus \psi)(O_t)\right)\right). \tag{6}$$

(6) updates the action-value function and population measure using the same online observations with the same learning rate. Replacing Line 3 in Algorithm 1 with (6) gives SemiSGD with LFA.

The next section shows that SemiSGD converges to a zero point of the mean-path semi-gradient $\bar{\mathfrak{g}}_\xi(\xi) := \mathbb{E}_{O \sim \mu_\xi} \mathfrak{g}(\xi; O)$, where $\mu_\xi$ is the observation distribution induced by the parameter $\xi$. The next proposition states that this point is an MFE, thus validating the derivation in this section.

**Proposition 2** (MFE as a stationary point). *For linear MFGs, $\xi$ is an MFE if and only if $\bar{\mathfrak{g}}_\xi(\xi) = 0$.*

We prove an extended version of Proposition 2 in Appendix F, showing that the stationary point actually corresponds to a *projected* MFE for general (non-linear) MFGs.

## 5 SAMPLE COMPLEXITY ANALYSIS

This section provides a finite-sample analysis of the convergence of SemiSGD with LFA, thus covering finite MFGs as a special case. We denote $P_M(s' \mid s, a) := P(s' \mid s, a, M)$ and $r_M(s, a) := r(s, a, M)$ for short. For transition kernels, we define the total variation operator norm $\|P\|_{TV} := \sup_{p \in \mathcal{M}(\mathcal{S} \times \mathcal{A}), \|p\|_{TV} \le 1} \|\int P(\cdot \mid s, a)p(\mathrm{d}s, \mathrm{d}a)\|_{TV}$. We now state the assumptions for the analysis.

**Assumption 1** (Lipschitz MDP). The transition kernel and reward function are Lipschitz continuous w.r.t. the population measure. That is, there exists positive constants $L_P$ and $L_r$ such that for any state-action pair $(s, a)$ and population measures $M_1, M_2$, we have

$$\|P_{M_1} - P_{M_2}\|_{TV} \le (L_P/\sqrt{d_2})\|M_1 - M_2\|_{TV}, \qquad \|r_{M_1} - r_{M_2}\|_\infty \le (L_r/\sqrt{d_2})\|M_1 - M_2\|_{TV}.$$

**Assumption 2** (Lipschitz policy operator). There exists a constant $L_\pi$ such that for any state $s$ and value functions $Q_1, Q_2$, we have $\|\Gamma_\pi(Q_1)(\cdot \mid s) - \Gamma_\pi(Q_2)(\cdot \mid s)\|_{TV} \le L_\pi\|Q_1 - Q_2\|_\infty$.

**Assumption 3** (Uniform ergodicity). The MDP is uniformly ergodic for any parameter any value function $Q$ and population measure $M$. That is, there exists constants $m \ge 1$, $\rho \in (0, 1)$, and $\mu_{(Q,M)}$, such that for any initial distribution $M_0 \in \mathcal{D}(\mathcal{S})$, it holds that $\|\mu_{(Q,M)} - \mathcal{P}_{(Q,M)}^t M_0\|_{TV} \le m\rho^t$.

For notational convenience, we define an ergodicity constant $\sigma := 2 + \hat{n} + m\rho^{\hat{n}}/(1 - \rho)$ with $\hat{n} := \lceil\log_\rho m^{-1}\rceil$, and $H := (1 + \gamma)D + R + 2F$, which can be regarded as the scale of the problem.

**Assumption 4.** The Lipschitz constants are sufficiently small such that $3\sigma \max\{L_P, L_\pi\}H + L_r \leq 2w$, where $w \in (0, 1/2]$ is a problem-dependent constant (defined in Lemma 9 in Appendix H.1)

*Remark* 2. Assumption 1 is a standard regularity condition for MFGs that do not assume a blanket contractivity assumption (Cui & Koeppl, 2021; Angiuli et al., 2023; Anahtarci et al., 2023; Yardim et al., 2023; Huang et al., 2024). Assumption 2 ensures the smoothness of policy updates, a condition typically met through regularization. Example policy operators satisfying Assumption 2 include softmax (Cui & Koeppl, 2021; Zaman et al., 2020; Angiuli et al., 2023), mirror descent (Perolat et al., 2021; Laurière et al., 2022b), and mirror ascent (Yardim et al., 2023). Assumption 3 is a standard mixing assumption for online methods with Markovian sampling (Bhandari et al., 2018; Zou et al., 2019; Angiuli et al., 2022; Zaman et al., 2023), ensuring that the agent's exploration adequately covers the state-action space. Notably, Assumption 3 is also typically satisfied by regularizing behavior policies (Yardim et al., 2023). It is noteworthy that even earlier works using a blanket contractivity assumption recognize that contractivity can be achieved through regularization, provided that "the transition kernel and reward function are Lipschitz and the corresponding Lipschitz constants are small enough" (Xie et al., 2021). Therefore, Assumption 4 is the key assumption that guarantees the contractivity of learning dynamics. In summary, our assumptions closely align with those in Angiuli et al. (2023); Anahtarci et al. (2023), and are not more restrictive than those in the literature for contractive MFGs. See also Appendix I for convergence results in the absence of Assumption 4.

We can now bound the mean squared error of SemiSGD recursively, which directly gives several finite-time error bounds.

**Theorem 1** (One-step progress). *Let $\xi_* = (\theta_*; \eta_*)$ be an MFE parameter. Let $\{\xi_t = (\theta_t; \eta_t)\}$ be a sequence of parameters generated by SemiSGD. Then, under Assumptions 1 to 4, we have*

$$\mathbb{E}\|\xi_{t+1} - \xi_*\|^2 \leq (1 - \alpha_t w)\mathbb{E}\|\xi_t - \xi_*\|^2 + H^2 \cdot O(\alpha_t^2 \log \alpha_t^{-1}) + \frac{\max\{L_P, L_\pi, L_r\}^2 H^4}{w} \cdot O(\alpha_t^3 \log^4 \alpha_t^{-1}).$$

**Corollary 1** (Constant step-size). *Given a constant step-size $\alpha_t \equiv \alpha_0$, we have*

$$\mathbb{E}\|\xi_T - \xi_*\|^2 \leq e^{-\alpha_0 wT}\|\xi_0 - \xi_*\|^2 + w^{-1}H^2 \cdot O(\alpha_0 \log \alpha_0^{-1}).$$

Let $\alpha_0 = (\log T)/(wT)$. Then Corollary 1 states $\mathbb{E}\|\xi_T - \xi_*\|^2 = O\left((H^2 \log^2 T)/(w^2 T)\right)$, giving an $O(\epsilon^{-2} \log^2 \epsilon^{-1})$ sample complexity for an $\epsilon$-MFE ($\mathbb{E}\|\xi_T - \xi_*\| \leq \epsilon$). A linearly decaying step-size sequence improves the logarithmic factor, giving a convergence rate of $O\left((H^2 \log T)/(w^2 T)\right)$ (see Corollary 3). Reducing to finite MFGs, the complexity becomes $\widetilde{O}\left(\frac{SAR^2}{\lambda^2(1-\gamma)^4 T}\right)$ (see Corollary 4). These results also imply the uniqueness of the MFE under the assumptions of Theorem 1. The sample complexity is tight up to a logarithmic factor, consistent with other stochastic approximation methods, and strictly better than existing results for learning MFGs (Table 2). See Appendix B.3 for more discussions.

## 6 APPROXIMATION ERROR FOR NON-LINEAR MFGS

SemiSGD with PA-LFA applies to non-linear MFGs as long as the feature map $\phi$ and measure basis $\psi$ are specified. However, for non-linear MFGs, it remains unclear: 1) What point does SemiSGD with PA-LFA converge to? 2) What factors characterize the approximation error caused by PA-LFA?

For the first question, it turns out that the convergence point of SemiSGD with PA-LFA, i.e., the zero point of the mean-path semi-gradient developed in Section 4, corresponds to a *projected* MFE $\langle\theta, \eta\rangle$ defined by $\langle\theta, \phi\rangle = \Pi_\phi \mathcal{T}_\xi \langle\theta, \phi\rangle$ and $\langle\eta, \psi\rangle = \Pi_\psi \mathcal{P}_\xi \langle\eta, \psi\rangle$, where $\Pi_\phi$ and $\Pi_\psi$ are orthogonal projection operators onto the linear spans of $\phi$ and $\psi$, respectively. The formal definitions and proof of the statement are deferred to Appendix F. For linear MFGs, as images of $\mathcal{T}$ and $\mathcal{P}$ are within the linear spans of $\phi$ and $\psi$ (Appendix E), the projected MFE coincides with the MFE itself (Proposition 2). The following theorem answers the second question.

**Theorem 2** (Approximation error). *Let $\xi_\diamond = (\theta_\diamond; \eta_\diamond)$ be the convergence point of SemiSGD with PA-LFA. Let $(q_*, \mu_*)$ be the actual MFE. Write $\mathcal{P}_\diamond := \mathcal{P}_{\xi_\diamond}$ for short, and similarly for $\mathcal{P}_*, \mathcal{T}_\diamond,$ and $\mathcal{T}_*$.*

*Then, Assumption 3 indicates the existence of $k \in \mathbb{N}$ such that*

$$\epsilon_q := \|q_* - \langle \phi, \theta_\diamond \rangle\|_\infty \leq \left(\gamma + \frac{\gamma \sigma R L_\pi}{1 - \gamma}\right) \epsilon_q + \left(L_r + \frac{\gamma \sigma R L_P}{(1 - \gamma)\sqrt{d_2}}\right) \epsilon_\mu + \|q_* - \Pi_\phi q_*\|_\infty$$

$$\epsilon_\mu := \|\mu_* - \langle \psi, \eta_\diamond \rangle\|_{\mathrm{TV}} \leq \left(\rho^k + \frac{k L_P}{\sqrt{d_2}}\right) \epsilon_\mu + k L_\pi \epsilon_q + k \|\mathcal{P}_* - \Pi_\psi \mathcal{P}_*\|_{\mathrm{TV}}.$$

*If the Lipschitz constants are small enough: $2L_\pi(\gamma \sigma R + (1 - \gamma)k) \leq (1 - \gamma)^2$ and $2(L_P(\gamma \sigma R + (1 - \gamma)k) + L_r(1 - \gamma)\sqrt{d_2} \leq (1 + \rho - 2\rho^k)(1 - \gamma)\sqrt{d_2}$, then*

$$\epsilon_q + \epsilon_\mu \leq \frac{2}{1 - \min\{\gamma, \rho\}} \left(\epsilon_\phi + k\epsilon_\psi\right),$$

*where $\epsilon_\phi := \|q_* - \Pi_\phi q_*\|_\infty$ and $\epsilon_\psi := \|\mathcal{P}_* - \Pi_\psi \mathcal{P}_*\|_{\mathrm{TV}}$ are* inherent *approximation error induced by the projection onto the linear spans of basis $\phi$ and $\psi$, which is independent of the algorithm. Moreover, if $\|\mathcal{P}_* - \mathcal{P}_*^\infty\|_{\mathrm{TV}} = \rho$ (e.g., $\mathcal{P}_*$ induces a reversible Markov chain), $k = 1$.*

The proof of Theorem 2 is given in Appendix J. The approximation error of PA-LFA scales with the inherent approximation error determined by the chosen basis, which is consistent with the approximation error of LFA for value function (Tsitsiklis & Van Roy, 1997).

## 7 NUMERICAL EXPERIMENTS

To evaluate SemiSGD with PA-LFA, we conduct six experiments on three MFG examples: speed control (Appendix C.2), flocking (Appendix C.4), and network routing (Appendix C.6). When comparing with FPI-type methods, alongside vanilla online FPI, we equip FPI with entropy regularization (ER), fictitious play (FP), and mirror descent (MD). Please refer to Appendix C for the detailed setup, full results and analysis, and additional experiments. Here, we present some highlights of the results.

Figure 2a reports MSE curves for the **speed control** game. This example showcases the sample efficiency, stability, and accuracy of SemiSGD, particularly compared to vanilla FPI. The **flocking game** is highly sensitive to the reward (Appendix C.5). Large regularization may obscure the reward signal, leading to near-constant policies and population distributions, while other methods except SemiSGD cannot handle non-regularized policies well. Figure 2b reports exploitability curves for the flocking game, showing that only SemiSGD can effectively learn the game with near-greedy policies. The goal of the **network routing** game is to direct the traffic from the origin to the destination. Figure 2c reports the population distribution learned by SemiSGD on this highly non-smooth game. Figure 3 examines how the **number of inner loop iterations** ($K$) affect the convergence of FPI (Appendix C.3): it suggests that, given the same number of total samples, reducing $K$ monotonically improves the convergence. Notably, when $K = 1$, online FPI is equivalent to SemiSGD, which removes the forward-backward structure. Figure 4 and Table 1 compare **PA-LFA with discretization** (Appendix C.7) on learned distribution and accuracy respectively, demonstrating that with a proper choice of measure basis, PA-LFA generalizes better and achieves higher accuracy.

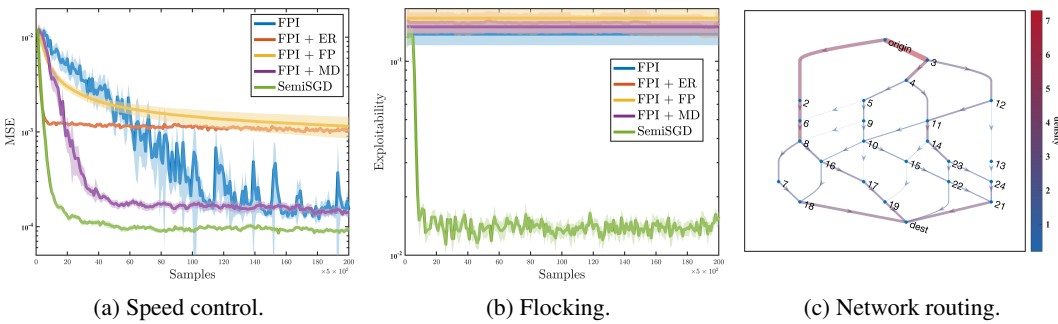

(a) Speed control.  (b) Flocking.  (c) Network routing.

Figure 2: Convergence performance of SemiSGD and FPI.

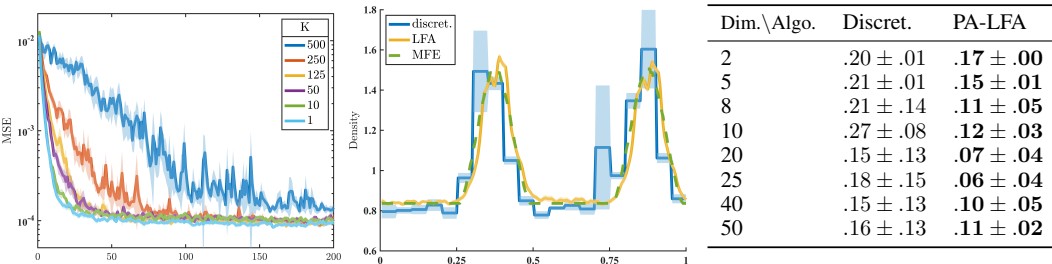

Figure 3: On # inner iterations. Figure 4: LFA & Discretization. Table 1: MSE of LFA & Discretization

## 8 CONCLUSION

This work proposes a simple SGD-type method for learning MFGs, eliminating the forward-backward structure in FPI-type methods and the need for complex stabilization mechanisms. Our approach leverages a novel perspective that unifies the value function and population measure as a single parameter, enabling a straightforward analysis that elegantly handle the coupling between policy and population while offering insights into the learning dynamics.

Building on this perspective, we develop the first population-aware linear function approximation for MFGs on large or continuous state-action spaces, with sample and operation efficiency guarantees, as well as approximation error characterization. Function approximation is just one example of how techniques from policy learning can be extended to population learning using this unified perspective, and we expect further advancements in this direction.

More broadly, our methodology is generalizable to other dynamical systems that can be parameterized by learnable parameters, including (finite) population games (Sandholm, 2010), evolutionary games (Hilbe et al., 2018), graphon games (Zhou et al., 2024), and policy optimization for MFGs (Zeng et al., 2025).

### ACKNOWLEDGMENTS

This work is sponsored by NSF under CAREER award number CMMI-1943998.

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

# Appendix

## Table of Contents

## A  EXTENDED LITERATURE REVIEW

Table 2: Comparison of learning methods for MFGs. In algorithm structure, 1L and $x$L with $x > 1$ denote single- and multi-loop algorithms, 1TS and $x$TS with $x > 1$ denote uni- and multi-timescale schemes, respectively. Additional mechanisms contain fictitious play (FP), mirror descent (MD), and entropy regularization (ER). MFG structures contain contractivity, monotonicity, and the herding structure. In learning dynamics assumptions, (C) denotes a blanket contractivity assumption, (R) denotes regularization, (L) denotes small Lipschitz constants condition, (M) denotes monotonicity, and (H) denotes the herding condition.

| Reference | Algorithm implementations | | | | | Theoretical properties | | |
|---|---|---|---|---|---|---|---|---|
| | Algo. struct. | Add. mech. | Oracle-free | Function approx. | | MFG struct. | Dyna. assmp. | Sample complex. |
| | | | | Population | Policy | | | |
| Guo et al. (2019) | FB | - | - | - | - | contract. | (C) | - |
| Elie et al. (2020) | FB | FP | - | - | - | mono. | (M) | - |
| Perrin et al. (2020) | FB | FP | - | - | - | mono. | (M) | - |
| Xie et al. (2021) | FB | MD | - | - | - | contract. | (C) | - |
| Cui & Koeppl (2021) | FB | ER | - | - | - | contract. | (R) | - |
| Laurière et al. (2022b) | FB | FP&MD | - | - | - | - | - | - |
| Mao et al. (2022) | 1L&2TS | - | - | - | ✓ | contract. | (C) | $\widetilde{O}(\epsilon^{-5})$ |
| Angiuli et al. (2022; 2023) | 1L&2TS | - | ✓ | - | - | contract. | (L&R) | - |
| Anahtarci et al. (2023) | FB | ER | - | - | - | contract. | (L&R) | - |
| Zaman et al. (2023) | 1L&2TS | - | ✓ | - | - | contract. | (C) | $O(\epsilon^{-4})$ |
| Yardim et al. (2023) | 3L | MD | ✓ | - | - | contract. | (R) | $O(\epsilon^{-2}\log^2 \epsilon^{-1})$ |
| Zhang et al. (2024a) | 2L | - | ✓ | - | - | contract. | (C) | $O(\epsilon^{-2}\log^2 \epsilon^{-1})$ |
| Huang et al. (2024) | FB | - | - | - | ✓ | - | - | $O(\epsilon^{-2}\log \epsilon^{-1})$ |
| Zeng et al. (2025) | 1L&1TS | - | ✓ | - | - | herding | (H) | $\widetilde{O}(\epsilon^{-4})$ |
| This paper | 1L&1TS | - | ✓ | ✓ | ✓ | contract. | (L) | $O(\epsilon^{-2}\log \epsilon^{-1})$ |

There is a growing body of literature on learning MFGs, with most relevant works summarized in Table 2. Readers can refer to Laurière et al. (2022a); Cui et al. (2022) for a comprehensive review.

We cast previous methods into the category that has a FPI-type **algorithm structure**, which typically use a forward-backward (FB) structure to calculate policy evaluation/optimization and induced population sequentially. To enable online learning and boost efficiency, several works have proposed multi-loop ($x$L) algorithms to update the policy and population simultaneously in an online fashion (Yardim et al., 2023; Zhang et al., 2024a). However, following an FPI-type structure, multi-loop algorithms *fix* the policy or population in the inner loops, which is observed to be more time-consuming and incur oscillations in the learned policies (Mao et al., 2022). In response, two-timescale (2TS) asynchronous update schemes have been proposed in Mao et al. (2022); Angiuli et al. (2022; 2023); Zaman et al. (2020), where their "single-loop" is highlighted as a key feature. However, in a two-timescale scheme, the policy or population updates *much slower* than the other to circumvent their strong coupling, essentially resembling a FPI-type structure. Moreover, due to the existence of a slower timescale, two-timescale methods typically have a larger sample complexity bound (Table 2). To accelerate the multi-timescale scheme, Zeng et al. (2025) adopt momentum-type buffers for parameter updates, resulting in an essentially uni-timescale step size with automatic stabilization similar to SemiSGD. In stark contrast, our method updates the policy and population simultaneously and *fully asynchronously*, resulting in a simple *single-loop* and *uni-timescale* method, completely removing the FPI-type structure and achieving an efficient $\widetilde{O}(\epsilon^{-2})$ sample complexity as an SGD method.

To address the instability issue of FPI, researchers have proposed various **additional mechanics** to stabilize the learning process, broadly categorized into three classes: 1) entropy regularization (ER) (Anahtarci et al., 2023; Cui & Koeppl, 2021); 2) fictitious play (FP) (Elie et al., 2020; Perrin et al., 2020; Cardaliaguet & Hadikhanloo, 2017); and 3) mirror descent (MD) (Perolat et al., 2021; Xie et al., 2021; Yardim et al., 2023). Notably, MD inherently includes regularization, with entropy regularization being a special case (Laurière et al., 2022b). In our algorithm implementation, we do not require any additional mechanics, demonstrating the automatic stabilization of SemiSGD.

Learning MFGs necessitates learning both the policy and the population. Some claimed online methods only learn the policy using online methods such as online reinforcement learning, but

assume an *oracle* for the population measure. As a result, the sample complexity results of these works (Mao et al., 2022; Huang et al., 2024) do not capture the population learning. We follow Zaman et al. (2023) to define an **oracle-free** method as one that does not assume access to an oracle which can provide the (estimated) population measure under a given policy. Among oracle-free methods, our work together with Angiuli et al. (2022); Zaman et al. (2023); Zeng et al. (2025) maintain a population estimate using online observations of a *single agent*. While Yardim et al. (2023) uses the empirical state distribution of $N$ agents to approximate the population measure.

To deal with large state and action spaces and obtain strong theoretical guarantees, (Mao et al., 2022; Huang et al., 2024) consider **function approximation** for the policy learning in MFGs. Our work is the first to apply function approximation to the population learning, which we argue is equally important as the policy learning for learning MFGs.

Turning to the theoretical properties of learning methods, two classes of **MFG structure** conditions are commonly considered that ensure convergence: the *monotonicity* condition imposes a structure on the reward function ensuring that the exploitability is a Lyapunov function (Elie et al., 2020; Perrin et al., 2021), and the *contractivity* condition requires that the algorithm gives a contraction mapping. Most recently, Zeng et al. (2025) propose a new class of MFGs called the *herding* class, which allows for multiple equilibria. There are many combinations of **dynamics assumptions** to ensure the contractivity of the algorithm. The most straightforward one is to assume a *blanket contractivity* condition, which for FPI-type methods is equivalent to require the composition of the forward-backward process to be a contractive mapping. Assuming that the mapping from the policy to the induced population is contractive, some works impose regularization on the policy to ensure that the mapping from the population measure to the policy is contractive, hence making their composition contractive (Cui & Koeppl, 2021). To further reduce the granularity of the assumptions, Yardim et al. (2023); Angiuli et al. (2023); Anahtarci et al. (2023) inspect the environment elements. Specifically, they assume the reward function and transition kernel are Lipschitz continuous with sufficiently small Lipschitz constants, and impose strong regularization, to ensure the contractivity of the FPI-type algorithm. Our assumptions closely resemble the last set of assumptions.

## B    More discussions on motivations

### B.1    Definitions of MFE and policy operators

We now extend the discussion in Section 2 on our definition of MFE.

**Standard definition of MFE using Bellman optimality equation.** Our definition of MFE follows the standard definition of defining the MFE as the fixed point of the Bellman operator and transition operator (Laurière et al., 2022b; Cui & Koeppl, 2021; Zaman et al., 2023; Anahtarci et al., 2023). These two fixed-point equations are also known as the "best response" condition and "consistency" condition in the MFG literature. The standard definition of the best response condition is defined using the Bellman optimality equation; see e.g., Angiuli et al. (2023). Our Definition 1 recovers the standard definition with an `argmax` policy operator:

$$\Gamma_\pi^{(\mathrm{argmax})}(Q)[s] = \mathbb{1}\left\{a = \mathrm{argmax}_{a'} Q(s, a')\right\}.$$

Specifically, with an `argmax` policy operator, (Bellman) becomes

$$\mathcal{T}_{(Q,M)}Q'(s,a) \coloneqq \mathbb{E}_{(Q,M)}\left[r(s,a,M) + \gamma \max_{a'} Q'(s',a')\right],$$
$$\text{with } a' = \underset{a'}{\mathrm{argmax}}\, Q(s',a'), s' \sim P(\cdot \mid s,a,M).$$

The fixed point of this operator satisfies

$$Q^*(s,a) = \mathbb{E}_{(Q^*,M)}\left[r(s,a,M) + \gamma \max_{a'} Q^*(s',a')\right],$$

which is exactly the Bellman optimality equation. Therefore, our definition of MFE covers the standard definition of MFE.

**Regularized MFE with regularized policy operators.** Our definition of MFE is more general as it accommodates general policy operators. For example, with a `softmax` policy operator,

$$\Gamma_\pi^{(\text{softmax})}(Q)[s] = \frac{L_\pi \exp(Q(s,a))}{\sum_{a'} L_\pi \exp(Q(s,a'))},$$

Definition 1 exactly recovers the definition of the Boltzmann MFE (Cui & Koeppl, 2021; Zaman et al., 2023). Notably, the above `softmax` policy operator satisfies Assumption 2 with Lipschitz constant $L_\pi$. With a general regularized policy operator, e.g., a mirror descent policy operator with regularizer $h$ (Perolat et al., 2021; Yardim et al., 2023),

$$\Gamma_\pi^{(\text{MD})}(Q)[s] = \underset{p \in \Delta\mathcal{A}}{\text{argmax}} \left\{ \langle p, Q(s,\cdot) \rangle - h(p) \right\},$$

Definition 1 corresponds to the regularized MFE, which is widely studied in the MFG literature (Mao et al., 2022; Anahtarci et al., 2023). Again, the mirror descent policy operator satisfies Assumption 2 when the regularizer $h$ is strongly convex (Yardim et al., 2023). Regularization is a common technique to stabilize the learning process and ensure the convergence of the FPI-type algorithm (Laurière et al., 2022a).

## B.2 MOTIVATIONS FOR PA-LFA

In the main text, we highlight that PA-LFA is a natural extension if we consider the value function and population measure as a unified parameter controlling the dynamics. On the other hand, there are other direct motivations for developing a function approximation method for the population learning in MFGs.

**Higher accuracy.** As illustrated in Example 2, given a continuous state space, discretizing the space gives a special linear function approximation that uses Dirac distributions as basis measures, denoted as $\delta$. Suggested by our analysis in Section 6, different measure bases lead to different representation spaces of the population measure, which in turn affects the approximation accuracy of applying LFA (Theorem 2). Therefore, if we choose a proper measure basis $\psi$ such that

$$\text{dist}(\mu_*, \text{span}(\psi)) < \text{dist}(\mu_*, \text{span}(\delta)),$$

or equivalently, $\epsilon_\psi < \epsilon_\delta$ in Theorem 2, then LFA with basis $\psi$ will have a higher accuracy than simple grid discretization, as the former has a representation space closer to the true equilibrium. This is validated in our experiments; see Figures 12 and 13 and Table 1.

**Incorporating prior knowledge.** Continuing our comparison between LFA and grid discretization, the latter is not appropriate for continuous or *dependent* distributions. For example, in our experiment on the ring road, the population measure is a continuous distribution, meaning that a high density in one state will lead to a high density in the neighboring states. However, discretization does not capture this dependency. Continuous population distributions are common in real-world applications, such as traffic flow (Chen et al., 2023a), flocking (Perrin et al., 2021), and crowd dynamics (Lachapelle & Wolfram, 2011; Burger et al., 2013). Furthermore, real-world population distributions can exhibit complex dependency structures, such as distributions on transportation networks (Zhang et al., 2024a). If we have prior knowledge on the dependency structure, we can choose a proper measure basis to capture this dependency, leading to a more efficient population learning process. Figure 12 also provides a clear illustration on this point.

**Sample efficiency.** As provided in Section 5, the sample complexity of SemiSGD scales with $H^2$, where $H$ can be regarded as the problem scale. In Appendix K, we show that for a finite MFG the problem scale is

$$H = O\left( \frac{\sqrt{|\mathcal{S}||\mathcal{A}|}R}{1-\gamma} \right),$$

which scales with the state-action space size. By choosing a low dimensional LFA ($d_1 d_2 \ll |\mathcal{S}||\mathcal{A}|$), we can achieve a much lower sample complexity.

**Operational efficiency.** Remark 1 analyzes the operation complexity of each update in SemiSGD with PA-LFA. Specifically, when $d_2 \approx |\mathcal{S}|$, the operation complexity of each update in SemiSGD with or without PA-LFA is comparable. When $d_2 \ll |\mathcal{S}|$, PA-LFA gives a significant reduction in operation complexity of each update. Note that the total operation complexity of the algorithm = sample (iteration) complexity $\times$ operation complexity per iteration. Thus, combining the sample efficiency gain from PA-LFA, an even larger operational efficiency gain is achieved.

A caveat is that if the environment simulator, especially the reward function and transition kernel, takes the full population measure as input, the operational efficiency gain in each update from PA-LFA may be offset. However, for many real-world applications, the simulator may not need the full population measure. For instance, for MFGs with sparse or local interaction, meaning that an agent's reward and transition only depend on the population density on a few states, the simulator only needs to know the population density on these states. Linear MFGs are another example. Recall the definition of linear MFGs (Definition 2); the transition simulator is a mapping:

$$ P : \mathcal{D}(\mathcal{S}) \ni M \mapsto \Omega_M \mapsto \langle \phi, \Omega_M \psi \rangle . $$

By Proposition 1, we know that any induced population measure of a linear MFG is within $\mathrm{span}(\psi)$. Additionally, the population measure estimate maintained by our method is also within $\mathrm{span}(\psi)$. Thus, the transition simulator can utilize this low-dimensional structure:

$$ P : \mathrm{span}(\psi) \ni M = \langle \psi, \eta \rangle \mapsto \Omega_M \mapsto \langle \phi, \Omega_M \psi \rangle $$
$$ \Rightarrow \quad \hat{P} : \mathbb{R}^{d_2} \ni \eta \mapsto \langle \psi, \eta \rangle \mapsto \Omega_{\langle \psi, \eta \rangle} \mapsto \langle \phi, \Omega_{\langle \psi, \eta \rangle} \psi \rangle . $$

That is, for any linear MFG, there exists a transition simulator $\hat{P}$ that takes low-dimensional vectors as input. This is also true for the reward function.

Nevertheless, since the sample efficiency gain always holds, PA-LFA always reduces the total operation complexity of the algorithm.

### B.3 COMPARISON OF SAMPLE COMPLEXITY

The worst-case sample complexity of SemiSGD analyzed in Section 5 is $O(\epsilon^{-2} \log \epsilon^{-1})$ (see also Corollary 3). This complexity aligns with that of SGD for Lipschitz functions ($\Theta(\epsilon^{-2})$ (Nemirovskij & Yudin, 1983)) MCMC methods ($O(\epsilon^{-2})$ (Łatuszyński et al., 2013)), and is better than the state-of-the-art complexity of TD methods (TD(0): $O(\epsilon^{-2} \log^2 \epsilon^{-1})$ (Bhandari et al., 2018), Q-learning: $O(\epsilon^{-2} \log^3 \epsilon^{-1})$ (Li et al., 2024), and SARSA: $O(\epsilon^{-2} \log^3 \epsilon^{-1})$ (Zou et al., 2019)). All these bounds are tight up to logarithmic factors. The consistency of these complexities is not surprising as they all belong to the class of stochastic approximation methods for a single parameter. In contrast, existing methods that treat the value function and population measure as two parameters and update them using a two-timescale scheme have a much worse sample complexity (Mao et al., 2022; Zaman et al., 2023).

Under the contractivity assumption, oracle-free methods Yardim et al. (2023, Theorem 4.3) and Zhang et al. (2024a, Theorem 1) have a comparable sample complexity $O(\epsilon^{-2} \log^2 \epsilon^{-1})$ to SemiSGD. Besides that their complexity is worse than ours up to a logarithmic factor, the source of the logarithmic dependence is also different. Both Yardim et al. (2023); Zhang et al. (2024a) use multi-loop FPI-type methods, with each outer loop being a contractive mapping. Due to this contraction, both methods need only $O(\log \epsilon^{-1})$ outer loops, with each loop having a sample complexity of $O(\epsilon^{-2} \log \epsilon^{-1})$. On the other hand, SemiSGD, being a single-loop method, only needs the same number of samples as that of one outer loops in Yardim et al. (2023); Zhang et al. (2024a). The logarithmic dependence in SemiSGD's sample complexity comes from the non-stationarity introduced by the time-varying population measure estimate and behavior policy. In summary, Yardim et al. (2023); Zhang et al. (2024a) need multiple runs of the policy evaluation/optimization and induced population calculation procedure, with each run having the same sample complexity as SemiSGD, which only needs one run of the stochastic approximation method.

## C  ADDITIONAL EXPERIMENTS

### C.1  GENERAL SETUP AND REMARK

All numerical results are averaged over 10 independent runs with random initialization. 95% confidence regions are reported. The following parameters are shared for all methods in this set experiments unless otherwise specified: total number of samples $T = 10^5$, with the number of inner loop iterations being $K = 500$ for FPI-type methods, constant step-size $\alpha = 10^{-3}$, policy operator $\Gamma_\pi = \text{softmax}$ with a large inverse temperature (near-greedy).

**Reference methods.**  We compare SemiSGD with vanilla online FPI and its variants with stabilization mechanisms entropy regularization (ER), fictitious play (FP), and online mirror descent (MD). For all methods, we use on-policy TD learning to learn the best response value function, and MCMC to calculate the induced population measure. Specifically, for online FPI, we follow Zhang et al. (2024a) to repeat the forward population calculation and backward policy evaluation alternatively; for FP, we follow Perrin et al. (2020); Laurière et al. (2022b) to mix the historical population measures with the current one after each induced population calculation; for MD, we follow Perolat et al. (2021); Laurière et al. (2022b) to conduct an incremental Q-value function update after each policy evaluation; for ER, we follow Cui & Koeppl (2021) to use the softmax policy operator with a large temperature (also called Boltzmann policies) for the policy update. We summarize the reference methods in Algorithm 2.

---

**Algorithm 2:** Reference methods

1  **input**: Initial value function $q_0$, population measure $\mu_0$, and policy $\pi_0$.
2  **for** $t = 0, 1, \ldots T/K$ **do**
3    Forward population calculation: $\mu_{t+1}$ is induced by policy $\pi_t$.
4    **if** FP **then**
5      $\mu_{t+1} \leftarrow \mu^{(\text{hist})} + \alpha_t \mu_{t+1}$.
6      $\mu^{(\text{hist})} \leftarrow \mu_{t+1}$.
7    Backward policy evaluation: $q_{t+1}$ evaluates $\pi_t$ with $\mu_{t+1}$.
8    **if** MD **then**
9      $q_{t+1} \leftarrow q^{(\text{hist})} + \alpha_t q_{t+1}$.
10     $q^{(\text{hist})} \leftarrow q_{t+1}$.
11   **if** ER **then**
12     Policy update: $\pi_{t+1} = \Gamma_\pi^{(\text{reg.})}(q_{t+1})$.
13   **else**
14     Policy update: $\pi_{t+1} = \Gamma_\pi(q_{t+1})$.
15 **return** $(\pi_{T/K}, \mu_{T/K})$.

---

**Regularization.**  To minimize regularization for all methods except ER, we use the softmax policy operator with a large inverse temperature such that

$$\log_{10}\left( L_\pi \left( \max_{a \in \mathcal{A}} Q(s,a) - \min_{a \in \mathcal{A}} Q(s,a) \right) \right) \approx 2, \quad \forall s \in \mathcal{S} \tag{7}$$

where we use $L_\pi$ to represent the inverse temperature, as softmax is Lipschitz continuous with the inverse temperature being a Lipschitz constant (Gao & Pavel, 2017). This setup ensures near-greedy policies with negligible regularization. One can verify that in our experiments, the Q-value differences between actions are typically small, leading to a large $L_\pi$. Consequently, the theoretical regularization implied by Assumption 4 is not enforced in our experiments. For ER, we use a small inverse temperature of $L_\pi^{(\text{ER})} = L_\pi / 10^5$ to implement the entropy regularization. Regularization can significantly impact learning dynamics; see Appendix C.5 for more experiments on regularization.

We focus on two convergence performance metrics: mean squared error (MSE) of the population distribution and exploitability of the policy.

**Reference MFE and mean squared error.** We compare our results with the reference MFE distribution $M_{\text{ref}}$, which is calculated by *model-based FPI with FP*. The model-based FPI consists of a value iteration (Sutton & Barto, 2018) for the best responses and direct computation of the induced population measure using the transition operator. Then, the (population measure) mean squared error (MSE) is calculated as

$$\text{MSE}(M) := \|M - M_{\text{ref}}\|_2^2 = \sum_{s \in \mathcal{S}} \left(M(s) - M_{\text{ref}}(s)\right)^2.$$

Note that this is equivalent to the 1-Wasserstein distance between the two measures for finite state spaces.

**Exploitablity.** Perrin et al. (2020) defines the exploitability of a policy as follows:

$$\text{exploitability}(\pi) := \max_{\pi'} \mathbb{E}_{s \sim \mu_\pi} V(s; \pi', \mu_\pi) - \mathbb{E}_{s \sim \mu_\pi} V(s; \pi, \mu_\pi),$$

where $V(s; \pi, \mu)$ is the value function determined by policy $\pi$ and population distribution $\mu$. The exploitability results in our experiments are calculated using model-based value iteration. We remark that although exploitability is a useful metric to evaluate the performance of the learned policy, it is an operator determined by the underlying MDP, which may not be smooth and hence does not directly reflect the learning dynamics.

### C.2 SPEED CONTROL ON A RING ROAD

We consider a speed control game on a ring road, i.e., the unit circle $\mathbb{S}^1 \cong [0, 1)$. At location $s \in \mathbb{S}^1$, the representative vehicle selects a speed $a$, and then moves to the next location following transition $s' = s + a\Delta t \pmod 1$, where $\Delta t$ is the time interval between two consecutive decisions. Without loss of generality, we assume that the speed is bounded by 1, i.e., the speed space is also $[0, 1)$. Then we discretize both the location space and the speed space using a granularity of $\Delta s = \Delta a = 0.02$. Thus, both our discretized state (location) space and action (speed) space can be represented by $\mathcal{S} = \mathcal{A} = \{0, 0.02, \ldots, 0.98\} \cong [50]$. By the Courant-Friedrichs-Lewy condition, we choose the time interval to be $\Delta t = 0.02 \leq \Delta s / \max a$. The objective of a vehicle is to maintain some desired speed while avoiding collisions with other vehicles. Thus, it needs to reduce the speed in areas with high population density. A classic cost function for this goal is the Lighthill-Whitham-Richards function:

$$r^{(\text{LWR})}(s, a, \mu) = -\frac{1}{2}\left(\left(1 - \frac{\mu(s)}{\mu_{\text{jam}}}\right) - \frac{a}{a_{\text{max}}}\right)^2 \Delta s,$$

where $\mu_{\text{jam}}$ is the jam density, and $a_{\text{max}}$ is the maximum speed. However, in an infinite horizon game, this cost function induces a *trivial* MFE, where the equilibrium policy and population are both constant across the state space. Therefore, we introduce a stimulus term $b$ that varies across different locations:

$$r(s, a, \mu) = -\frac{1}{2}\left(b(s) + \frac{1}{2}\left(1 - \frac{\mu(s)}{\mu_{\text{jam}}}\right) - \frac{a}{a_{\text{max}}}\right)^2 \Delta s,$$

where the factor of one-half before the population distribution term is included to account for the presence of the new stimulus term. This new cost function makes the MFE more complex and corresponds to real-world situations where vehicles may have distinct desired speeds at different locations due to environmental variations. Specifically, we choose the stimulus term as $b(s) = 0.2(\sin(4\pi s) + 2)$, and set $\mu_{\text{jam}} = 3/S$ and $a_{\text{max}} = 1$. The discount factor is set as $\gamma = 1 - \Delta s = 0.98$. To ensure the condition in (7), we choose a large inverse temperature of $L_\pi = 10^9$ for all methods except ER, which uses $L_\pi^{(\text{ER})} = L_\pi/10^5$.

The convergence of model-based FPI with FP is shown in Figure 5, arguing the near-optimality of the reference MFE.

The convergence performance is reported in Figure 6. The MSE comparison (Figure 6a) demonstrates that SemiSGD 1) is more efficient, with a much faster convergence, 2) is more stable without any stabilization techniques, 3) and is more accurate, with a lower MSE. The efficiency gain of SemiSGD is partially due to its removal of the forward-backward structure typical in FPI-type methods; we will explore more on this in Appendix C.3. Another observation is that although FPI equipped with ER

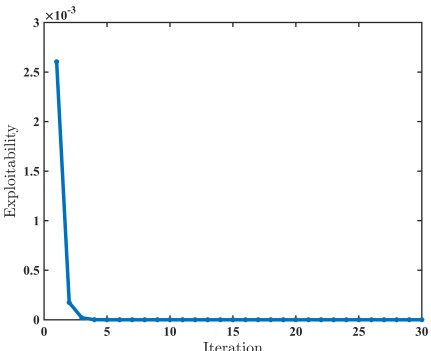

Figure 5: Exploitablity of model-based FPI with FP.

or FP stabilizes, its learned population distribution significantly deviates from the reference MFE, evidenced by Figures 6a and 6c. This illustrates the fact that the regularized MFE may be far from the true MFE. For the exploitability, SemiSGD seems to oscillate. However, exploitability is a function determined by the underlying MDP, which may not be smooth. Therefore, exploitability does not directly reflect the learning dynamics.

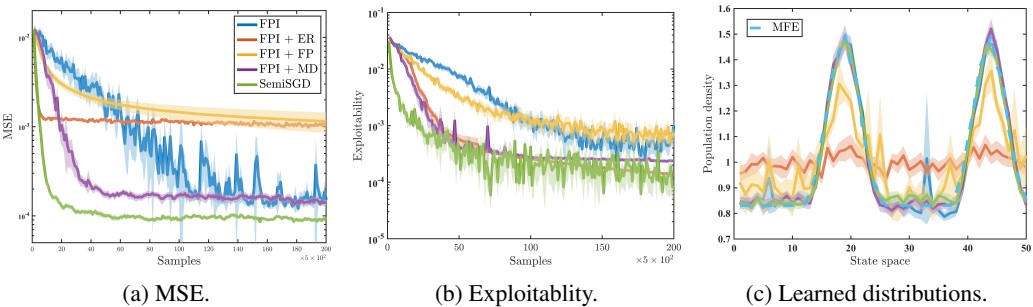

(a) MSE.  (b) Exploitablity.  (c) Learned distributions.

Figure 6: Convergence performance on speed control.

### C.3  NUMBER OF INNER LOOP ITERATIONS

When implementing FPI-type methods, we choose a relatively large number of inner loop iterations $K = 500$ to demonstrate the difference of a double-loop structure from a single-loop fully asynchronous structure. In this section, we explore the impact of this important hyperparameter. Notably, when $K = 1$, online FPI is equivalent to SemiSGD.

Fixing the number of total samples $T = 10^5$, we vary the number of inner loop iterations $K$ from 1 to 500 for vanilla FPI. The convergence performance is reported in Figure 7. Evident from both the MSE and exploitability plots, reducing the number of inner loop iterations leads to a *monotonic* improvement. This backs our theoretical analysis in Section 2.2 and suggests removing the forward-backward or multi-loop structure can significantly improve the convergence performance.

### C.4  FLOCKING GAME

We consider a flocking game on a one dimensional space $\mathcal{S} = [0, 1]$. We set the destination point as $s_{\text{det}} = 1$. To form an infinite-horizon game, we connect the destination point to the starting point $s_{\text{start}} = 0$. Similar to the speed control game Appendix C.2, at location $s \in \mathcal{S}$, the agent selects a speed $a \in [0, 1]$, and then moves to the next location following transition $s' = s + a\Delta t$ (mod 1). Then we discretize both the location space and the speed space using a granularity of $\Delta s = \Delta a = 0.02$ and choose the time interval to be $\Delta t = 0.02 \leq \Delta s / \max a$. The objective of an agent is to reach the destination as fast as possible while aligning its speed with its neighbors. A cost function for this goal is:

$$r(s, a, \mu) = -(a^2 + c(s_{\text{det}} - \text{neighbor}(\mu, s))^2)\Delta s,$$

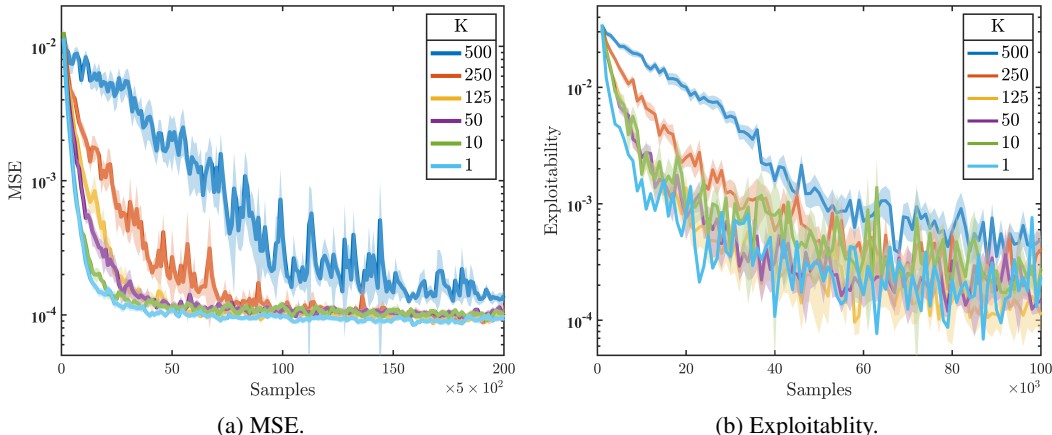

(a) MSE.

(b) Exploitablity.

Figure 7: Convergence performance with different numbers of inner loop iterations.

where $c$ is a positive constant; $\mathrm{neighbor}(\mu, s)$ calculates the average location of the neighbors of $s$:

$$\mathrm{neighbor}(\mu, s) = \frac{\int_{s-r}^{s+r} s' \mu(s') \mathrm{d}s'}{\int_{s-r}^{s+r} \mu(s') \mathrm{d}s'},$$

where $r$ is the radius of the neighborhood, and we pad the population measure with zero beyond the boundary. Specifically, we choose $c = 0.5$ and $r = 0.1$. The discount factor is set as $\gamma = 1 - \Delta s = 0.98$. To ensure the condition in (7), we choose a large inverse temperature of $L_\pi = 10^6$ for all methods except ER, which uses $L_\pi^{(\mathrm{ER})} = L_\pi/10^5$.

The convergence of model-based FPI with FP is shown in Figure 8, arguing the near-optimality of the reference MFE.

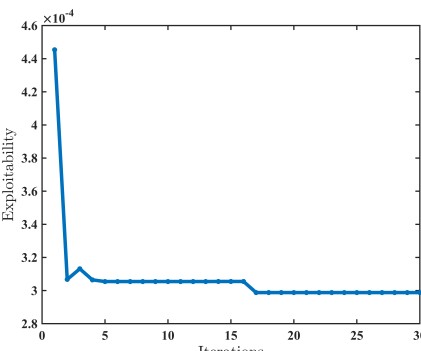

Figure 8: Exploitablity of model-based FPI with FP.

The convergence performance is reported in Figure 9. We first want to highlight that the MFE population distribution has a gathering behavior near the destination, and SemiSGD is the only method that captures this behavior (see Figure 9c). Judging from the reported metric, we conclude that other methods fails to learn the game. One possible explanation is that the flocking game is highly *sensitive* to the reward function, whose supremum norm is small ($R \leq \Delta s = 0.02$). As a result, large regularization obscures the reward signal, leading to a near-constant policy and population distribution. However, other methods except SemiSGD cannot handle near-greedy policies well, leading to a poor performance. To justify this explanation, we further explore the impact of regularization in Appendix C.5.

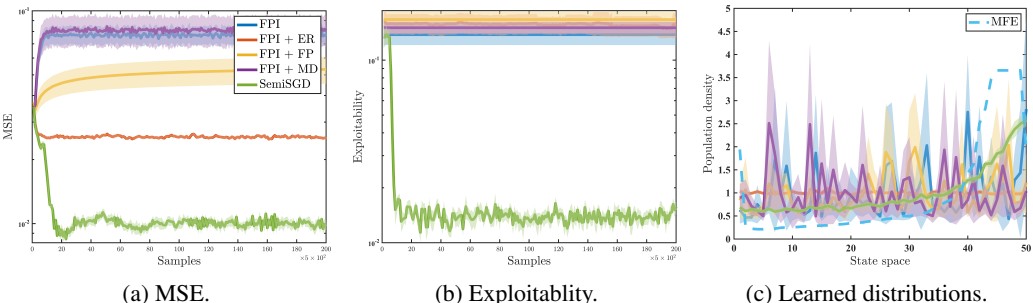

(a) MSE.      (b) Exploitablity.      (c) Learned distributions.

Figure 9: Convergence performance on flocking game.

## C.5 Regularization

As we can see from other experiments, regularization plays a crucial role in the learning process. In this section, we explore the impact of regularization on the convergence performance of learning the flocking game. We impose different levels of regularization on SemiSGD, and report the convergence performance in Figure 10. Recall that a small $L_\pi$ implies a large regularization. Consistent with Appendix C.4, large regularization ($L_\pi \leq 10^2$) obscures the reward signal, leading to a near-constant policy and population distribution. With a moderate regularization ($L_\pi = 10^4$), SemiSGD begins to capture the gathering behavior near the destination. And with negligible regularization ($L_\pi \geq 10^6$), SemiSGD achieves small MSE. However, the greediness may lead to a high exploitability, as shown in Figure 10a.

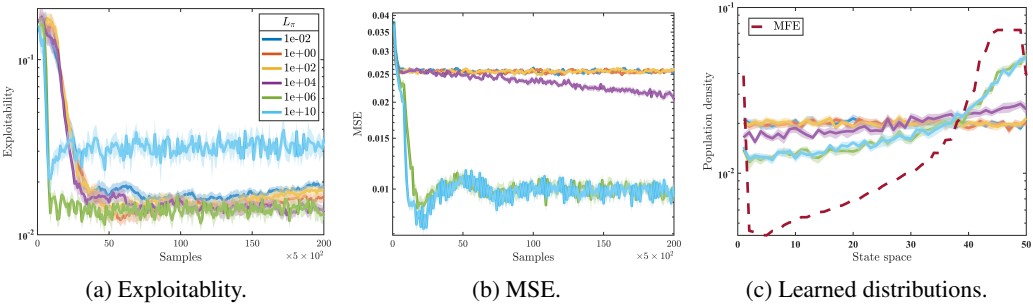

(a) Exploitablity.      (b) MSE.      (c) Learned distributions.

Figure 10: Convergence performance with different levels of regularization.

In summary, this experiment demonstrates the trade-off between regularization and accuracy. Regularization helps stabilize the learning process, but too much regularization may *smooth out* important information and drive the regularized MFE away from the true MFE.

## C.6 Network routing

We consider a routing game on the Sioux Falls network,[2] a graph with $24$ nodes and $74$ directed edges. We designate node $1$ as the starting point and node $20$ as the destination. To construct an infinite-horizon game, we add a *restart* edge $e_{75}$ from the destination back to the starting point. On each edge, a vehicle selects its next edge to travel to. We consider a deterministic environment, meaning that the vehicle will follow the chosen edge without any randomness. Therefore, both the state space and the action space can be represented by the edge set, i.e., $\mathcal{S} = \mathcal{A} = \{e_1, \ldots, e_{75}\} \cong [75]$, where $e_{75}$ is the restart edge. It is worth noting that a vehicle can only select from the outgoing edges of its current location as its next edge.

The objective of a vehicle is to reach the destination as fast as possible. Due to congestion, a vehicle spends a longer time on an edge with higher population distribution. Specifically, the cost (time) on a

---

[2]The topology of the network is available at `https://github.com/bstabler/TransportationNetworks`.

non-restart edge is $r^{(\text{cong.})}(s, a, \mu) = -c_1 \mu(s)^2 \mathbb{1}\{s \neq e_{75}\}$, where $c_1$ is a cost constant. To drive the vehicle to the destination, we impose a reward at the restart edge: $r^{(\text{term.})}(s, a, \mu) = c_2 \mathbb{1}\{s = e_{75}\}$. Together, we get the cost function:

$$r(s, a, \mu) = \underbrace{-c_1 \mu(s)^2 \mathbb{1}\{s \neq e_{75}\}}_{\text{congestion cost}} + \underbrace{c_2 \mathbb{1}\{s = e_{75}\}}_{\text{terminal reward}}.$$

We set $c_1 = 10^5$ and $c_2 = 10$. The other algorithmic parameters are chosen as follows: the discount factor $\gamma = 0.5$, the initial state is uniformly sampled, the initial value function is set as all-zero, the initial population is randomly generated.

Similarly, to satisfy (7), we choose a large inverse temperature of $L_\pi = 10^3$ for all methods except ER, which uses $L_\pi^{(\text{ER})} = L_\pi / 10^5$. The performance comparison is reported in Figure 11. We only plot the learned population distribution by SemiSGD in Figure 11c as all methods learn similar distributions. Notably, all methods except FPI with ER have an oscillatory behavior in the exploitability around the same value. This is due to the highly non-smooth nature of the underlying environment and the choice of near-greedy policies. In terms of MSE, only SemiSGD and FPI with ER achieve a low MSE, with SemiSGD achieving the lowest error and being the most stable.

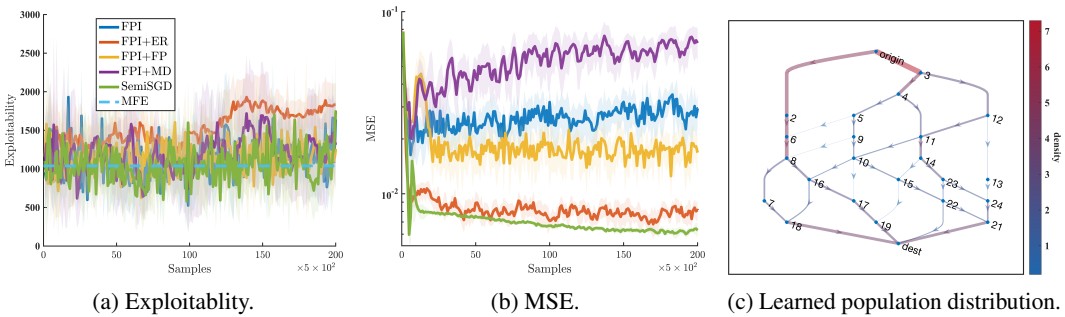

(a) Exploitablity.  (b) MSE.  (c) Learned population distribution.

Figure 11: Convergence performance on network routing.

### C.7 PA-LFA AND GRID DISCRETIZATION

To demonstrate the effectiveness of PA-LFA, we compare it with grid discretization on the speed control example (Appendix C.2). The reference MFE is calculated using a grid discretization with a granularity of $1/200$. We only apply LFA to the population measure estimate, as this is our main focus. The measure basis is chosen as

$$\psi_i(s) = c f_\mathcal{N}(0) - f_\mathcal{N}(\tan((s - s_i)\pi)),$$

where $f_\mathcal{N}$ is probability density function of the normal distribution with zero mean and variation $v$, and $s_i$ is the center of the basis function. Specifically, we set $c = 1.2$ and $v = d_2/2$, and evenly distribute $\{s_i\}_{i=1}^{d_2}$.

We run SemiSGD with $d_2$ states (grid discretization) or $d_2$ basis functions (PA-LFA) for $T = 10^4$ steps. Recall that PA-LFA with a $d_2$ dimensional feature space has comparable operation complexity to grid discretization with $d_2$ states. Other parameters are the same as the general setup. The final MSE (accuracy) comparison is reported in Table 1. The MSE curves and learned population distributions are shown in Figure 13 and Figure 12, respectively. As we can see, PA-LFA achieves a better accuracy than grid discretization for all $d_2 \leq 50$. This is because by choosing an appropriate measure basis, PA-LFA generalizes better than grid discretization, as the representation capacity of span $\psi$ is larger than the grid discretization and the representation space of $\psi$ is closer to the MFE population measure. This is more evident when $d_2$ is small. As illustrated in Figure 12, with just a few basis functions, PA-LFA can capture the general shape of the MFE population distribution much better than grid discretization.

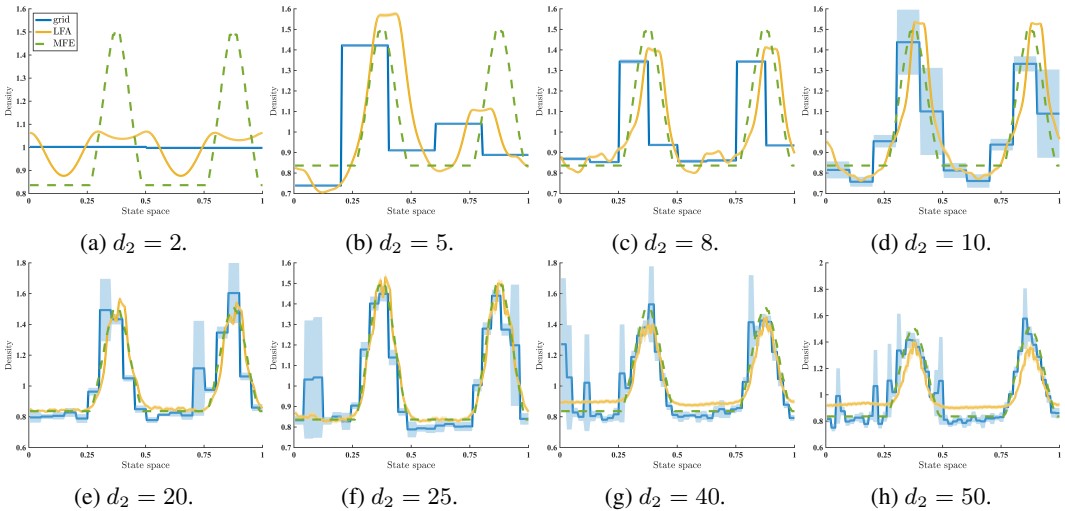

Figure 12: LFA versus discretization on learned distributions.

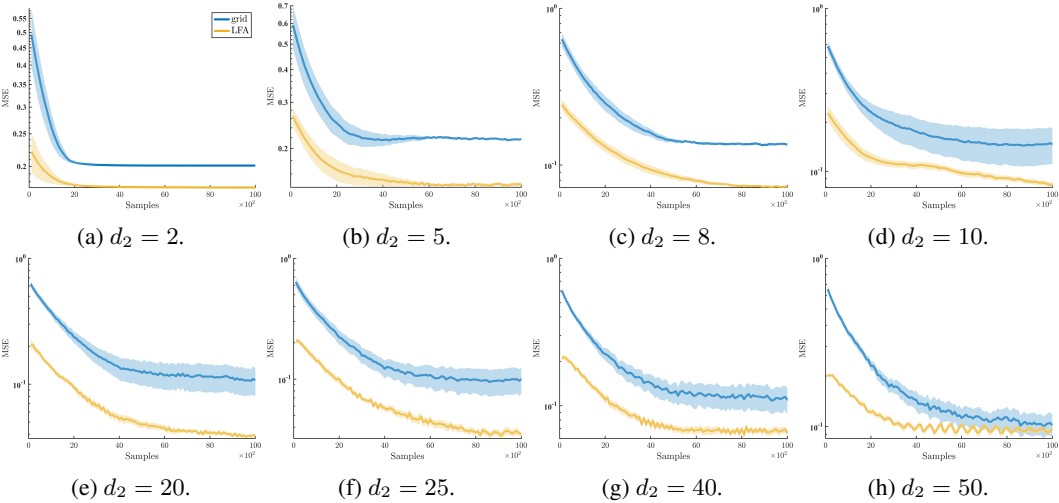

Figure 13: LFA versus discretization on MSE.

## D NOTATION

Table 3 provides a summary of the symbol notation used in this paper. We introduce some supplementary notation to assist analysis.

**Concatenation and direct sum.** We consider finite-dimensional Euclidean spaces as the parameter spaces. Thus, for any $x \in \mathbb{R}^{d_1}$ and $y \in \mathbb{R}^{d_2}$, we denote their concatenation as $(x; y) \in \mathbb{R}^{d_1 + d_2}$. We sometimes write it as the general direct sum between two vectors $x \oplus y \in \mathbb{R}^{d_1} \oplus \mathbb{R}^{d_2} \cong \mathbb{R}^{d_1 + d_2}$. For matrices and operators, we have $(A \oplus B)(x \oplus y) = Ax \oplus By$. This notation is especially useful for handling the unified parameter $\xi = \theta \oplus \eta$ (see, e.g., Lemma 7). Additionally, we use $\Xi$ to denote the unified parameter space $\Xi := \mathbb{R}^{d_1} \oplus \mathbb{R}^{d_2}$.

**Unprojected parameters.** We denote the unprojected parameters as $\breve{\xi}$, $\breve{\theta}$, and $\breve{\eta}$. That is,

$$\xi_{t+1} = \Pi(\breve{\xi}) = \Pi(\xi_t - \alpha_t \mathfrak{g}_t(\xi_t)).$$

For any parameter $\xi_*$ in the projected region, we have $\|\xi_* - \Pi(\breve{\xi})\| \le \|\xi_* - \breve{\xi}\|$.

Table 3: Notation.

| Notation | Definition |
| --- | --- |
| $\mathcal{D}(\mathcal{X}), \mathcal{M}(\mathcal{X})$ | Space of probability and signed measures on space $\mathcal{X}$ |
| $\Delta$ | Probability simplex |
| $\delta$ | Dirac delta measure |
| $a, \mathcal{A}$ | Action and action space |
| $s, \mathcal{S}$ | State and state space |
| $r, R$ | Reward function and its supremum norm bound |
| $\gamma$ | Discount factor |
| $P, \mathcal{P}$ | Transition kernel and operator |
| $\mathcal{T}$ | Bellman operator |
| $Q, q$ | Action-value function |
| $M, \mu$ | Population measure |
| $\pi, \Gamma_\pi$ | Policy and policy operator |
| $\phi$ | State-action feature map |
| $\psi$ | State measure basis |
| $F$ | Norm bound of $\psi$ |
| $d$ | Feature/measure space dimension |
| $\nu$ | Linear reward function parameter |
| $\omega, \Omega$ | Linear transition kernel parameter |
| $\eta$ | Population measure parameter |
| $\theta$ | Value function parameter |
| $\xi, \Xi$ | Concatenated parameter and its parameter space |
| $\Pi$ | Projection operator |
| $D$ | Value function projection bound |
| $H$ | Problem scale ($H = (1 + \gamma)D + R + 2F$) |
| $\mathfrak{g}$ | Semi-gradient |
| $G_\phi, \varphi$ | Temporal difference operator |
| $G$ | Gram matrix |
| $L$ | Lipschitz constant |
| $w$ | Contraction constant |
| $\alpha$ | Step-size |
| $m, \rho, \sigma, k$ | Ergodicity constants |
| $\tau$ | Backtracking period |

**Steady distributions.** We denote $\mu_\xi \in \mathcal{D}(\mathcal{S})$ as the steady state distribution induced by parameter $\xi = (\theta; \eta)$, i.e., by policy $\Gamma_\pi(\theta)$ and transition kernel and reward function determined by population measure parameter $\eta$. $\mu_\xi$ is the marginal distribution of the following two steady distributions:

$$\mu_\xi^\dagger(s, a) := \mu_\xi(s)\pi_\theta(a \mid s), \quad \mu_\xi^\ddagger(s, a, s', a') := \mu_\xi(s)\pi_\theta(a \mid s)P(s' \mid s, a, \eta)\pi_\theta(a' \mid s').$$

We write $\mathbb{E}_\xi$ as the expectation over the steady distribution induced by $\xi$; it should be clear from the context which steady distribution is used.

**Semi-gradients and temporal difference operators.** Here we review the definition of semi-gradients. With a slight abuse of notation, we use a single operator $\mathfrak{g}$ to return semi-gradients for both the action-value function and population measure. It should be clear from the argument of $\mathfrak{g}$ which parameter the semi-gradient is for. Specifically, with a sample tuple $O = (s, a, r, s', a')$, we have

$$\mathfrak{g}(\xi; O) = (\mathfrak{g}(\theta; O); \mathfrak{g}(\eta; O)) = (G_\phi(O)\theta - \varphi(O); G_\psi\eta - \psi(s')),$$

where $G_\psi$ is the gram matrix of the measure basis $\psi$, and $G_\phi$ and $\varphi$ are the temporal difference operators defined as

$$G_\phi(O) = \phi(s, a)(\phi(s, a) - \gamma\phi(s', a'))^T, \quad \varphi(O) = \phi(s, a)r.$$

Notably, $G_\psi$ is a constant matrix, and $G_\phi$ is a Gram-like matrix that depends on the sample tuple. Therefore, we sometimes drop the subscript $\phi$ in $G_\phi$ and use other subscripts to indicate its dependence

on the sample tuple. It should be clear that any $G$ with a subscript other than $\psi$ refers to $G_\phi$. When the sample tuple is an online observation at time step $t$, i.e., $O_t = (s_t, a_t, r_t, s_{t+1}, a_{t+1})$, we use shorthand

$$\mathfrak{g}_t(\cdot) = \mathfrak{g}(\cdot; O_t), \quad G_t = G_\phi(O_t), \quad \varphi_t = \varphi(O_t), \quad \psi_t = \psi(O_t).$$

Backtracking is an analysis technique introduced to tackle *rapidly changing Markov chains* (Zou et al., 2019; Zhang et al., 2023; 2024b). It considers a virtual stationary Markov chain by backtracking a period $\tau$, fixing the parameter $\xi_{t-\tau}$, and then sampling the Markovian observations with the fixed parameter. By the ergodicity of stationary Markov chains (Assumption 3), the virtual trajectory rapidly converges to the stationary distribution induced by $\xi_{t-\tau}$. We denote $\widetilde{O}_t$ as the virtual observation tuple on this virtual trajectory at time $t$. When we consider the semi-gradients on this virtual trajectory, we write out its dependence on $\widetilde{O}_t$ explicitly:

$$\mathfrak{g}_{t-\tau}(\cdot; \widetilde{O}_t), \quad G_{t-\tau}(\widetilde{O}_t), \quad \varphi_{t-\tau}(\widetilde{O}_t), \quad \psi_{t-\tau}(\widetilde{O}_t)$$

with the subscript indicating the backtracking period $\tau$.

*Mean-path* semi-gradients are the expectation of semi-gradients over a steady distribution induced by a parameter:

$$\bar{\mathfrak{g}}_\xi = \mathbb{E}_\xi \mathfrak{g}(\xi; O),$$

where the subscript $\xi$ indicates that the observation tuple $O$ follows the steady distribution induced by $\xi = (\theta; \eta)$. More explicitly, the states follows the steady distribution corresponding to transition kernel $P(\cdot \mid \cdot, \cdot, \eta)$, and the actions follow policy $\Gamma_\pi(\theta)$, and the rewards are generated by $r(\cdot, \cdot, \eta)$. Similarly, we have

$$\bar{G}_\xi = \mathbb{E}_\xi G_\phi(O), \quad \bar{\varphi}_\xi = \mathbb{E}_\xi \varphi(O), \quad \bar{\psi}_\xi = \mathbb{E}_\xi \psi(s').$$

When the parameter has a subscript $\xi_\circ$, we also use shorthand

$$\bar{\mathfrak{g}}_\circ = \bar{\mathfrak{g}}_{\xi_\circ}, \quad \bar{G}_\circ = \bar{G}_{\xi_\circ}, \quad \bar{\varphi}_\circ = \bar{\varphi}_{\xi_\circ}, \quad \bar{\psi}_\circ = \bar{\psi}_{\xi_\circ}.$$

For example, $\bar{\mathfrak{g}}_t = \bar{\mathfrak{g}}_{\xi_t}$ and $\bar{\mathfrak{g}}_* = \bar{\mathfrak{g}}_{\xi_*}$.

## E  PROOF OF PROPOSITION 1

*Proof.* By the Bellman equation, we have

$$Q_M^\pi(s, a) = r(s, a, M) + \gamma \int_{\mathcal{S} \times \mathcal{A}} Q_M^\pi(s', a') \pi(a' \mid s') P(s' \mid s, a, M) \mathrm{d}s' \mathrm{d}a'$$

$$= \langle \phi(s, a), \nu_M \rangle + \gamma \int_{\mathcal{S} \times \mathcal{A}} Q_M^\pi(s', a') \pi(a' \mid s') \langle \phi(s, a), \Omega_M \psi(s') \rangle \mathrm{d}s' \mathrm{d}a'$$

$$= \left\langle \phi(s, a), \underbrace{\nu_M + \gamma \int_{\mathcal{S} \times \mathcal{A}} Q_M^\pi(s', a') \pi(a' \mid s') \Omega_M \psi(s') \mathrm{d}s' \mathrm{d}a'}_{\theta} \right\rangle.$$

By the transition equation, we have

$$\mu_M^\pi(s') = \int_{\mathcal{S} \times \mathcal{A}} P(s' \mid s, a, M) \pi(a \mid s) \mu_M^\pi(s) \mathrm{d}s \mathrm{d}a$$

$$= \int_{\mathcal{S} \times \mathcal{A}} \langle \phi(s, a), \Omega_M \psi(s') \rangle \pi(a \mid s) \mu_M^\pi(s) \mathrm{d}s \mathrm{d}a$$

$$= \int_{\mathcal{S} \times \mathcal{A}} \left\langle \psi(s'), \Omega_M^T \phi(s, a) \right\rangle \pi(a \mid s) \mu_M^\pi(s) \mathrm{d}s \mathrm{d}a$$

$$= \left\langle \psi(s'), \underbrace{\int_{\mathcal{S} \times \mathcal{A}} \Omega_M^T \phi(s, a) \pi(a \mid s) \mu_M^\pi(s) \mathrm{d}s \mathrm{d}a}_{\eta} \right\rangle.$$

$\square$

## F    Projected MFE and stationary point

We call a value function-population measure pair a *stationary point* if the mean-path semi-gradient evaluated at this point is zero. Section 5 and Appendix H show that SemiSGD converges to a stationary point. Ideally, we want the stationary point to be an MFE (Definition 1). Proposition 1 and Appendix E show that for linear MFGs, images of $\mathcal{T}$ and $\mathcal{P}$ are within the linear spans of $\phi$ and $\psi$, indicating the linear structure of the MFE. However, this does not hold for general (non-linear) MFGs, hinting the discrepancy between the stationary point and the MFE for non-linear MFGs.

We define the *projected MFE* using the Bellman operator and transition operator composed with the projection operators.

**Definition 3** (Projected MFE). We say $\xi = (\theta; \eta)$ constitutes a projected MFE if

$$\langle \phi, \theta \rangle = \Pi_\phi \mathcal{T}_\xi \langle \phi, \theta \rangle, \quad \langle \psi, \eta \rangle = \Pi_\psi \mathcal{P}_\xi \langle \psi, \eta \rangle,$$

where $\Pi_\phi$ and $\Pi_\psi$ are orthogonal projection operators onto the linear spans of $\phi$ and $\psi$, respectively.

It should be noted that the projection operators are determined by the inner product structure of the function spaces. They can be explicitly expressed as

$$\Pi_\phi = \phi^T (\langle \phi, \phi^T \rangle_\circ)^{-1} \langle \phi, \cdot \rangle_\circ,$$

where $\phi$ is the function basis and $\langle \cdot, \cdot \rangle_\circ$ is the chosen inner product. Specifically, we choose the $L_2$ inner product on $\mathcal{M}(\mathcal{S})$, giving

$$\Pi_\psi = \psi^T G_\psi^{-1} \langle \psi, \cdot \rangle_{L_2}.$$

For the projection acting on $\mathcal{T}_\xi$, we choose the inner product induced by the steady distribution $\mu_\xi$, i.e., $\langle f, g \rangle_{\mu_\xi} = \int f(o)g(o)\mu_\xi^\ddagger(\mathrm{d}o)$. Then, we have

$$\Pi_\phi \mathcal{T}_\xi = \phi^T \widehat{G}_\xi^{-1} \langle \phi, \mathcal{T}_\xi \cdot \rangle_{\mu_\xi},$$

where $\widehat{G}_\xi$ is the Gram matrix of $\phi$ w.r.t. the inner product $\langle \cdot, \cdot \rangle_{\mu_\xi}$. Note that $\widehat{G}_\xi$ is different from the TD operator $G_\phi$ or $G_\xi$ defined in Appendix D, which is only Gram-like.

We are now ready to prove a generalized version of Proposition 2. Recall that for linear MFGs, the projected MFE is the MFE itself.

**Proposition 3** (Projected MFE as a stationary point). *$\xi$ is a projected MFE if and only if $\bar{\mathfrak{g}}_\xi(\xi) = 0$.*

*Proof.* For a parameter $\xi = (\theta; \eta)$, by the definition of mean-path semi-gradients, we have

$$\bar{\mathfrak{g}}_\xi(\theta) = \mathbb{E}_\xi \left[ \phi(s, a) \left( \phi^T(s, a)\theta - \gamma \phi^T(s', a')\theta - r(s, a, \eta) \right) \right],$$
$$\bar{\mathfrak{g}}_\xi(\eta) = \mathbb{E}_\xi \left[ G_\psi \eta - \psi(s') \right],$$

where the observation tuple $(s, a, s', a')$ follows the steady distribution induced by $\xi$. On the other hand, by the definition of the projection operators, we have

$$(\Pi_\phi \mathcal{T}_\xi - \mathrm{Id}) \langle \phi, \theta \rangle = \phi^T \widehat{G}_\xi^{-1} \langle \phi, \mathcal{T}_\xi \langle \phi, \theta \rangle \rangle_{\mu_\xi} - \phi^T \theta,$$

where $\mathrm{Id}$ is the identity operator. Suppose $\phi$ is linearly independent. Then, we get

$$(\Pi_\phi \mathcal{T}_\xi - \mathrm{Id}) \langle \phi, \theta \rangle = 0 \iff \widehat{G}_\xi^{-1} \langle \phi, \mathcal{T}_\xi \langle \phi, \theta \rangle \rangle_{\mu_\xi} = \theta$$
$$\iff \langle \phi, \mathcal{T}_\xi \langle \phi, \theta \rangle \rangle_{\mu_\xi} = \widehat{G}_\xi \theta$$
$$\iff \left\langle \phi, \mathbb{E}_\xi \left[ r(\cdot, \cdot, \eta) + \gamma \phi^T(s', a')\theta \right] \right\rangle_{\mu_\xi} - \mathbb{E}_\xi [\phi(s, a)\phi^T(s, a)]\theta = 0$$
$$\iff \mathbb{E}_\xi \left[ \phi(s, a) \left( r(s, a, \eta) + \gamma \phi^T(s', a')\theta \right) \right] - \mathbb{E}_\xi [\phi(s, a)\phi^T(s, a)]\theta = 0$$
$$\iff \mathbb{E}_\xi \left[ \phi(s, a) \left( \phi^T(s, a)\theta - \gamma \phi^T(s', a')\theta - r(s, a, \eta) \right) \right] = 0$$
$$\iff \bar{\mathfrak{g}}_\xi(\theta) = 0.$$

Similarly, for the projected transition operator, we have

$$
\begin{aligned}
(\Pi_\psi \mathcal{P}_\xi - \mathrm{Id}) \langle \psi, \eta \rangle = 0 &\iff \psi^T G_\psi^{-1} \langle \psi, \mathcal{P}_\xi \langle \psi, \eta \rangle \rangle_{L_2} - \psi^T \eta = 0 \\
&\iff \langle \psi, \mathcal{P}_\xi \langle \psi, \eta \rangle \rangle_{L_2} = G_\psi \eta \\
&\iff \int_{\mathcal{S}^2 \times \mathcal{A}} \psi(s') P(s' \mid s, a, \eta) \pi_\theta(a \mid s) \psi^T(s) \eta \, \mathrm{ds} \mathrm{da} \mathrm{ds}' - G_\psi \eta = 0 \\
&\iff \mathbb{E}_\xi[\psi(s')] - G_\phi \eta = 0 \\
&\iff \bar{\mathfrak{g}}_\xi(\eta) = 0.
\end{aligned}
$$

Therefore, by Definition 3, $\xi$ is a projected MFE if and only if $\mathfrak{g}_\xi(\xi) = 0$. $\qquad \square$

## G    PRELIMINARY LEMMAS

We present some preliminary lemmas that are used throughout the analysis.

**Lemma 2** (Norm relations). *For any vectors $x, y$, we have*

- $\|x \oplus y\|_1 = \|x\|_1 + \|y\|_1, \quad \|x \oplus y\|_1^2 \leq \|y\|_1^2 + \|x\|_1^2.$

- $\|x \oplus y\|_2 \leq \|x\|_2 + \|y\|_2, \quad \|x \oplus y\|_2^2 = \|y\|_2^2 + \|x\|_2^2.$

- $\|x\|_2 + \|y\|_2 \leq \sqrt{\max\{d_1, d_2\}} \|x \oplus y\|_2.$

- $\|x \oplus y\|_\infty = \max\{\|x\|_\infty, \|y\|_\infty\}, \quad \|x \oplus y\|_\infty^2 \leq \|y\|_\infty^2 + \|x\|_\infty^2.$

- $\|x\|_1 \|y\|_1 \leq \frac{1}{4} \|x \oplus y\|_1^2.$

- $\|x\|_2 \|y\|_2 \leq \frac{1}{2} \|x \oplus y\|_2^2.$

- $\|x\|_\infty \|y\|_\infty \leq \|x \oplus y\|_\infty^2.$

*Proof.* All relations are basic facts of norms and can be easily verified. $\qquad \square$

**Lemma 3** (Gradient bounds). *For any parameter $\xi = (\theta; \eta)$ and any observation tuple $O$, we have*

$$
\begin{aligned}
\|\mathfrak{g}(\theta; O)\| &\leq (1 + \gamma)\|\theta\| + R, \\
\|\mathfrak{g}(\eta; O)\| &\leq F\|\eta\| + F.
\end{aligned}
$$

*Moreover, suppose $\|\theta\| \leq D$ and $\|\eta\|_1 = 1$. Let $H := (1 + \gamma)D + R + 2F$. Then, we have*

$$
\|\mathfrak{g}(\xi; O)\| \leq H.
$$

*Proof.* By definition, we have

$$
\begin{aligned}
\|\mathfrak{g}(\theta; O)\| &= \|G_\phi(O)\theta - \varphi(O)\| \leq \|G_\phi(O)\| \|\theta\| + \|\varphi(O)\| \\
&\leq \|\phi(s, a)\| \|\phi(s, a) - \gamma\phi(s', a')\| \|\theta\| + \|\phi(s, a)\| |r(s, a, \eta)| \\
&\leq (1 + \gamma)\|\theta\| + R,
\end{aligned}
$$

where we use the fact that $\|\phi(s, a)\| \leq 1$. Similarly, we have

$$
\|\mathfrak{g}(\eta; O)\| \leq \|\mathfrak{g}(\eta; O)\|_1 = \|G_\psi \eta - \psi(s')\|_1 \leq \|G_\psi\|_{\mathrm{op}} \|\eta\|_1 + \|\psi(s')\|_1.
$$

The operator norm of $G_\psi$ satisfies

$$
\|G_\psi\|_{\mathrm{op}} = \sup_{\|\eta\|_1 = 1} \|G_\psi \eta\|_1 = \sup_{\|\eta\|_1 = 1} \left\| \int_{\mathcal{S}} \psi(s) \psi^T(s) \eta \, \mathrm{ds} \right\|_1 \leq \sup_{\|\eta\|_1 = 1} \int_{\mathcal{S}} \|\psi(s)\|_1 \langle \psi, \eta \rangle \, \mathrm{ds} \leq F,
$$

where the last inequality uses the norm bound of $\psi$ and the fact that $\langle \psi, \eta \rangle$ is a probability measure. Therefore, we get

$$
\|\mathfrak{g}(\eta; O)\| \leq 2F.
$$

Then, Lemma 2 indicates that $\|\mathfrak{g}(\xi; O)\| \leq H$ given that $\|\theta\| \leq D$. $\qquad \square$

To be more general, Assumptions 1 and 2 are stated in terms of the differences of population measures and value functions. For the ease of presentation, we will develop our results in terms of the parameters, and state the more general results in terms of the differences of population measures and value functions without proof. We need to first translate the Lipschitzness assumptions in terms of the parameters. In the rest of the paper, we refer to the following lemma when we need to use the Lipschitzness assumptions in terms of the parameters.

**Lemma 4** (Lipschitzness in parameters). *Assumption 1 and Assumption 2 imply that for any two parameters $\xi_1 = (\theta_1; \eta_1)$ and $\xi_2 = (\theta_2; \eta_2)$, we have*

$$\|P_{\eta_1} - P_{\eta_2}\|_{\mathrm{TV}} \le L_P \|\eta_1 - \eta_2\|, \quad \|r_{\eta_1} - r_{\eta_2}\|_\infty \le L_r \|\eta_1 - \eta_2\|,$$
$$\|\pi_{\theta_1}(\cdot \mid s) - \pi_{\theta_2}(\cdot \mid s)\|_{\mathrm{TV}} \le L_\pi \|\theta_1 - \theta_2\|.$$

*Proof.* The proof is straightforward noticing the fact that

$$\| \langle \psi, \eta \rangle \|_{\mathrm{TV}} = \int_{\mathcal{S}} |\eta^T \psi(s)| \mathrm{d}s = \int_{\mathcal{S}} \|\eta\|_1 \left| \frac{\eta^T}{\|\eta\|_1} \psi(s) \right| \mathrm{d}s \le \|\eta\|_1 \le \sqrt{d_2} \|\eta\|,$$

and

$$\| \langle \phi, \theta \rangle \|_\infty = \|\theta^T \phi(s, a)\|_\infty \le \|\|\theta\| \|\phi(s,a)\|\|_\infty \le \|\theta\|.$$

$\square$

**Lemma 5** (Lipschitz steady distributions). *For any two steady distributions $\mu^\ddagger_{\xi_1}$ and $\mu^\ddagger_{\xi_2}$ induced by parameters $\xi_1 = (\theta_1; \eta_1)$ and $\xi_2 = (\theta_2; \eta_2)$, we have*

$$\|\mu^\ddagger_{\xi_1} - \mu^\ddagger_{\xi_2}\|_{\mathrm{TV}} \le \sigma L \left( \|\theta_1 - \theta_2\| + \|\eta_1 - \eta_2\| \right),$$

*where $\sigma := 2 + \hat{n} + m\rho^{\hat{n}}/(1 - \rho)$, $\hat{n} := \lceil \log_\rho m^{-1} \rceil$, and $L = \max\{L_P, L_\pi\}$. Involved constants are defined in Assumptions 1 to 3. Since $\mu^\dagger_\xi$ and $\mu_\xi$ are marginal distributions of $\mu^\ddagger_\xi$, as a corollary, we have*

$$\|\mu^\dagger_{\xi_1} - \mu^\dagger_{\xi_2}\|_{\mathrm{TV}} \le \sigma L \left( \|\theta_1 - \theta_2\| + \|\eta_1 - \eta_2\| \right),$$
$$\|\mu_{\xi_1} - \mu_{\xi_2}\|_{\mathrm{TV}} \le \sigma L \left( \|\theta_1 - \theta_2\| + \|\eta_1 - \eta_2\| \right).$$

*Proof.* We first prove the last inequality in the lemma. By Mitrophanov (2005, Corollary 3.1), we have

$$\|\mu_{\xi_1} - \mu_{\xi_2}\|_{\mathrm{TV}} \le (\sigma - 2) \|P_{\xi_1} - P_{\xi_2}\|_{\mathrm{TV}},$$

where $P_\xi$ represents the transition kernel determined by policy $\Gamma_\pi(\theta)$ and population measure $\eta$, and

$$\|P_\xi\|_{\mathrm{TV}} := \sup_{\substack{q \in \mathcal{M}(\mathcal{S}) \\ \|q\|_{\mathrm{TV}}=1}} \left\| \int_{\mathcal{S}} q(s) P(\cdot \mid s, \theta, \eta) \mathrm{d}s \right\|_{\mathrm{TV}} = \sup_{\substack{q \in \mathcal{M}(\mathcal{S}) \\ \|q\|_{\mathrm{TV}}=1}} \left\| \int_{\mathcal{S} \times \mathcal{A}} q(s) \pi_\theta(a \mid s) P(\cdot \mid s, a, \eta) \mathrm{d}s \mathrm{d}a \right\|_{\mathrm{TV}}.$$

By the triangle inequality, we have

$$\|\mu_{\xi_1} - \mu_{\xi_2}\|_{\mathrm{TV}} \le \sigma \|P_{(\theta_1; \eta_1)} - P_{(\theta_1; \eta_2)}\|_{\mathrm{TV}} + \sigma \|P_{(\theta_1; \eta_2)} - P_{(\theta_2; \eta_2)}\|_{\mathrm{TV}}. \tag{8}$$

For the first term, by Assumption 1, we have

$$\left\| P_{(\theta_1; \eta_1)} - P_{(\theta_1; \eta_2)} \right\|_{\mathrm{TV}} = \sup_{\substack{q \in \mathcal{M}(\mathcal{S}) \\ \|q\|_{\mathrm{TV}}=1}} \left\| \iint_{\mathcal{S} \times \mathcal{A}} q(s) \pi_{\theta_1}(a \mid s) \left( P(\cdot \mid s, a, \eta_1) - P(\cdot \mid s, a, \eta_2) \right) \mathrm{d}s \mathrm{d}a \right\|_{\mathrm{TV}}$$

$$\le \sup_{\substack{q \in \mathcal{M}(\mathcal{S} \times \mathcal{A}) \\ \|q\|_{\mathrm{TV}}=1}} \left\| \iint_{\mathcal{S} \times \mathcal{A}} q(s, a) \left( P(\cdot \mid s, a, \eta_1) - P(\cdot \mid s, a, \eta_2) \right) \mathrm{d}s \mathrm{d}a \right\|_{\mathrm{TV}}$$

$$= \|P_{\eta_1} - P_{\eta_2}\|_{\mathrm{TV}}$$

$$\le L_P \|\eta_1 - \eta_2\|.$$

Similarly, for the second term in (8), by Assumption 2, we have

$$
\begin{aligned}
\left\| P_{(\theta_1;\eta_2)} - P_{(\theta_2;\eta_2)} \right\|_{\mathrm{TV}} &= \sup_{\substack{q \in \mathcal{D}(\mathcal{S}) \\ \|q\|_{\mathrm{TV}}=1}} \left\| \int_{\mathcal{S} \times \mathcal{A}} q(s) P(\cdot \,|\, s, a, \eta_2)(\pi_{\theta_1}(a \,|\, s) - \pi_{\theta_2}(a \,|\, s)) \mathrm{d}s \mathrm{d}a \right\|_{\mathrm{TV}} \\
&\leq \sup_{\substack{q \in \mathcal{D}(\mathcal{S}) \\ \|q\|_{\mathrm{TV}}=1}} \int_{\mathcal{S}^2 \times \mathcal{A}} q(s) P(s' \,|\, s, a, \eta_2) |\pi_{\theta_1}(a \,|\, s) - \pi_{\theta_2}(a \,|\, s)| \mathrm{d}a \mathrm{d}s \mathrm{d}s' \\
&= \sup_{\substack{q \in \mathcal{D}(\mathcal{S}) \\ \|q\|_{\mathrm{TV}}=1}} \int_{\mathcal{S}} q(s) \left\| \pi_{\theta_1}(\cdot \,|\, s) - \pi_{\theta_2}(\cdot \,|\, s) \right\|_{\mathrm{TV}} \mathrm{d}s \\
&\leq \sup_{\substack{q \in \mathcal{D}(\mathcal{S}) \\ \|q\|_{\mathrm{TV}}=1}} \int_{\mathcal{S}} q(s) \cdot L_\pi \|\theta_1 - \theta_2\| \mathrm{d}s \\
&= L_\pi \|\theta_1 - \theta_2\|.
\end{aligned}
$$

Let $L := \max\{L_P, L_\pi\}$. Plugging the above two inequalities into (8) gives

$$
\|\mu_{\xi_1} - \mu_{\xi_2}\|_{\mathrm{TV}} \leq \sigma L \left( \|\eta_1 - \eta_2\| + \|\theta_1 - \theta_2\| \right).
$$

Then, by the definition of $\mu_\xi^\dagger$, we have

$$
\begin{aligned}
\|\mu_{\xi_1}^\dagger - \mu_{\xi_2}^\dagger\|_{\mathrm{TV}} &= \int_{\mathcal{S} \times \mathcal{A}} |\mu_{\xi_1}(s) \pi_{\theta_1}(a \,|\, s) - \mu_{\xi_2}(s) \pi_{\theta_2}(a \,|\, s)| \, \mathrm{d}s \mathrm{d}a \\
&\leq \int_{\mathcal{S} \times \mathcal{A}} \left( |\mu_{\xi_1}(s) - \mu_{\xi_2}(s)| \, \pi_{\theta_1}(a \,|\, s) + \mu_{\xi_2}(s) \, |\pi_{\theta_1}(a \,|\, s) - \pi_{\theta_2}(a \,|\, s)| \right) \mathrm{d}s \mathrm{d}a \\
&= \|\mu_{\xi_1} - \mu_{\xi_2}\|_{\mathrm{TV}} + \int_{\mathcal{S}} \mu_{\xi_2}(s) \left\| \pi_{\theta_1}(\cdot \,|\, s) - \pi_{\theta_2}(\cdot \,|\, s) \right\|_{\mathrm{TV}} \mathrm{d}s \\
&\leq (\sigma - 2) L \left( \|\eta_1 - \eta_2\| + \|\theta_1 - \theta_2\| \right) + L_\pi \|\theta_1 - \theta_2\| \\
&\leq (\sigma - 1) L \left( \|\eta_1 - \eta_2\| + \|\theta_1 - \theta_2\| \right).
\end{aligned}
$$

Similarly, by the definition of $\mu_\xi^\ddagger$, we have

$$
\begin{aligned}
\|\mu_{\xi_1}^\ddagger - \mu_{\xi_2}^\ddagger\|_{\mathrm{TV}} &\leq \|\mu_{\xi_1}^\dagger - \mu_{\xi_2}^\dagger\|_{\mathrm{TV}} + L_P \|\eta_1 - \eta_2\| + L_\pi \|\theta_1 - \theta_2\| \\
&\leq \sigma L \left( \|\eta_1 - \eta_2\| + \|\theta_1 - \theta_2\| \right).
\end{aligned}
$$

$\square$

**Corollary 2.** *Similarly, for steady distributions induced by general value functions and population measures $\mu_1 := \mu_{(Q_1, M_1)}, \mu_2 := \mu_{(Q_2, M_2)}$, we have*

$$
\max \left\{ \|\mu_1 - \mu_2\|_{\mathrm{TV}}, \|\mu_1^\dagger - \mu_2^\dagger\|_{\mathrm{TV}}, \|\mu_1^\ddagger - \mu_2^\ddagger\|_{\mathrm{TV}} \right\} \leq \sigma \left( \frac{L_P}{\sqrt{d_2}} \|M_1 - M_2\| + L_\pi \|Q_1 - Q_2\|_\infty \right).
$$

**Lemma 6** (Lipschitz temporal difference operators). *For any two sets of mean-path temporal difference operators $\bar{G}_{\xi_1}, \bar{\varphi}_{\xi_1}, \bar{\psi}_{\xi_1}$ and $\bar{G}_{\xi_2}, \bar{\varphi}_{\xi_2}, \bar{\psi}_{\xi_2}$ determined by parameters $\xi_1 = (\theta_1; \eta_1)$ and $\xi_2 = (\theta_2; \eta_2)$, we have*

$$
\begin{aligned}
\|\bar{G}_{\xi_1} - \bar{G}_{\xi_2}\| &\leq \sigma L (1 + \gamma) \left( \|\theta_1 - \theta_2\| + \|\eta_1 - \eta_2\| \right), \\
\|\bar{\varphi}_{\xi_1} - \bar{\varphi}_{\xi_2}\| &\leq \sigma L R \left( \|\theta_1 - \theta_2\| + \|\eta_1 - \eta_2\| \right) + L_r \|\eta_1 - \eta_2\| \\
\|\bar{\psi}_{\xi_1} - \bar{\psi}_{\xi_2}\| &\leq \sigma L F \left( \|\theta_1 - \theta_2\| + \|\eta_1 - \eta_2\| \right),
\end{aligned}
$$

*where $\sigma$ and $L$ are defined in Lemma 5, and $L_r$ is defined in Assumption 1.*

*Proof.* By definition, we have

$$
\begin{aligned}
\|\bar{G}_{\xi_1} - \bar{G}_{\xi_2}\| &= \left\| \int_{\mathcal{S}^2 \times \mathcal{A}^2} \phi(s, a)(\phi(s, a) - \gamma \phi(s', a'))^T \left( \mu_{\xi_1}^\ddagger(s, a, s', a') - \mu_{\xi_2}^\ddagger(s, a, s', a') \right) \mathrm{d}s \mathrm{d}a \mathrm{d}s' \mathrm{d}a' \right\| \\
&\leq \int_{\mathcal{S}^2 \times \mathcal{A}^2} \left\| \phi(s, a)(\phi(s, a) - \gamma \phi(s', a'))^T \right\| \left( \mu_{\xi_1}^\ddagger(s, a, s', a') - \mu_{\xi_2}^\ddagger(s, a, s', a') \right) \mathrm{d}s \mathrm{d}a \mathrm{d}s' \mathrm{d}a'.
\end{aligned}
$$

Since $\|\phi(s,a)\| \leq 1$, we have $\left\|\phi(s,a)(\phi(s,a) - \gamma\phi(s',a'))^T\right\| \leq 1 + \gamma$. Then, by Lemma 5, we have

$$\|\bar{G}_{\xi_1} - \bar{G}_{\xi_2}\| \leq (1 + \gamma)\sigma L \left(\|\theta_1 - \theta_2\| + \|\eta_1 - \eta_2\|\right).$$

Similarly, by definition, we have

$$
\begin{aligned}
\|\bar{\varphi}_{\xi_1} - \bar{\varphi}_{\xi_2}\| &= \left\|\int_{\mathcal{S}\times\mathcal{A}} \phi(s,a)r(s,a,\eta_1)\mu_{\xi_1}^\dagger(s,a) - \phi(s,a)r(s,a,\eta_2)\mu_{\xi_2}^\dagger(s,a)\mathrm{d}s\mathrm{d}a\right\| \\
&\leq \int_{\mathcal{S}\times\mathcal{A}} \left(r(s,a,\eta_1)\left|\mu_{\xi_1}^\dagger(s,a) - \mu_{\xi_2}^\dagger(s,a)\right| + \mu_{\xi_2}^\dagger(s,a)\left|r(s,a,\eta_1) - r(s,a,\eta_2)\right|\right)\mathrm{d}s\mathrm{d}a \\
&\leq R\|\mu_{\xi_1}^\dagger - \mu_{\xi_2}^\dagger\|_{\mathrm{TV}} + L_r\|\eta_1 - \eta_2\| \\
&\leq \sigma LR\left(\|\theta_1 - \theta_2\| + \|\eta_1 - \eta_2\|\right) + L_r\|\eta_1 - \eta_2\|,
\end{aligned}
$$

and

$$\|\bar{\psi}_{\xi_1} - \bar{\psi}_{\xi_2}\| \leq F\|\mu_{\xi_1} - \mu_{\xi_2}\|_{\mathrm{TV}} \leq \sigma LF\left(\|\theta_1 - \theta_2\| + \|\eta_1 - \eta_2\|\right). \qquad \square$$

**Lemma 7** (Lipschitz semi-gradient). *Given a fixed semi-gradient operator $\mathfrak{g}$ (which can be a mean-path semi-gradient), for any parameters $\xi_1, \xi_2 \in \Xi$, we have*
$$\|\mathfrak{g}(\xi_1) - \mathfrak{g}(\xi_2)\| \leq H\|\xi_1 - \xi_2\|.$$
*Let $\xi_1 = (\theta_1; \eta_1)$ and $\xi_2 = (\theta_2; \eta_2)$. For any $\xi = (\theta; \eta)$ such that $\|\theta\| \leq G$, we have*
$$\|\bar{\mathfrak{g}}_{\xi_1}(\xi) - \bar{\mathfrak{g}}_{\xi_2}(\xi)\| \leq \sigma H \max\{L_P, L_\pi\}\left(\|\theta_1 - \theta_2\| + \|\eta_1 - \eta_2\|\right) + L_r\|\eta_1 - \eta_2\|.$$

*Proof.* We first show that the semi-gradient operator is Lipschitz in its argument. By definition, we have

$$\|\mathfrak{g}(\xi_1; O) - \mathfrak{g}(\xi_2; O)\| = \|\left(G_\phi(O) \oplus G_\psi\right)(\xi_1 - \xi_2)\| \leq \|G_\phi(O) \oplus G_\psi\|_{\mathrm{op}}\|\xi_1 - \xi_2\|.$$

We get the result by noticing that

$$\|G_\phi(O) \oplus G_\psi\|_{\mathrm{op}} \leq \max\{\|G_\phi(O)\|_{\mathrm{op}}, \|G_\psi\|_{\mathrm{op}}\} \leq \max\{1 + \gamma, F\} \leq H.$$

Next, we show that the mean-path semi-gradient operator is Lipschitz in the parameter. Another representation of the mean-path semi-gradient is

$$\bar{\mathfrak{g}}_{\xi_1} = \left(\bar{G}_{\xi_1}\theta - \bar{\varphi}_{\xi_1}; G_\psi\eta - \bar{\psi}_{\xi_1}\right).$$

Therefore, by Lemma 6, we have

$$
\begin{aligned}
\|\bar{\mathfrak{g}}_{\xi_1}(\xi) - \bar{\mathfrak{g}}_{\xi_2}(\xi)\| &\leq \left\|\bar{G}_{\xi_1}\theta - \bar{G}_{\xi_2}\theta - \bar{b}_{\xi_1} + \bar{b}_{\xi_2}\right\| + \left\|G_\psi\eta_1 - G_\psi\eta_2 - \bar{\psi}_{\xi_1} + \bar{\psi}_{\xi_2}\right\| \\
&\leq D\left\|\bar{G}_{\xi_1} - \bar{G}_{\xi_2}\right\| + \left\|\bar{b}_{\xi_1} - \bar{b}_{\xi_2}\right\| + \left\|\bar{\psi}_{\xi_1} - \bar{\psi}_{\xi_2}\right\| \\
&\leq \sigma L((1 + \gamma)D + R + F)\left(\|\theta_1 - \theta_2\| + \|\eta_1 - \eta_2\|\right) + L_r\|\eta_1 - \eta_2\|. \qquad \square
\end{aligned}
$$

## H  SAMPLE COMPLEXITY ANALYSIS

### H.1  KEY LEMMAS

In this section, we first decompose the mean square error into different terms, and then bound each term separately.

**Lemma 8** (Error decomposition). *Let $\xi_*$ be the optimal parameter. Recall that by Proposition 2, $\bar{\mathfrak{g}}_*(\xi_*) = 0$. We have*

$$
\begin{aligned}
\mathbb{E}\|\xi_{t+1} - \xi_*\|^2 &\leq \mathbb{E}\|\breve{\xi}_{t+1} - \xi_*\|^2 \\
&= \mathbb{E}\|\xi_t - \xi_*\|^2 - 2\alpha_t\mathbb{E}\left\langle\xi_t - \xi_*, \mathfrak{g}_t(\xi_t)\right\rangle + \alpha_t^2\|\mathfrak{g}_t(\xi_t)\|^2 \\
&= \mathbb{E}\|\xi_t - \xi_*\|^2 + \alpha_t^2\|\mathfrak{g}_t(\xi_t)\|^2 \\
&\quad - 2\alpha_t\mathbb{E}\left\langle\xi_t - \xi_*, \bar{\mathfrak{g}}_t(\xi_t) - \bar{\mathfrak{g}}_*(\xi_*)\right\rangle && \text{(descent)} \\
&\quad - 2\alpha_t\mathbb{E}\left\langle\xi_t - \xi_*, \bar{\mathfrak{g}}_{t-\tau}(\xi_t) - \bar{\mathfrak{g}}_t(\xi_t)\right\rangle && \text{(progress)} \\
&\quad - 2\alpha_t\mathbb{E}\left\langle\xi_t - \xi_*, \mathfrak{g}_{t-\tau}(\xi_t) - \bar{\mathfrak{g}}_{t-\tau}(\xi_t)\right\rangle && \text{(mix)} \\
&\quad - 2\alpha_t\mathbb{E}\left\langle\xi_t - \xi_*, \mathfrak{g}_t(\xi_t) - \mathfrak{g}_{t-\tau}(\xi_t)\right\rangle. && \text{(backtrack)}
\end{aligned}
$$

**Lemma 9** (Descent). *Let $\xi_* = (\theta_*; \eta_*)$ be a projected MFE. For any parameter $\xi = (\theta; \eta) \in B_D^{d_1} \times \Delta^{d_2}$, we denote $\Delta\theta = \theta - \theta_*$ and $\Delta\eta = \eta - \eta_*$. We have*

$$-\langle \eta - \eta_*, \bar{\mathfrak{g}}_\xi(\eta) - \bar{\mathfrak{g}}_{\xi_*}(\eta_*) \rangle \leq -\lambda_{\min}(G_\psi)\|\Delta\eta\|^2 + \sigma LF\|\Delta\eta\|\left(\|\Delta\theta\| + \|\Delta\eta\|\right),$$

$$-\langle \theta - \theta_*, \bar{\mathfrak{g}}_\xi(\theta) - \bar{\mathfrak{g}}_{\xi_*}(\theta_*) \rangle \leq -(1-\gamma)\lambda_{\min}(\hat{G}_{\xi_*})\|\Delta\theta\|^2 + \sigma LH\|\Delta\theta\|(\|\Delta\theta\| + \|\Delta\eta\|) + L_r\|\Delta\eta\|\|\Delta\theta\|.$$

*That is, neither $-\bar{\mathfrak{g}}_\xi(\eta)$ nor $-\bar{\mathfrak{g}}_\xi(\theta)$ is guaranteed to be a descent direction. Let $w := \frac{1}{2}\min\left\{\lambda_{\min}(G_\psi), (1-\gamma)\lambda_{\min}(\hat{G}_{\xi_*})\right\}$. Suppose $3\sigma LH + L_r \leq 2w$. Then, we have*

$$-\langle \xi - \xi_*, \bar{\mathfrak{g}}_\xi(\xi) - \bar{\mathfrak{g}}_{\xi_*}(\xi_*) \rangle \leq -w\|\xi - \xi_*\|^2.$$

*Proof.* We first focus on the first two inequalities. We denote $\Delta\eta := \eta - \eta_*$ and $\Delta\theta := \theta - \theta_*$. For the population measure parameter, by definition, we have

$$-\langle \Delta\eta, \bar{\mathfrak{g}}_\xi(\eta) - \bar{\mathfrak{g}}_{\xi_*}(\eta_*) \rangle = \langle \Delta\eta, -G_\psi \Delta\eta + \bar{\psi}_\xi - \bar{\psi}_{\xi_*} \rangle.$$

Note that $G_\psi$ is a positive definite Gram matrix if $\psi$ is linearly independent. Then, by Lemma 6, we have

$$-\langle \Delta\eta, \bar{\mathfrak{g}}_\xi(\eta) - \bar{\mathfrak{g}}_{\xi_*}(\eta_*) \rangle \leq -\langle \Delta\eta, G_\psi \Delta\eta \rangle + \|\bar{\psi}_\xi - \bar{\psi}_{\xi_*}\|\|\Delta\eta\|$$

$$\leq -\lambda_{\min}(G_\psi)\|\Delta\eta\|^2 + \sigma LF\|\Delta\eta\|(\|\Delta\eta\| + \|\Delta\theta\|). \tag{9}$$

For the value function parameter, we have

$$-\langle \Delta\theta, \bar{\mathfrak{g}}_\xi(\theta) - \bar{\mathfrak{g}}_{\xi_*}(\theta_*) \rangle = \langle \Delta\theta, \bar{G}_{\xi_*}\theta_* - \bar{\varphi}_{\xi_*} - \bar{G}_\xi \theta + \bar{\varphi}_\xi \rangle$$

$$= \langle \Delta\theta, -\bar{G}_{\xi_*}\Delta\theta + (\bar{G}_{\xi_*} - \bar{G}_\xi)\theta + \bar{\varphi}_\xi - \bar{\varphi}_{\xi_*} \rangle$$

$$\leq -\langle \Delta\theta, \bar{G}_{\xi_*}\Delta\theta \rangle + D\|\bar{G}_{\xi_*} - \bar{G}_\xi\|\|\Delta\theta\| + \|\bar{\varphi}_\xi - \bar{\varphi}_{\xi_*}\|\|\Delta\theta\|.$$

By Lemma 6, we have

$$-\langle \Delta\theta, \bar{\mathfrak{g}}_\xi(\theta) - \bar{\mathfrak{g}}_{\xi_*}(\theta_*) \rangle \leq -\langle \Delta\theta, \bar{G}_{\xi_*}\Delta\theta \rangle + \|\Delta\theta\|((1+\gamma)\sigma LD(\|\Delta\theta\| + \|\Delta\eta\|)$$

$$+ \sigma LR(\|\Delta\theta\| + \|\Delta\eta\|) + L_r\|\Delta\eta\|). \tag{10}$$

For a matrix $G$, we define $w_{\min}(G) := \min_{\|x\|=1}\langle x, Gx \rangle$. For $\bar{G}_{\xi_*}$, we have

$$w_{\min}(\bar{G}_{\xi_*}) = \min_{\|\theta\|=1}\mathbb{E}_{\xi_*}\left[\theta^T \phi(s,a)(\phi(s,a) - \gamma\phi(s',a'))^T\theta\right] =: \min_{\|\theta\|=1}\mathbb{E}_{\xi_*}\left[u^2 - \gamma u u'\right],$$

where $u := \theta^T \phi(s,a)$ and $u' := \theta^T \phi(s',a')$. We have

$$\mathbb{E}[uu'] \leq \frac{1}{2}\left(\mathbb{E}[u^2] + \mathbb{E}[u'^2]\right) = \frac{1}{2}\left(\mathbb{E}[u^2] + \mathbb{E}[u^2]\right) = \mathbb{E}[u^2].$$

Therefore,

$$w_{\min}(\bar{G}_{\xi_*}) \geq (1-\gamma)\min_{\|\theta\|=1}\mathbb{E}_{\xi_*}[u^2] = (1-\gamma)w_{\min}\left(\mathbb{E}_{\xi_*}[\phi\phi^T]\right) = (1-\gamma)w_{\min}\left(\hat{G}_{\xi_*}\right) = (1-\gamma)\lambda_{\min}\left(\hat{G}_{\xi_*}\right), \tag{11}$$

where $\hat{G}_{\xi_*}$ is the Gram matrix of the feature map $\phi$ under the steady distribution induced by $\xi_*$, and the last equality uses the property of normal matrices. Plugging the above derivation into (10) gives

$$-\langle \Delta\theta, \bar{\mathfrak{g}}_\xi(\theta) - \bar{\mathfrak{g}}_{\xi_*}(\theta_*) \rangle \leq -(1-\gamma)\lambda_{\min}(\hat{G}_{\xi_*})\|\Delta\theta\|^2 + \sigma L(H-2F)\|\Delta\theta\|(\|\Delta\theta\| + \|\Delta\eta\|) + L_r\|\Delta\eta\|\|\Delta\theta\|. \tag{12}$$

It is clear that (9) and (12) are controlled by both $\|\Delta\theta\|$ and $\|\Delta\eta\|$. When $\|\Delta\eta\| \gg \|\Delta\theta\|$, (12) suggests that $-\bar{\mathfrak{g}}_\xi(\theta)$ may not be a descent direction for $\|\Delta\theta\|$; when $\|\Delta\theta\| \gg \|\Delta\eta\|$, (9) suggests that $-\bar{\mathfrak{g}}_\xi(\eta)$ may not be a descent direction for $\|\Delta\eta\|$.

However, combining (9) and (12) gives

$$-\langle \Delta\xi, \bar{\mathfrak{g}}_\xi(\xi) - \bar{\mathfrak{g}}_{\xi_*}(\xi_*) \rangle \leq -(1-\gamma)\lambda_{\min}(\hat{G}_{\xi_*})\|\Delta\theta\|^2 - \lambda_{\min}(G_\psi)\|\Delta\eta\|^2 + \sigma LF\|\Delta\eta\|^2$$

$$+ \sigma L(H-2F)\|\Delta\theta\|^2 + (\sigma LF + \sigma L(H-2F) + L_r)\|\Delta\theta\|\|\Delta\eta\|.$$

By Lemma 2, we have

$$- \langle \Delta\xi, \bar{\mathfrak{g}}_\xi(\xi) - \bar{\mathfrak{g}}_{\xi_*}(\xi_*) \rangle$$

$$\leq - \min\{(1-\gamma)\lambda_{\min}(\hat{G}_{\xi_*}), \lambda_{\min}(G_\psi)\}\|\Delta\xi\|^2 + \sigma LH\|\Delta\xi\|^2 + \frac{1}{2}(\sigma LH + L_r)\|\Delta\xi\|^2$$

$$\leq \left( - \min\{(1-\gamma)\lambda_{\min}(\hat{G}_{\xi_*}), \lambda_{\min}(G_\psi)\} + \frac{1}{2}(3\sigma LH + L_r) \right) \|\Delta\xi\|^2.$$

Let $w := \frac{1}{2}\min\left\{(1-\gamma)\lambda_{\min}(\hat{G}_{\xi_*}), \lambda_{\min}(G_\psi)\right\}$. Suppose $3\sigma LH + L_r \leq 2w$. Then, we have

$$- \langle \Delta\xi, \bar{g}_\xi(\xi) - \bar{\mathfrak{g}}_{\xi_*}(\xi_*) \rangle \leq -w\|\Delta\xi\|^2.$$

The above inequality suggests that $-\bar{\mathfrak{g}}_\xi(\xi)$ is a descent direction for $\|\Delta\xi\|$ if the Lipschitz constants are small enough. $\qquad\square$

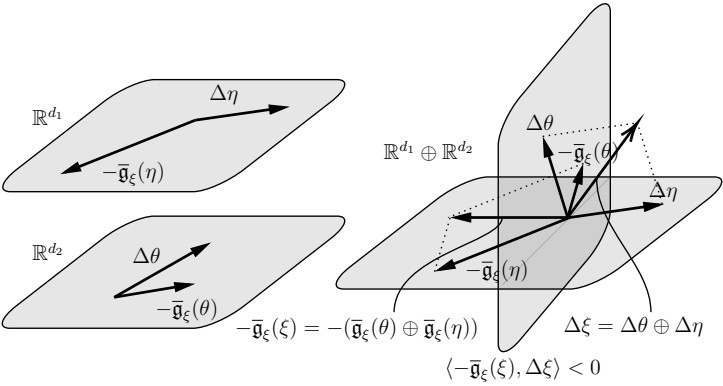

Figure 14: An illustrative example for Lemma 1, where $-\bar{\mathfrak{g}}_\xi(\xi)$ gives a descent direction, while $-\bar{\mathfrak{g}}_\xi(\theta)$ does not.

**Lemma 10** (Progress). *For any $\xi = (\theta; \eta)$ in $B_D^{d_1} \times \Delta^{d_2}$, and for any time step $t$ and period $\tau$, we have*

$$\|\bar{\mathfrak{g}}_t(\xi) - \bar{\mathfrak{g}}_{t-\tau}(\xi)\| \leq \alpha_{t-\tau}\tau C_{\text{prog}},$$

*where $C_{\text{prog}} := \sigma\max\{L_\pi, L_P\}H^2 + 2L_r F$.*

*Proof.* By Lemma 7, we have

$$\|\bar{\mathfrak{g}}_t(\xi) - \bar{\mathfrak{g}}_{t-\tau}(\xi)\| \leq \sigma LH \left(\|\theta_t - \theta_{t-\tau}\| + \|\eta_t - \eta_{t-\tau}\|\right) + L_r\|\eta_t - \eta_{t-\tau}\|.$$

According to the update rule of $\theta$ and $\eta$, we have

$$\|\theta_t - \theta_{t-1}\| \leq \|\breve{\theta}_t - \theta_{t-1}\| = \|\alpha_{t-1}\mathfrak{g}_{t-1}(\theta_{t-1})\|$$
$$\|\eta_t - \eta_{t-1}\| \leq \|\breve{\eta}_t - \eta_{t-1}\| = \|\alpha_{t-1}\mathfrak{g}_{t-1}(\eta_{t-1})\|.$$

Since $\|\theta_{t-1}\| \leq D$ and $\|\eta_{t-1}\| \leq \|\eta_{t-1}\|_1 = 1$, by Lemma 3, we have

$$\|\theta_t - \theta_{t-1}\| + \|\eta_t - \eta_{t-1}\| \leq \alpha_{t-1}\left((1+\gamma)D + R\right) + \alpha_{t-1} \cdot 2F = \alpha_{t-1}H.$$

By the triangle inequality, we get

$$\|\bar{\mathfrak{g}}_t(\xi) - \bar{\mathfrak{g}}_{t-\tau}(\xi)\| \leq \sigma LH \sum_{l=0}^{\tau-1}\left(\|\theta_{t-l} - \theta_{t-l-1}\| + \|\eta_{t-l} - \eta_{t-l-1}\|\right) + L_r\sum_{l=0}^{\tau-1}\|\eta_{t-l} - \eta_{t-l-1}\|$$

$$\leq \alpha_{t-\tau}\tau\sigma LH^2 + 2\alpha_{t-\tau}\tau L_r F$$

$$=: \alpha_{t-\tau}\tau C_{\text{prog}},$$

where we require that the step size $\alpha_t$ is non-increasing, and $C_{\text{prog}} := \sigma\max\{L_\pi, L_P\}H^2 + 2L_r F$. $\qquad\square$

**Lemma 11** (Mix). *Let $\mathcal{F}_{t-\tau}$ be the filtration generated by the history up to time $t - \tau$. For any $\xi = (\theta; \eta) \in B_D^{d_1} \times \Delta^{d_2}$ that is independent of $\mathfrak{g}_{t-\tau}$ and $\bar{\mathfrak{g}}_{t-\tau}$ conditioned on $\mathcal{F}_{t-\tau}$, and for any time step $t$ and period $\tau$, we have*

$$\left\| \mathbb{E}\left[ \mathfrak{g}_{t-\tau}(\xi; \widetilde{O}_t) - \bar{\mathfrak{g}}_{t-\tau}(\xi) \,\middle|\, \mathcal{F}_{t-\tau} \right] \right\| \le m\rho^\tau H.$$

*Proof.* We denote $P_{t-\tau}$ as the observation distribution on the virtual trajectory by fixing the transition kernel $P_{\xi_{t-\tau}}$ at time $t - \tau$. We have

$$\left\| \mathbb{E}\left[ \mathfrak{g}_{t-\tau}(\xi, \widetilde{O}_t) - \bar{\mathfrak{g}}_{t-\tau}(\xi) \,\middle|\, \mathcal{F}_{t-\tau} \right] \right\|$$

$$= \left\| \int_{\mathcal{S}^2 \times \mathcal{A}^2} \left( \phi(s,a)\left(r(s,a) + \gamma\phi^T(s',a')\theta - \phi^T(s,a)\theta\right); \hat{\psi}(s') \right) \left( P_{t-\tau}(\widetilde{O}_t = o \,|\, \mathcal{F}_{t-\tau}) - \mu_{t-\tau}^{\ddagger}(o) \right) \mathrm{d}o \right\|$$

$$\le ((1+\gamma)D + R + F) \left\| P_{t-\tau} - \mu_{\xi_{t-\tau}}^{\ddagger} \right\|_{\mathrm{TV}}.$$

Since $P_{t-\tau}$ and $\mu_{\xi_{t-\tau}}^{\ddagger}$ share the same policy $\pi_{\theta_{t-\tau}}$ and transition kernel $P_{\xi_{t-\tau}}$, we have

$$\left\| P_{t-\tau} - \mu_{\xi_{t-\tau}}^{\ddagger} \right\|_{\mathrm{TV}} = \left\| P_{t-\tau}(s_t = \cdot \,|\, \mathcal{F}_{t-\tau}) - \mu_{\xi_{t-\tau}} \right\|_{\mathrm{TV}} \le m\rho^\tau H,$$

where the last inequality uses Assumption 3. $\qquad\square$

**Lemma 12** (Bakctrack). *Let $\mathcal{F}_{t-\tau}$ be the filtration generated by the history up to time $t - \tau$. For any $\xi = (\theta; \eta) \in B_D^{d_1} \times \Delta^{d_2}$ that is independent of $\mathfrak{g}_t$ and $\mathfrak{g}_{t-\tau}$ conditioned on $\mathcal{F}_{t-\tau}$, and for any time step $t$ and period $\tau$, we have*

$$\left\| \mathbb{E}\left[ \mathfrak{g}_t(\xi, O_t) - \mathfrak{g}_{t-\tau}(\xi, \widetilde{O}_t) \,\middle|\, \mathcal{F}_{t-\tau} \right] \right\| \le \alpha_{t-\tau}\tau C_{\mathrm{back}}(\tau),$$

*where $C_{\mathrm{back}}(\tau) := (\tau+1)LH^2 + 2L_r F$.*

*Proof.* By definition, we have

$$\left\| \mathbb{E}\left[ \mathfrak{g}_t(\xi, O_t) - \mathfrak{g}_{t-\tau}(\xi, \widetilde{O}_t) \,\middle|\, \mathcal{F}_{t-\tau} \right] \right\|$$

$$= \left\| \int_{\mathcal{S}^2 \times \mathcal{A}^2} (G_\phi(o) \oplus G_\psi)\xi \cdot \left( P_t(O_t = o \,|\, \mathcal{F}_{t-\tau}) - P_{t-\tau}(\widetilde{O}_t = o \,|\, \mathcal{F}_{t-\tau}) \right) \mathrm{d}o \right.$$

$$\left. - \int_{\mathcal{S}^2 \times \mathcal{A}^2} (\varphi_{\eta_t} \oplus \psi)(o)P_t(O_t = o \,|\, \mathcal{F}_{t-\tau}) - \left(\varphi_{\eta_{t-\tau}} \oplus \psi\right)(o)P_{t-\tau}(\widetilde{O}_t = o \,|\, \mathcal{F}_{t-\tau})\mathrm{d}o \right\|$$

$$\le ((1+\gamma)D + F)\|P_t - P_{t-\tau}\|_{\mathrm{TV}}$$

$$+ \left\| \int_{\mathcal{S}^2 \times \mathcal{A}^2} (\varphi_{\eta_t} \oplus \psi)(o)\left( P_t(O_t = o \,|\, \mathcal{F}_{t-\tau}) - P_{t-\tau}(\widetilde{O}_t = o \,|\, \mathcal{F}_{t-\tau}) \right)\mathrm{d}o \right\|_{\mathrm{TV}}$$

$$+ \left\| \int_{\mathcal{S}^2 \times \mathcal{A}^2} (\varphi_{\eta_t} - \varphi_{\eta_{t-\tau}})(o)P_{t-\tau}(\widetilde{O}_t = o \,|\, \mathcal{F}_{t-\tau})\mathrm{d}o \right\|_{\mathrm{TV}}$$

$$\le ((1+\gamma)D + F + R + F)\|P_t - P_{t-\tau}\|_{\mathrm{TV}} + L_r\|\eta_t - \eta_{t-\tau}\|$$

where $P_t$ and $P_{t-\tau}$ are the distributions of observation at time step $t$ on the actual trajectory and the virtual trajectory, respectively. By the proof of Lemma 10, we have $\|\eta_t - \eta_{t-\tau}\| \le \alpha_{t-\tau}\tau \cdot 2F$. Therefore, we have

$$\left\| \mathbb{E}\left[ \mathfrak{g}_t(\xi, O_t) - \mathfrak{g}_{t-\tau}(\xi, \widetilde{O}_t) \,\middle|\, \mathcal{F}_{t-\tau} \right] \right\| \le H\|P_t - P_{t-\tau}\|_{\mathrm{TV}} + 2\alpha_{t-\tau}\tau L_r F. \tag{13}$$

Let $\Xi$ be the set of all parameters. We first expand $P_t$ with conditional probabilities:

$$P_t(O_t = (s, a, s', a') \,|\, \mathcal{F}_{t-\tau})$$

$$= \int_{\Xi^2} P_t(s_t = s \,|\, \xi_{t-\tau}, s_{t-\tau})P_t(\xi_{t-1} = \xi \,|\, \xi_{t-\tau}, s_{t-\tau}, s_t = s)\pi_\theta(a \,|\, s)$$

$$P(s' \,|\, s, a, \eta')P_t(\xi_t = \xi' \,|\, \xi_{t-\tau}, s_{t-\tau}, \xi_{t-1} = \xi, s_t = s, a_t = a)\pi_{\theta'}(a' \,|\, s')\mathrm{d}\xi\mathrm{d}\xi',$$

where $\xi = (\theta; \eta)$ and $\xi' = (\theta'; \eta')$. We then decompose $P_{t-\tau}$ into a similar form:

$$P_{t-\tau}(\widetilde{O}_t = (s, a, s', a') \,|\, \mathcal{F}_{t-\tau})$$
$$= P_{t-\tau}(\widetilde{s}_t = s \,|\, \xi_{t-\tau}, s_{t-\tau})\pi_{\theta_{t-\tau}}(a \,|\, s)P(s' \,|\, s, a, \eta_{t-\tau})\pi_{\theta_{t-\tau}}(a' \,|\, s')$$
$$= P_{t-\tau}(\widetilde{s}_t = s \,|\, \xi_{t-\tau}, s_{t-\tau})\pi_{\theta_{t-\tau}}(a \,|\, s)P(s' \,|\, s, a, \eta_{t-\tau})\pi_{\theta_{t-\tau}}(a' \,|\, s')$$
$$\cdot \int_{\Xi^2} P_t(\xi_{t-1} = \xi \,|\, \xi_{t-\tau}, s_{t-\tau}, s_t = s)P_t(\xi_t = \xi' \,|\, \xi_{t-\tau}, s_{t-\tau}, \xi_{t-1} = \xi, s_t = s, a_t = a)\mathrm{d}\xi\mathrm{d}\xi'.$$

Therefore, we can decompose the distribution difference into four parts:

$$P_t - P_{t-\tau}$$
$$= \int_{\Xi^2} \left(P_t(s_t = s \,|\, \mathcal{F}_{t-\tau}) - P_{t-\tau}(\widetilde{s}_t = s \,|\, \mathcal{F}_{t-\tau})\right) P_t(\xi_{t-1} = \xi \,|\, \xi_{t-\tau}, s_{t-\tau}, s_t = s)$$
$$\pi_\theta(a \,|\, s)P(s' \,|\, s, a, \eta)P_t(\xi_t = \xi' \,|\, \xi_{t-\tau}, s_{t-\tau}, \xi_{t-1} = \xi, s_t = s, a_t = a)\pi_{\theta'}(a' \,|\, s')\mathrm{d}\xi\mathrm{d}\xi' \tag{$S_1$}$$

$$+ \int_{\Xi^2} P_{t-\tau}(\widetilde{s}_t = s \,|\, \mathcal{F}_{t-\tau})P_t(\xi_{t-1} = \xi \,|\, \xi_{t-\tau}, s_{t-\tau}, s_t = s) \left(\pi_\theta(a \,|\, s) - \pi_{\theta_{t-\tau}}(a \,|\, s)\right)$$
$$P(s' \,|\, s, a, \eta)P_t(\xi_t = \xi' \,|\, \xi_{t-\tau}, s_{t-\tau}, \xi_{t-1} = \xi, s_t = s, a_t = a)\pi_{\theta'}(a' \,|\, s')\mathrm{d}\xi\mathrm{d}\xi' \tag{$S_2$}$$

$$+ \int_{\Xi^2} P_{t-\tau}(\widetilde{s}_t = s \,|\, \mathcal{F}_{t-\tau})P_t(\xi_{t-1} = \xi \,|\, \xi_{t-\tau}, s_{t-\tau}, s_t = s)\pi_{\theta_{t-\tau}}(a \,|\, s) \left(P(s' \,|\, s, a, \eta') - P(s' \,|\, s, a, \eta_{t-\tau})\right)$$
$$P_t(\xi_t = \xi' \,|\, \xi_{t-\tau}, s_{t-\tau}, \xi_{t-1} = \xi, s_t = s, a_t = a)\pi_{\theta'}(a' \,|\, s')\mathrm{d}\xi\mathrm{d}\xi' \tag{$S_3$}$$

$$+ \int_{\Xi^2} P_{t-\tau}(\widetilde{s}_t = s \,|\, \mathcal{F}_{t-\tau})P_t(\xi_{t-1} = \xi \,|\, \xi_{t-\tau}, s_{t-\tau}, s_t = s)\pi_{\theta_{t-\tau}}(a \,|\, s)P(s' \,|\, s, a, \eta_{t-\tau})$$
$$P_t(\xi_t = \xi' \,|\, \xi_{t-\tau}, s_{t-\tau}, \xi_{t-1} = \xi, s_t = s, a_t = a) \left(\pi_{\theta'}(a' \,|\, s') - \pi_{\theta_{t-\tau}}(a' \,|\, s')\right) \mathrm{d}\xi\mathrm{d}\xi'. \tag{$S_4$}$$

We now bound each part separately. By integrating out the later part, $S_1$ becomes the marginal difference of the state distribution:

$$\int_{\mathcal{S}^2 \times \mathcal{A}^2} S_1 \mathrm{d}s\mathrm{d}a\mathrm{d}s'\mathrm{d}a' = \int_{\mathcal{S}} \left(P_t(s_t = s \,|\, \mathcal{F}_{t-\tau}) - P_{t-\tau}(\widetilde{s}_t = s \,|\, \mathcal{F}_{t-\tau})\right)$$
$$\cdot \left(\int_{\mathcal{S} \times \mathcal{A}^2 \times \Xi^2} P_t(\xi_{t-1} = \xi \,|\, \xi_{t-\tau}, s_{t-\tau}, s_t = s)\pi_\theta(a \,|\, s)P(s' \,|\, s, a, \eta')\right.$$
$$\left. P_t(\xi_t = \xi' \,|\, \xi_{t-\tau}, s_{t-\tau}, \xi_{t-1} = \xi, s_t = s, a_t = a)\pi_{\theta'}(a' \,|\, s')\mathrm{d}\xi\mathrm{d}\xi'\mathrm{d}a\mathrm{d}s'\mathrm{d}a'\right)\mathrm{d}s$$
$$= \int_{\mathcal{S}} \left(P_t(s_t = s \,|\, \mathcal{F}_{t-\tau}) - P_{t-\tau}(\widetilde{s}_t = s \,|\, \mathcal{F}_{t-\tau})\right)\mathrm{d}s$$
$$\leq \|P_t(s_t = \cdot \,|\, \mathcal{F}_{t-\tau}) - P_{t-\tau}(\widetilde{s}_t = \cdot \,|\, \mathcal{F}_{t-\tau})\|_{\mathrm{TV}}$$
$$= \|P_{t-1}(s_t = \cdot \,|\, \mathcal{F}_{t-\tau}) - P_{t-\tau}(\widetilde{s}_t = \cdot \,|\, \mathcal{F}_{t-\tau})\|_{\mathrm{TV}}.$$

$\square$

By Jensen's inequality, we have

$$\|P_{t-1}(s_t = \cdot \,|\, \mathcal{F}_{t-\tau}) - P_{t-\tau}(\widetilde{s}_t = \cdot \,|\, \mathcal{F}_{t-\tau})\|_{\mathrm{TV}}$$
$$= \left\|\int_{\mathcal{S} \times \mathcal{A}^2} P_{t-1}(O_{t-1} = (s, a, \cdot, a') \,|\, \mathcal{F}_{t-\tau}) - P_{t-\tau}(\widetilde{O}_{t-1} = (s, a, \cdot, a') \,|\, \mathcal{F}_{t-\tau})\mathrm{d}s\mathrm{d}a\mathrm{d}a'\right\|_{\mathrm{TV}}$$
$$\leq \int_{\mathcal{S} \times \mathcal{A}^2} \left\|P_{t-1}(O_{t-1} = (s, a, \cdot, a') \,|\, \mathcal{F}_{t-\tau}) - P_{t-\tau}(\widetilde{O}_{t-1} = (s, a, \cdot, a') \,|\, \mathcal{F}_{t-\tau})\right\|_{\mathrm{TV}} \mathrm{d}s\mathrm{d}a\mathrm{d}a'$$
$$= \left\|P_{t-1}(O_t = \cdot \,|\, \mathcal{F}_{t-\tau}) - P_{t-\tau}(\widetilde{O} = \cdot \,|\, \mathcal{F}_{t-\tau})\right\|_{\mathrm{TV}}.$$

That is, $S_1$ is recursively bounded by $\|P_{t-1} - P_{t-\tau}\|_{\mathrm{TV}}$.

For $S_2$, we have

$$\int_{\mathcal{S}^2 \times \mathcal{A}^2} S_2 \mathrm{d}s \mathrm{d}a \mathrm{d}s' \mathrm{d}a'$$

$$= \int_{\mathcal{S} \times \mathcal{A} \times \Xi} P_{t-\tau}(\widetilde{s}_t = s \mid \mathcal{F}_{t-\tau}) P_t(\xi_{t-1} = \xi \mid \xi_{t-\tau}, s_{t-\tau}, s_t = s) \left( \pi_\theta(a \mid s) - \pi_{\theta_{t-\tau}}(a \mid s) \right)$$

$$\cdot \left( \int_{\mathcal{S} \times \mathcal{A} \times \Xi} P(s' \mid s, a, \eta') P_t(\xi_t = \xi' \mid \xi_{t-\tau}, s_{t-\tau}, \xi_{t-1} = \xi, s_t = s, a_t = a) \pi_{\theta'}(a' \mid s') \mathrm{d}s' \mathrm{d}a' \mathrm{d}\xi' \right) \mathrm{d}s \mathrm{d}a \mathrm{d}\xi$$

$$= \int_{\mathcal{S} \times \mathcal{A} \times \Xi} P_{t-\tau}(\widetilde{s}_t = s \mid \mathcal{F}_{t-\tau}) P_t(\xi_{t-1} = \xi \mid \xi_{t-\tau}, s_{t-\tau}, s_t = s) \left( \pi_\theta(a \mid s) - \pi_{\theta_{t-\tau}}(a \mid s) \right) \mathrm{d}s \mathrm{d}a \mathrm{d}\xi$$

$$\leq \int_{\mathcal{S} \times \Xi} P_{t-\tau}(\widetilde{s}_t = s \mid \mathcal{F}_{t-\tau}) P_t(\xi_{t-1} = \xi \mid \xi_{t-\tau}, s_{t-\tau}, s_t = s) \left\| \pi_\theta(\cdot \mid s) - \pi_{\theta_{t-\tau}}(\cdot \mid s) \right\|_{\mathrm{TV}} \mathrm{d}s \mathrm{d}\xi.$$

By Assumption 2 and Lemma 10, for any $\xi \in \Xi$ such that $P_t(\xi_{t-1} = \xi \mid \mathcal{F}_{t-\tau}) \neq 0$ (which is equivalent to $P_t(\xi_{t-1} = \xi \mid \xi_{t-\tau}, s_{t-\tau}, s_t = s) \neq 0$ as all policies are ergodic), we have

$$\|\pi_\theta(\cdot \mid s) - \pi_{\theta_{t-\tau}}(\cdot \mid s)\|_{\mathrm{TV}} \leq \alpha_{t-\tau}(\tau - 1) L_\pi((1 + \gamma)G + R).$$

Therefore, we get

$$\int_{\mathcal{S}^2 \times \mathcal{A}^2} S_2 \mathrm{d}s \mathrm{d}a \mathrm{d}s' \mathrm{d}a' \leq \alpha_{t-\tau}(\tau - 1) L_\pi((1 + \gamma)G + R) \int_{\mathcal{S} \times \Xi} P_{t-\tau}(\widetilde{s}_t = s \mid \mathcal{F}_{t-\tau}) P_t(\xi_{t-1} = \xi \mid \xi_{t-\tau}, s_{t-\tau}, s_t = s) \mathrm{d}s \mathrm{d}\xi$$

$$= \alpha_{t-\tau}(\tau - 1) L_\pi((1 + \gamma)G + R).$$

For $S_3$, we have

$$\int_{\mathcal{S}^2 \times \mathcal{A}^2} S_3 \mathrm{d}s \mathrm{d}a \mathrm{d}s' \mathrm{d}a'$$

$$= \int_{\mathcal{S}^2 \times \mathcal{A} \times \Xi^2} P_{t-\tau}(\widetilde{s}_t = s \mid \mathcal{F}_{t-\tau}) P_t(\xi_{t-1} = \xi \mid \xi_{t-\tau}, s_{t-\tau}, s_t = s) \pi_{\theta_{t-\tau}}(a \mid s)$$

$$\cdot \left( P(s' \mid s, a, \eta') - P(s' \mid s, a, \eta_{t-\tau}) \right) P_t(\xi_t = \xi' \mid \xi_{t-\tau}, s_{t-\tau}, \xi_{t-1} = \xi, s_t = s, a_t = a)$$

$$\cdot \left( \int_{\mathcal{A}} \pi_{\theta'}(a' \mid s') \mathrm{d}a' \right) \mathrm{d}s \mathrm{d}a \mathrm{d}s' \xi \mathrm{d}\xi'$$

$$\leq \int_{\mathcal{S} \times \mathcal{A} \times \Xi^2} P_{t-\tau}(\widetilde{s}_t = s \mid \mathcal{F}_{t-\tau}) P_t(\xi_{t-1} = \xi \mid \xi_{t-\tau}, s_{t-\tau}, s_t = s) \pi_{\theta_{t-\tau}}(a \mid s)$$

$$\cdot \|P(\cdot \mid s, a, \eta') - P(\cdot \mid s, a, \eta_{t-\tau})\|_{\mathrm{TV}} P_t(\xi_t = \xi' \mid \xi_{t-\tau}, s_{t-\tau}, \xi_{t-1} = \xi, s_t = s, a_t = a) \mathrm{d}s \mathrm{d}a \mathrm{d}\xi \mathrm{d}\xi'.$$

Similar to the bound for $S_2$, by Assumption 1 and Lemma 10, for any $\xi' \in \Xi$ such that $P_t(\xi_t = \xi' \mid \mathcal{F}_{t-\tau}) \neq 0$, we have

$$\|P(\cdot \mid s, a, \eta') - P(\cdot \mid s, a, \eta_{t-\tau})\|_{\mathrm{TV}} \leq 2\alpha_{t-\tau}\tau L_P.$$

Therefore, we get

$$\int_{\mathcal{S}^2 \times \mathcal{A}^2} S_3 \mathrm{d}s \mathrm{d}a \mathrm{d}s' \mathrm{d}a' \leq \alpha_{t-\tau}\tau L_P H \int_{\mathcal{S} \times \mathcal{A} \times \Xi^2} P_{t-\tau}(\widetilde{s}_t = s \mid \mathcal{F}_{t-\tau}) P_t(\xi_{t-1} = \xi \mid \xi_{t-\tau}, s_{t-\tau}, s_t = s)$$

$$\pi_{\theta_{t-\tau}}(a \mid s) P_t(\xi_t = \xi' \mid \xi_{t-\tau}, s_{t-\tau}, \xi_{t-1} = \xi, s_t = s, a_t = a) \mathrm{d}s \mathrm{d}a \mathrm{d}\xi \mathrm{d}\xi'$$

$$= 2\alpha_{t-\tau}\tau L_P.$$

Similar to the bound for $S_2$, for $S_4$, we have

$$\int_{\mathcal{S}^2 \times \mathcal{A}^2} S_4 \mathrm{d}s \mathrm{d}a \mathrm{d}s' \mathrm{d}a' \leq \alpha_{t-\tau}\tau L_\pi((1 + \gamma)G + R).$$

Plugging back the above bounds for $S_1$, $S_2$, $S_3$, and $S_4$, we get

$$\|P_t - P_{t-\tau}\|_{\mathrm{TV}} \leq \|P_{t-1} - P_{t-\tau}\|_{\mathrm{TV}} + 2\tau\alpha_{t-\tau}(L_\pi((1 + \gamma)D + R) + L_P F)$$

.

Recursively, we get

$$\|P_t - P_{t-\tau}\|_{\mathrm{TV}} \le 2LH \sum_{l=1}^{\tau} (l\alpha_{t-l}) \le \alpha_{t-\tau}\tau(\tau+1)LH,$$

where we require the step size to be non-increasing. Plugging the above bound back to (13) gives the desired result.

## H.2 PROOF OF THEOREM 1

**Theorem 1.** *Let $\xi_* = (\theta_*; \eta_*)$ be a (projected) MFE. Let $\{\xi_t = (\theta_t; \eta_t)\}$ be a sequence of parameters generated by Algorithm 1. Then, under Assumptions 1 to 3, if $3\sigma LH + L_r \le 2w$, we have*

$$\mathbb{E}\|\xi_{t+1} - \xi_*\|^2 \le (1 - \alpha_t w)\mathbb{E}\|\xi_t - \xi_*\|^2 + \alpha_t^2 H^2 + O\left(\frac{\alpha_t^3 \log^4 \alpha_t^{-1} L^2 H^4}{w}\right).$$

*Proof.* We denote $\Delta \xi_t = \xi_t - \xi_*$. We first plug Lemmas 1 and 3 into Lemma 8 to get

$$\mathbb{E}\|\Delta\xi_{t+1}\|^2 \le (1 - 2\alpha_t w)\mathbb{E}\|\Delta\xi_t\|^2 + \alpha_t^2 H^2$$
$$+ 2\alpha_t \mathbb{E}\langle \Delta\xi, \bar{\mathfrak{g}}_t(\xi_t) - \bar{\mathfrak{g}}_{t-\tau}(\xi_t)\rangle \tag{14}$$
$$+ 2\alpha_t \mathbb{E}\langle \Delta\xi, \bar{\mathfrak{g}}_{t-\tau}(\xi_t) - \mathfrak{g}_{t-\tau}(\xi_t)\rangle \tag{15}$$
$$+ 2\alpha_t \mathbb{E}\langle \Delta\xi, \mathfrak{g}_{t-\tau}(\xi_t) - \mathfrak{g}_t(\xi_t)\rangle. \tag{16}$$

For (14), by Young's inequality, for any $\beta > 0$, we have

$$2\mathbb{E}\langle \Delta\xi_t, \bar{\mathfrak{g}}_t(\xi_t) - \bar{\mathfrak{g}}_{t-\tau}(\xi_t)\rangle \le \beta\mathbb{E}\|\Delta\xi_t\|^2 + \frac{1}{\beta}\mathbb{E}\|\bar{\mathfrak{g}}_{t-\tau}(\xi_t) - \bar{\mathfrak{g}}_t(\xi_t)\|^2.$$

By Lemma 10, we get

$$2\mathbb{E}\langle \Delta\xi_t, \bar{\mathfrak{g}}_t(\xi_t) - \bar{\mathfrak{g}}_{t-\tau}(\xi_t)\rangle \le \beta\mathbb{E}\|\Delta\xi_t\|^2 + \beta^{-1}\alpha_{t-\tau}^2\tau^2 C_{\mathrm{prog}}^2. \tag{17}$$

For (15), note that conditioned on $\mathcal{F}_{t-\tau}$, $\bar{\mathfrak{g}}_{t-\tau}$ is determined and $\mathfrak{g}_{t-\tau}$ is only dependent on the virtual Markovian trajectory. Thus, $\xi_t$ is independent of $\mathfrak{g}_{t-\tau}$ and $\bar{\mathfrak{g}}_{t-\tau}$ conditioned on $\mathcal{F}_{t-\tau}$. Therefore, by Lemma 11 and Young's inequality, we have

$$2\mathbb{E}\langle \Delta\xi_t, \bar{\mathfrak{g}}_{t-\tau}(\xi_t) - \mathfrak{g}_{t-\tau}(\xi_t)\rangle = 2\mathbb{E}\left[\langle \mathbb{E}\left[\Delta\xi_t \mid \mathcal{F}_{t-\tau}\right], \mathbb{E}\left[\mathfrak{g}_{t-\tau}(\xi_t) - \bar{\mathfrak{g}}_{t-\tau}(\xi_t) \mid \mathcal{F}_{t-\tau}\right]\rangle\right]$$
$$\le 2\mathbb{E}\left[\mathbb{E}\left[\|\Delta\xi_t\| \mid \mathcal{F}_{t-\tau}\right] \cdot \|\mathbb{E}\left[\mathfrak{g}_{t-\tau}(\xi_t) - \bar{\mathfrak{g}}_{t-\tau}(\xi_t) \mid \mathcal{F}_{t-\tau}\right]\|\right]$$
$$\le 2m\rho^\tau H\mathbb{E}\|\Delta\xi_t\|$$
$$\le \beta\mathbb{E}\|\Delta\xi_t\|^2 + \beta^{-1}m^2\rho^{2\tau}H^2. \tag{18}$$

For (16), we cannot directly apply Lemma 12 as $\xi_t$ and $\mathfrak{g}_t$ are dependent conditioned on $\mathcal{F}_{t-\tau}$. To proceed, we employ the following decomposition:

$$\mathbb{E}\langle \Delta\xi_t, \mathfrak{g}_{t-\tau}(\xi_t) - \mathfrak{g}_t(\xi_t)\rangle = \underbrace{\mathbb{E}\langle \Delta\xi_t, (\mathfrak{g}_{t-\tau}(\xi_t) - \mathfrak{g}_{t-\tau}(\xi_{t-\tau})) - (\mathfrak{g}_t(\xi_t) - \mathfrak{g}_t(\xi_{t-\tau}))\rangle}_{H_1}$$
$$+ \underbrace{\mathbb{E}\langle \xi_t - \xi_{t-\tau}, \mathfrak{g}_{t-\tau}(\xi_{t-\tau}) - \mathfrak{g}_t(\xi_{t-\tau})\rangle}_{H_2} + \underbrace{\mathbb{E}\langle \Delta\xi_{t-\tau}, \mathfrak{g}_{t-\tau}(\xi_{t-\tau}) - \mathfrak{g}_t(\xi_{t-\tau})\rangle}_{H_3}.$$

For $H_1$, by the Lipschitzness of the semi-gradient (Lemma 7) and Lemma 10, we have

$$H_1 \le 2\alpha_{t-\tau}\tau H^2 \cdot \mathbb{E}\|\Delta\xi_t\|.$$

For $H_2$, by Lemmas 3 and 10, we get

$$H_2 \le 2H \cdot \alpha_{t-\tau}\tau H.$$

For $H_3$, $\xi_{t-\tau}$ is independent of $\mathfrak{g}_t$ and $\mathfrak{g}_{t-\tau}$ conditioned on $\mathcal{F}_{t-\tau}$. Similar to the bound of (18), by Lemma 12, we have

$$H_3 = \mathbb{E}\left[\langle \mathbb{E}\left[\Delta\xi_{t-\tau} \mid \mathcal{F}_{t-\tau}\right], \mathbb{E}\left[\mathfrak{g}_{t-\tau}(\xi_{t-\tau}) - \mathfrak{g}_t(\xi_{t-\tau}) \mid \mathcal{F}_{t-\tau}\right]\rangle\right]$$
$$\le \mathbb{E}\left[\mathbb{E}\left[\|\Delta\xi_{t-\tau}\| \mid \mathcal{F}_{t-\tau}\right] \cdot \|\mathbb{E}\left[\mathfrak{g}_t(\xi_{t-\tau}) - \mathfrak{g}_{t-\tau}(\xi_{t-\tau}) \mid \mathcal{F}_{t-\tau}\right]\|\right]$$
$$\le \alpha_{t-\tau}\tau C_{\mathrm{back}}\mathbb{E}\|\Delta\xi_{t-\tau}\|.$$

Then, applying Lemma 10 and Young's inequality gives

$$2H_3 \leq \alpha_{t-\tau}\tau C_{\text{back}} \left(\alpha_{t-\tau}\tau H + \mathbb{E}\|\Delta\xi_t\|^2\right) \leq 2\alpha_{t-\tau}^2\tau^2 HC_{\text{back}} + \beta\mathbb{E}\|\Delta\xi_t\|^2 + \beta^{-1}\alpha_{t-\tau}^2\tau^2 C_{\text{back}}^2.$$

Plugging $H_1$, $H_2$, and $H_3$ back into the decomposition and applying Young's inequality, we get

$$2\mathbb{E}\langle\Delta\xi_t, \mathfrak{g}_{t-\tau}(\xi_t) - \mathfrak{g}_t(\xi_t)\rangle \leq 2\beta\mathbb{E}\|\Delta\xi_t\|^2 + 4\alpha_{t-\tau}\tau H^2 + \alpha_{t-\tau}^2\tau^2\left(2HC_{\text{back}} + \beta^{-1}\left(C_{\text{back}}^2 + 4H^4\right)\right). \tag{19}$$

Finally, plugging the bounds of (17–19) back into (14–16) gives

$$\mathbb{E}\|\Delta\xi_{t+1}\|^2 \leq (1 - 2\alpha_t w + 4\alpha_t\beta)\mathbb{E}\|\Delta\xi_t\|^2 + \alpha_t^2 H^2 + 4\alpha_t\alpha_{t-\tau}\tau H^2$$
$$+ \alpha_t\left(\beta^{-1}\alpha_{t-\tau}^2\tau^2 C_{\text{prog}}^2 + \beta^{-1}m^2\rho^{2\tau}H^2 + \alpha_{t-\tau}^2\tau^2(2HC_{\text{back}} + \beta^{-1}(C_{\text{back}}^2 + 4H^4))\right).$$

Now we choose $\tau = \left\lceil\frac{\log\alpha_t^{-1}}{\log\rho^{-1}}\right\rceil$. Then, $\rho^\tau \leq \alpha_t$. We require the step-size to be non-increasing, and not too small such that

$$\sum_{t=1}^{\infty}\alpha_t = +\infty \quad \text{and} \quad \log\alpha_t^{-1} = o(T).$$

The first condition indicates that $\limsup_{t\to\infty}\alpha_{t/2}/\alpha_t := C_{\alpha,1} < \infty$. And there exists $C_{\alpha,2} > 0$ such that $\alpha_{t/2}/\alpha_t \leq C_{\alpha,2}\cdot\limsup_{t\to\infty}\alpha_{t/2}/\alpha_t$ for any time step $t$. The second condition indicates the existence of $C_{\alpha,2} > 0$ such that $\alpha_{t-\tau} \leq C_{\alpha,3}\alpha_{t/2}$ for any $t$. In conclusion, we have

$$\alpha_{t-\tau} \leq C_{\alpha,3}\alpha_{t/2} \leq C_{\alpha,3}C_{\alpha,2}C_{\alpha,1}\alpha_t =: C_\alpha\alpha_t.$$

Then, we choose $\beta = w/4$. Together, we get

$$\mathbb{E}\|\Delta\xi_{t+1}\|^2 \leq (1 - \alpha_t w)\mathbb{E}\|\Delta\xi_t\|^2 + \alpha_t^2(H^2 + 4\tau C_\alpha H^2) \tag{20}$$
$$+ \alpha_t^3\left(\frac{4}{w}\left(\tau^2 C_\alpha^2(C_{\text{prog}}^2 + C_{\text{back}}^2 + 4H^4) + m^2 H^2\right) + 2\tau^2 C_\alpha^2 C_{\text{back}}H\right)$$
$$=: (1 - \alpha_t w)\mathbb{E}\|\Delta\xi_t\|^2 + \alpha_t^2\cdot C_2(\alpha_t) + \alpha_t^3\cdot C_3(\alpha_t),$$

where we use $C_2$ and $C_3$ to encapsulate the terms in the right-hand side of (20). Plugging back the definitions of $C_{\text{prog}}$ and $C_{\text{back}}$ gives the final result

$$\mathbb{E}\|\Delta\xi_{t+1}\|^2 = (1 - \alpha_t w)\mathbb{E}\|\Delta\xi_t\|^2 + H^2\cdot O(\alpha_t^2\log\alpha_t^{-1}) + \frac{\max\{L_\pi, L_P, L_r\}^2 H^4}{w}\cdot O(\alpha_t^3\log^4\alpha_t^{-1}),$$

where the asymptotic equivalence holds as $\alpha_t \to 0$. □

## H.3 PROOF OF COROLLARIES

**Corollary 1.** *For a constant step-size $\alpha_t \equiv \alpha_0$, we have*

$$\mathbb{E}\|\xi_t - \xi_*\|^2 \leq e^{-\alpha_0 wt}\|\xi_0 - \xi_*\|^2 + \frac{\alpha_0 H^2}{w} + O\left(\frac{L^2 H^4\alpha_0^2}{w^2}\right).$$

*Proof.* By Theorem 1, we have

$$\mathbb{E}\|\xi_T - \xi_*\|^2 \leq (1 - \alpha_0 w)^T\|\xi_0 - \xi_*\|^2 + \sum_{t=0}^{T}(1 - \alpha_0 w)^{T-t}(\alpha_0^2 C_2 + \alpha_0^3 C_3)$$

$$\leq e^{-\alpha_0 wT}\|\xi_0 - \xi_*\|^2 + \frac{\alpha_0}{w}(C_2 + \alpha_0 C_3)$$

$$\leq e^{-\alpha_0 wT}\|\xi_0 - \xi_*\|^2 + w^{-1}H^2\cdot O(\alpha_0\log\alpha_0^{-1}).$$

□

**Corollary 3** (Linearly decaying step-size)**.** *We define a linearly decaying step-size sequence $\alpha_t = 4/(w(t+1))$, and the convex combination $\tilde{\xi}_T := \frac{2}{T(T+1)}\sum_{t=0}^{T}t\xi_t$. Then, we have*

$$\mathbb{E}\|\tilde{\xi}_T - \xi_*\|^2 \leq O\left(\frac{H^2\log T}{w^2 T}\right).$$

*Proof.* Rearranging the result of Theorem 1 gives

$$\frac{1}{2}\mathbb{E}\|\Delta\xi_t\|^2 \leq \left(\frac{1}{\alpha_t w} - \frac{1}{2}\right)\mathbb{E}\|\Delta\xi_t\|^2 - \frac{1}{\alpha_t w}\mathbb{E}\|\Delta\xi_{t+1}\|^2 + \frac{\alpha_t}{w}C_2 + \frac{\alpha_t^2}{w}C_3.$$

Substituting $\alpha_t = 4/(w(t+1))$ and multiplying by $t$ gives

$$t\mathbb{E}\|\Delta\xi_t\|^2 \leq \frac{(t-1)t}{2}\mathbb{E}\|\Delta\xi_t\|^2 - \frac{t(t+1)}{2}\mathbb{E}\|\Delta\xi_{t+1}\|^2 + \frac{8t}{w^2(t+1)}C_2 + \frac{32t}{w^3(t+1)^2}C_3. \quad (21)$$

By Jensen's inequality, we have

$$\mathbb{E}\|\tilde{\xi}_T - \xi_*\|^2 \leq \frac{1}{T(T+1)}\sum_{t=0}^{T} t\mathbb{E}\|\Delta\xi_t\|^2. \quad (22)$$

Combining (21) and (22) gives

$$\mathbb{E}\|\tilde{\xi}_T - \xi_*\|^2 \leq \frac{4}{w^2 T(T+1)}\left(C_2\sum_{t=0}^{T}\frac{t}{t+1} + \frac{4C_3}{w}\sum_{t=0}^{T}\frac{t}{(t+1)^2}\right) = O\left(\frac{H^2\log T}{w^2 T}\right).$$

$\square$

## I    CONVERGENCE WITH GENERAL POLICY OPERATORS

Assumption 4 for ensuring contractivity can be restrictive in practice. Its requirement for the policy operator to have a small Lipschitz constant typically requires large regularization (Yardim et al., 2023; Angiuli et al., 2023). However, this may limit the algorithm to only learning near-uniform policies (Zhang et al., 2023), and the learned regularized MFE (Cui & Koeppl, 2021; Anahtarci et al., 2023) may be distant from the true MFE.

On the other hand, without any contractivity or monotonicity assumption, the existence and uniqueness of the MFE are not guaranteed. Assuming an MFE exists that satisfies Proposition 2, we can replace Assumption 4 with an alternative assumption on the reward function and measure basis to ensure that SemiSGD converges to a neighborhood of the MFE.

**Assumption 5.** The reward function and measure basis satisfy that $L_r \leq \bar{w}/2$ and $R + F \leq \bar{w}^2/(4\bar{L})$, where $\bar{L} := \sigma\sqrt{\max\{d_1, d_2\}}\max\{L_P, L_\pi\}$ and $\bar{w} \in (0, w]$ is a problem-dependent constant.

**Theorem 3** (Convergence with general policy operators). *Suppose $\xi_* = (\theta_*; \eta_*)$ is an MFE and Assumptions 1 to 3 and 5 hold. Then, $\xi_*$ is the unique MFE and a linearly decaying step-size sequence $\alpha_t = 4/(1 + \bar{w}t)$ gives*

$$\mathbb{E}\|\xi_T - \xi_*\| \leq O\left(\frac{D(L_r + \bar{L}(R+F)\bar{w}^{-1})}{\bar{w}^2} + \frac{\log T}{\sqrt{T}}\right).$$

Theorem 3 establishes that, SemiSGD with a general policy operator (no restriction on $L_\pi$) converges to a neighborhood centered at the MFE, whose radius scales with the sup norm of the reward function ($R$), its variation ($L_r$), and the measure basis ($F$). The convergence rate is not compromised. To the best of our knowledge, this is the first theoretical convergence result for learning MFGs without a contractivity or monotonicity assumption, although the convergence is to a region rather than a point.

As discussed in Section 2.2, with general Lipschitz constants, SemiSGD is essentially a stochastic approximation method on a rapidly changing Markov chain.

*Remark* 3. Assumption 4 or Assumption 5 is purely for theoretical convergence analysis. None of them is enforced in our numerical experiments. Guided by the insights in Section 2.2, we use near-greedy policies *without regularization* or additional stabilization mechanisms in our algorithm implementation. Our numerical results demonstrate that SemiSGD with general policy operators converges efficiently to a point.

We acknowledge the remaining gaps between our theoretical analysis and practical implementation: 1) Although the condition in Theorem 3 is arguably more practical than the small Lipschitz constants condition (Assumption 4), as it is a *structural* condition satisfied by a class of MFGs, it cannot

explain the empirical success of SemiSGD for general MFGs;[3] 2) The convergence region bound in Theorem 3 may not be tight, and empirically the convergence region can be significantly smaller and may even converge to a point. These gaps partially result from our limited understanding of stochastic approximation methods on rapidly changing Markov chains (Zhang et al., 2023) and call for further investigation into the structure of this class of methods.

## I.1 PROOF OF THEOREM 3

Our convergence result for general policy operators leverages Zhang et al. (2023, Corollary 3.10) for stochastic approximation methods on rapidly changing Markov chains. Thus, we only need to verify that SemiSGD satisfies all the assumptions in Zhang et al. (2023) and specify the corresponding parameters and constants. Table 4 provides a side-by-side comparison of the notations and constants for easy reference.

Table 4: Comparison of notations and constants in Zhang et al. (2023) and our setting.

| Description | Ours | Zhang et al. (2023) |
|---|---|---|
| Parameter | $\xi_t = (\theta_t; \eta_t)$ | $w_t \equiv \theta_t$ |
| Observation sample | $O_t$ | $Y_t$ |
| Algorithm operator | $-\mathfrak{g}(\xi_t; O_t) + \xi_t$ | $F_{\theta_t}(w_t, Y_t)$ |
| Mean-path operator | $\mathrm{id} - \alpha \bar{\mathfrak{g}}_\xi$ | $f_w^\alpha$ |
| Projection operator | $\Pi$ | $\Gamma$ |
| Update rule (6) | $\xi_{t+1} = \Pi(\xi_t - \alpha_t \mathfrak{g}(\xi_t; O_t))$ | $w_{t+1} = \Gamma(w_t - \alpha_t(F_{w_t}(w_t, Y_t) - w_t))$ |
| Steady distribution | $\mu_\xi$ | $d_\theta$ |
| | Assumption 3.4 | |
| Zero/fixed point | $\bar{\mathfrak{g}}_\xi(\Lambda(\xi)) = 0$ | $\bar{F}_\theta(w_\theta^*) = w_\theta^*$ |
| Assumption | Algorithm 1 | Assumption 3.1 |
| Assumption | Assumption 3 | Assumption 3.2 and Lemma 3.3 |
| Assumption | contractivity of Bellman and transition operators | Assumption 3.4 |
| Assumption | Lemma 7 | Assumption 3.5 (i) |
| Assumption | Algorithm 1 | Assumption 3.5 (ii) |
| Assumption | Lemma 3 | Assumption 3.5 (iii) |
| Assumption | Lemma 7 | Assumption 3.5 (iv) |
| Assumption | Lemma 1 | Assumption 3.5 (vi) |
| Assumption | Assumption 1 and Lemma 4 | Assumption 3.5 (vii) |
| Assumption | $D > \bar{w}^{-1}$ | Assumption 3.6 |
| Assumption | $\alpha_t = 4/(1 + \bar{w}t)$ | Assumption 3.7 |
| Constant | $\sqrt{1 - \alpha \bar{w}}$ | $\kappa_\alpha$ |
| Constant | $\bar{w}$ | $\eta$ |
| Constant | $\max\{1 + \gamma, F\} + 1$ | $L_F$ |
| Constant | $0$ | $L_F'$ |
| Constant | $R + F$ | $U_F$ |
| Constant | $\sigma L(1 + \gamma)\sqrt{\max\{d_1, d_2\}}$ | $L_F''$ |
| Constant | $\frac{R+F}{1+\gamma} + \frac{L_r}{\sigma L(1+\gamma)}$ | $U_F''$ |
| Constant | $\bar{w}^{-1}/2$ | $U_{\mathrm{inv}}$ |
| Constant | $\bar{w}^{-1}(R + F)/2$ | $U_w$ |
| Constant | (30) | $L_w$ |
| Constant | $L_P$ | $L_P$ |

---

[3]For general MFGs, we can always scale the reward function and state space to meet the sup norm condition in Assumption 5. However, it is unclear whether the MFE after scaling corresponds to an original MFE.

Zhang et al. (2023, Assumption 3.1) is satisfied by our action selection rule in Algorithm 1:

$$
\begin{aligned}
\Pr\left(O_{t+1} = o = (s, a, s', a')\right) &= \Pr\left((s_{t+1}, a_{t+1}, s_{t+2}, a_{t+2}) = (s, a, s', a')\right) \\
&= P_{\eta_{t+1}}\left(s_{t+1}, a_{t+1}, s'\right) \pi_{\theta_{t+1}}(a' \mid s') \\
&=: P_{\xi_{t+1}}\left(O_{t+1}, o\right).
\end{aligned}
$$

That is, the transition depends on the current parameter.

Zhang et al. (2023, Assumption 3.2) is directly satisfied by Assumption 3.

Zhang et al. (2023, Assumption 3.4 (i)) is satisfied by the contraction property of the Bellman and transition operators. To ease representation, we define a mapping $\Lambda$ that maps any parameter $\xi$ to the zero point of $\bar{\mathfrak{g}}_\xi$. Recall that

$$
\bar{\mathfrak{g}}_\xi(\xi') = \left(\bar{G}_\xi \oplus G_\psi\right) \xi - \left(\bar{\varphi}_\xi \oplus \bar{\psi}_\xi\right).
$$

By previous analysis (e.g., Lemma 9), we know that $\bar{G}_\xi$ and $G_\psi$ are positive definite. Thus, we have

$$
\Lambda(\xi) = \left(\bar{G}_\xi \oplus G_\psi\right)^{-1} \left(\bar{\varphi}_\xi \oplus \bar{\psi}_\xi\right),
$$

indicating the existence and uniqueness of the zero point of $\bar{\mathfrak{g}}_\xi$.

Zhang et al. (2023, Assumption 3.4 (ii)) can be verified by modifying the proof of Lemma 9. For a given parameter $\zeta \in \Xi$, we denote $\Lambda(\zeta) = \bar{\xi} = (\bar{\theta}; \bar{\eta})$, $\Delta\xi = \xi - \bar{\xi}$, $\Delta\theta = \theta - \bar{\theta}$, and $\Delta\eta = \eta - \bar{\eta}$ for any $\xi = (\theta; \eta) \in \Xi$. Then, similar to the proof of Lemma 9, we have

$$
\begin{aligned}
-\langle \Delta\eta, \bar{\mathfrak{g}}_\zeta(\eta)\rangle &\leq -\lambda_{\min}(G_\psi)\|\Delta\eta\|^2, \\
-\langle \Delta\theta, \bar{\mathfrak{g}}_\zeta(\theta)\rangle &\leq -(1-\gamma)\lambda_{\min}(\hat{G}_\zeta)\|\Delta\theta\|^2.
\end{aligned}
\tag{23}
$$

Note that here the coupling between $\theta$ and $\eta$ is gone as we fix the parameter $\zeta$ that controls the dynamics, resulting in a stationary setting. Let

$$
\bar{w} := \frac{1}{2} \inf_{\zeta \in \Xi} \min\left\{(1-\gamma)\lambda_{\min}(\hat{G}_\zeta), \lambda_{\min}(G_\psi)\right\}.
$$

Note that $\zeta$ determines $\hat{G}_\zeta$ through the transition kernel. Since the set of transition kernels on a compact state space is compact, the infimum is achieved. And by definition, we have $0 < \bar{w} < w \leq 1/2$. By Lemma 7, we have

$$
\|\bar{\mathfrak{g}}_\zeta(\xi)\| = \|\bar{\mathfrak{g}}_\zeta(\xi) - \bar{\mathfrak{g}}_\zeta(\bar{\xi})\| \leq \max\{1 + \gamma, F\}\|\xi - \bar{\xi}\|.
\tag{24}
$$

Combining (23) and (24) gives

$$
\begin{aligned}
\left\|\xi - \alpha\bar{\mathfrak{g}}_\zeta(\xi) - \bar{\xi}\right\|^2 &\leq \|\Delta\xi\|^2 - 2\alpha\langle\Delta\eta, \bar{\mathfrak{g}}_\zeta(\eta)\rangle - 2\alpha\langle\Delta\theta, \bar{\mathfrak{g}}_\zeta(\theta)\rangle + \alpha^2\|\bar{\mathfrak{g}}_\zeta(\xi)\|^2 \\
&\leq \|\Delta\xi\|^2 - 2\alpha\bar{w}\|\Delta\xi\|^2 + \alpha^2 \max\{1+\gamma, F\}^2 \|\Delta\xi\|^2 \\
&= (1 - 2\alpha\bar{w} + \alpha^2 \max\{1+\gamma, F\}^2)\|\Delta\xi\|^2.
\end{aligned}
$$

Therefore, let $\bar{\alpha} := \bar{w}/(\max\{1+\gamma, F\})^2$. When $\alpha \leq \bar{\alpha}$, the above inequality gives a *pseudo-contraction* (Zhang et al., 2023, Assumption 3.4 (ii)):

$$
\left\|\xi - \alpha\bar{\mathfrak{g}}_\zeta(\xi) - \bar{\xi}\right\| \leq \sqrt{1 - \alpha\bar{w}}\|\xi - \bar{\xi}\|^2.
$$

In Zhang et al. (2023, Assumption 3.5), (i) is satisfied by Lemma 7. (ii) is satisfied by our update rule, because given an observation $O$, $\mathfrak{g}(\xi; O)$ is independent of the parameter controlling the dynamics. (iii) is satisfied by Lemma 3. (iv) is satisfied by Lemma 7:

$$
\begin{aligned}
\|\bar{\mathfrak{g}}_{\xi_1}(\xi) - \bar{\mathfrak{g}}_{\xi_2}(\xi)\| &\leq \sigma L\left((1+\gamma)\|\theta\| + R + F\right)\sqrt{\max\{d_1, d_2\}}\|\xi_1 - \xi_2\| + L_r\|\eta_1 - \eta_2\| \\
&\leq \sigma L(1+\gamma)\sqrt{\max\{d_1, d_2\}}\left(\|\theta\| + \frac{R+F}{1+\gamma} + \frac{L_r}{\sigma L(1+\gamma)}\right)\|\xi_1 - \xi_2\|.
\end{aligned}
$$

Before verifying Zhang et al. (2023, Assumption 3.5 (v) and (vi)), we first show the following norm bound:

$$
\left\| \left( \bar{G}_\xi \oplus G_\psi \right)^{-1} \right\|_{\mathrm{op}} = \sigma_{\min}^{-1} \left( \bar{G}_\xi \oplus G_\psi \right) \tag{25}
$$

$$
\leq \lambda_{\min}^{-1} \left( \mathrm{sym} \left( \bar{G}_\xi \oplus G_\psi \right) \right) \tag{26}
$$

$$
= \min \left\{ \lambda_{\min} \left( \mathrm{sym} \left( \bar{G}_\xi \right) \right), \lambda_{\min} \left( G_\psi \right) \right\}^{-1} \tag{27}
$$

$$
= \min \left\{ w_{\min} \left( \bar{G}_\xi \right), \lambda_{\min} \left( G_\psi \right) \right\}^{-1}
$$

$$
\leq \min \left\{ (1 - \gamma) \lambda_{\min} \left( \hat{G}_\xi \right), \lambda_{\min} \left( G_\psi \right) \right\}^{-1} \tag{28}
$$

$$
\leq \bar{w}^{-1}/2, \tag{29}
$$

where (25) is the spectral norm equality; (26) uses the Fan-Hoffman inequality (Bhatia, 2013) and the fact that $\mathrm{sym}(\bar{G}_\xi \oplus G_\psi)$ is positive definite, where $\mathrm{sym}(A) = (A + A^T)/2$; (27) uses the fact that $\lambda(A \oplus B) = \lambda(A) \cup \lambda(B)$, where $\lambda(A)$ is the set of eigenvalues of $A$; (28) is by (11); and (29) is by the definition of $\bar{w}$.

Therefore, we have

$$
\|\Lambda(\xi_1) - \Lambda(\xi_2)\| \leq \left\| \left( \bar{G}_{\xi_1} \oplus G_\psi \right)^{-1} \left( \bar{\phi}_{\xi_1} \oplus \bar{\psi}_{\xi_1} \right) - \left( \bar{G}_{\xi_2} \oplus G_\psi \right)^{-1} \left( \bar{\phi}_{\xi_2} \oplus \bar{\psi}_{\xi_2} \right) \right\|
$$

$$
\leq \underbrace{ \left\| \left( \bar{G}_{\xi_1} \oplus G_\psi \right)^{-1} \right\|_{\mathrm{op}} \left\| \left( \bar{\phi}_{\xi_1} \oplus \bar{\psi}_{\xi_1} \right) - \left( \bar{\phi}_{\xi_2} \oplus \bar{\psi}_{\xi_2} \right) \right\| }_{U_1}
$$

$$
+ \underbrace{ \left\| \left( \bar{G}_{\xi_1} \oplus G_\psi \right)^{-1} - \left( \bar{G}_{\xi_2} \oplus G_\psi \right)^{-1} \right\|_{\mathrm{op}} \left\| \bar{\phi}_{\xi_2} \oplus \bar{\psi}_{\xi_2} \right\| }_{U_2}.
$$

For $U_1$, by Lemma 6 and (29), we have

$$
U_1 \leq \bar{w}^{-1} \left( \sigma L \sqrt{\max\{d_1, d_2\}}(R + F) + L_r \right) \|\xi_1 - \xi_2\|/2.
$$

For $U_2$, we have

$$
U_2 \leq \left\| \left( \bar{G}_{\xi_1} \oplus G_\psi \right)^{-1} \right\|_{\mathrm{op}} \left\| \left( \bar{G}_{\xi_1} \oplus G_\psi \right) - \left( \bar{G}_{\xi_2} \oplus G_\psi \right) \right\|_{\mathrm{op}} \left\| \left( \bar{G}_{\xi_2} \oplus G_\psi \right)^{-1} \right\|_{\mathrm{op}} \left\| \bar{\phi}_{\xi_2} \oplus \bar{\psi}_{\xi_2} \right\|
$$

$$
\leq \bar{w}^{-2} \sigma L (1 + \gamma) \sqrt{\max\{d_1, d_2\}}(R + F)\|\xi_1 - \xi_2\|/4,
$$

where the last inequality uses (29) and Lemma 6. Plugging $U_1$ and $U_2$ back gives

$$
\|\Lambda(\xi_1) - \Lambda(\xi_2)\| \leq L_w \|\xi_1 - \xi_2\|,
$$

where

$$
L_w = \bar{w}^{-1} \left( \bar{w}^{-1} \sigma L \sqrt{\max\{d_1, d_2\}}(R + F) + L_r/2 \right), \tag{30}
$$

and thus Zhang et al. (2023, Assumption 3.5 (v)) is satisfied.

By the definition of $\Lambda$ and (29), (Zhang et al., 2023, Assumption 3.5 (vi)) is satisfied:

$$
\sup_{\xi \in \Xi} \|\Lambda(\xi)\| \leq \bar{w}^{-1}(R + F)/2.
$$

Zhang et al. (2023, Assumption 3.5 (vii)) is satisfied by Assumption 1 and Lemma 4.

Let $D > \max \left\{ \bar{w}^{-1}/2, \|\xi_0\| \right\}$. Zhang et al. (2023, Assumption 3.6) is satisfied.

Finally, let $\alpha_t = 4/(1 + \bar{w}t)$. Zhang et al. (2023, Assumption 3.7) is satisfied. Note that we do not need the parameter $t_0$ in Zhang et al. (2023, Assumption 3.7) because we use asymptotic notations.

*Proof of Theorem 3.* By Zhang et al. (2023, Corollary 3.10), we have

$$
\mathbb{E}\left[ \|\xi_T - \Lambda(\xi_T)\|^2 \right] = O\left( \frac{L_w^2 D^2}{\bar{w}^2} + \frac{\log^2 T}{T} \right).
$$

By Assumption 5, we get

$$L_w \leq \bar{w}^{-1}(\bar{w}/4 + \bar{w}/4) \leq 1/2.$$

Therefore, $\Lambda$ is a contraction mapping. Suppose there exists another MFE $\xi_{**}$. We have

$$\|\xi_* - \xi_{**}\| = \|\Lambda(\xi_*) - \Lambda(\xi_{**})\| \leq L_w \|\xi_* - \xi_{**}\| \leq 1/2 \|\xi_* - \xi_{**}\|,$$

which implies $\xi_* = \xi_{**}$. We also get

$$
\begin{aligned}
\mathbb{E}\left[\|\xi_T - \xi_*\|\right] &\leq \mathbb{E}\left[\|\xi_T - \Lambda(\xi_T)\|\right] + \mathbb{E}\left[\|\Lambda(\xi_T) - \xi_*\|\right] \\
&= \mathbb{E}\left[\|\xi_T - \Lambda(\xi_T)\|\right] + \mathbb{E}\left[\|\Lambda(\xi_T) - \Lambda(\xi_*)\|\right] \\
&\leq \mathbb{E}\left[\|\xi_T - \Lambda(\xi_T)\|\right] + L_w \mathbb{E}\left[\|\xi - \xi_*\|\right],
\end{aligned}
$$

which gives

$$\mathbb{E}\left[\|\xi_T - \xi_*\|\right] \leq \frac{1}{1 - L_w} \mathbb{E}\left[\|\xi_T - \Lambda(\xi_T)\|\right] = O\left(\frac{D(L_r + \bar{w}^{-1}\bar{L}(R + F))}{\bar{w}^2} + \frac{\log T}{\sqrt{T}}\right),$$

where $\bar{L} = \sigma L \sqrt{\max\{d_1, d_2\}}$. $\qquad\qquad\square$

## I.2 CONVERGENCE REGION

The radius of the convergence region in Theorem 3 needs to be further inspected to make the bound informative. We compare it with two constants: the projection radius $D$ and the magnitude of the optimal policy parameter $\|\xi_*\|$.

A desirable property of the convergence region is that it can be much smaller than the projection radius $D$. This property is not enjoyed by some algorithms with linear function approximation. For instance, linear Q-learning with asymptotically visits every part of the projection ball. To achieve this property, we need

$$\frac{L_r + \bar{w}^{-1}\bar{L}(R + F)}{\bar{w}^2} \lesssim 1,$$

which is equivalent to

$$L_r \lesssim \bar{w}^2 \text{ and } R + F \lesssim \bar{w}^3. \tag{31}$$

Recall from Assumption 5 that

$$L_r \lesssim \bar{w} \text{ and } R + F \lesssim \bar{w}^2.$$

Therefore, to make the convergence region radius smaller than the projection radius, we need to further scale down the bounds on $L_r$, $R$, and $F$ by $\bar{w}$.

We now discuss the condition for the convergence region radius to be smaller than $\|\xi_*\|$, ensuring that the algorithm finds a better parameter than $\xi = 0$. Note that the only requirement on the projection radius $D$ is that it should be no smaller than the magnitude of the optimal policy parameter $\theta_{-*}$, i.e., $D \geq \|\theta\|$. We first consider the case where we have a good estimate of $\theta$. Let $D \asymp \|\xi\|$. Then, one can see that (31) ensures that the convergence region radius is smaller than $\|\xi_*\|$.

Next, we consider a general $D$. Denote $c := (L_r + \bar{L}(R + F)\bar{w}^{-1})/\bar{w}^2$, which is independent of $D$. Theorem 3 implies that, in expectation, $\xi_T$ will be bounded by $\|\xi_T\| \lesssim \|\xi_*\| + cD$. Thus, the projection will be inactive if $D \gtrsim \|\xi_*\| + cD$. Let $D' \asymp \|\xi_*\|/(1 - c)$. We know that running the algorithm with projection radius $D$ or $D'$ is equivalent for large $T$, as the projection will be inactive in both cases. Consequently, we only need $\|\xi_*\|$ to be smaller than the convergence region radius if $D = D' \asymp \|\xi_*\|/(1 - c)$, which is equivalent to

$$\frac{\|\xi_*\|}{1 - c} \cdot c \lesssim \|\xi_*\| \impliedby L_r \lesssim \bar{w}^2/4 \text{ and } R + F \lesssim \bar{w}^3/4.$$

Therefore, we conclude that the convergence region in Theorem 3 is especially informative if (31) holds.

## J  APPROXIMATION ERROR ANALYSIS

This section aims to characterize the approximation error of SemiSGD for general (non-linear) MFGs. Recall that SemiSGD converges to the projected MFE (Appendix F). Let $\xi_\diamond = (\theta_\diamond; \eta_\diamond)$ be the convergence point of SemiSGD with PA-LFA. Let $(q_*, \mu_*)$ be the actual MFE. The approximation error is defined as

$$\epsilon_q := \|q_* - \langle \phi, \theta_\diamond \rangle\|_\infty, \quad \epsilon_\mu := \|\mu_* - \langle \psi, \eta_\diamond \rangle\|_{\mathrm{TV}}. \tag{32}$$

Additionally, we define the inherent error of the chosen basis as

$$\epsilon_\phi := \|q_* - \Pi_\phi q_*\|_\infty, \quad \epsilon_\psi := \|\mathcal{P}_* - \Pi_\psi \mathcal{P}_*\|_{\mathrm{TV}}.$$

The next two lemmas bound the approximation errors in (32) separately, showing how they are correlated. Theorem 2 is a direct corollary of Lemmas 13 and 14 under the small Lipschitz constants assumption.

**Lemma 13** (Value function approximation error).

$$\epsilon_q \leq \left( \gamma + \frac{\gamma \sigma R L_\pi}{1 - \gamma} \right) \epsilon_q + \left( L_r + \frac{\gamma \sigma R L_P}{(1 - \gamma)\sqrt{d_2}} \right) \epsilon_\mu + \epsilon_\phi.$$

*Proof.* We first have the decomposition:

$$\epsilon_q \leq \| \langle \phi, \theta_\diamond \rangle - \Pi_\phi q_* \|_\infty + \|\Pi_\phi q_* - q_*\|_\infty, \tag{33}$$

where $\Pi_\phi$ is the orthogonal projection operator onto the linear span of basis $\phi$ w.r.t. the inner product induced by the $\xi_\diamond$. Since $\xi_\diamond$ is a projected MFE, by Definition 3, we get

$$\| \langle \phi, \theta_\diamond \rangle - \Pi_\phi q_* \| = \| \Pi_\phi \mathcal{T}_\diamond \langle \phi, \theta_\diamond \rangle - \Pi_\phi q_* \| \leq \| \mathcal{T}_\diamond \langle \phi, \theta_\diamond \rangle - q_* \|, \tag{34}$$

where $\mathcal{T}_\diamond := \mathcal{T}_{\xi_\diamond}$ and the inequality uses the fact that $\Pi_\phi$ is a non-expansive operator. Since $q_*$ is an equilibrium value function, we have

$$\| \mathcal{T}_\diamond \langle \phi, \theta_\diamond \rangle - q_* \|_\infty = \| \mathcal{T}_\diamond \langle \phi, \theta_\diamond \rangle - \mathcal{T}_* q_* \|_\infty \leq \| \mathcal{T}_\diamond (\langle \phi, \theta_\diamond \rangle - q_*) \|_\infty + \| (\mathcal{T}_\diamond - \mathcal{T}_*) q_* \|_\infty. \tag{35}$$

For the first term in (35), the Bellman operator's definition gives

$$\| \mathcal{T}_\diamond (\langle \phi, \theta_\diamond \rangle - q_*) \|_\infty \leq \gamma \| \langle \phi, \theta_\diamond \rangle - q_* \|_\infty = \gamma \epsilon_q. \tag{36}$$

Similar to the Lipschitzness of TD operators (Lemma 6), the second term in (35) can be bounded as follows:

$$\| (\mathcal{T}_\diamond - \mathcal{T}_*) q_*(s, a) \|_\infty$$
$$= \left\| r(s, a, \langle \psi, \eta_\diamond \rangle) - r(s, a, \mu_*) + \int_{\mathcal{S} \times \mathcal{A}} \gamma q_*(s', a') \left( \mu_{\xi_\diamond}^\ddagger(s, a, s', a') - \mu_*^\ddagger(s, a, s', a') \right) \mathrm{d}s' \mathrm{d}a' \right\|_\infty$$
$$\leq L_r \| \langle \psi, \eta_\diamond \rangle - \mu_* \| + \gamma \|q_*\|_\infty \|\mu_{\xi_\diamond}^\ddagger - \mu_*^\ddagger\|_{\mathrm{TV}} \tag{37}$$
$$\leq L_r \epsilon_\mu + \gamma \|q_*\|_\infty \cdot \sigma \left( \frac{L_P}{\sqrt{d_2}} \epsilon_\mu + L_\pi \epsilon_q \right), \tag{38}$$

where (37) uses Assumption 1 and (38) uses Corollary 2. Since $q_*$ is the best response to $\mu_*$, we have

$$\|q_*\|_\infty = \left\| \mathbb{E}_{(q_*, \mu_*)} \sum_{t=0}^{\infty} \gamma^t r(s_t, a_t, \mu_*) \right\|_\infty \leq \sum_{t=0}^{\infty} \gamma^t R = \frac{R}{1 - \gamma}. \tag{39}$$

Plugging (39) back into (38) gives

$$\| (\mathcal{T}_\diamond - \mathcal{T}_*) q_* \|_\infty \leq L_r \epsilon_\mu + \frac{\gamma \sigma R}{1 - \gamma} \left( \frac{L_P}{\sqrt{d_2}} \epsilon_\mu + L_\pi \epsilon_q \right). \tag{40}$$

Plugging (36) and (40) back into (35) gives

$$\| \mathcal{T}_\diamond \langle \phi, \theta_\diamond \rangle - q_* \|_\infty \leq \left( \gamma + \frac{\gamma \sigma R L_\pi}{1 - \gamma} \right) \epsilon_q + \left( L_r + \frac{\gamma \sigma R L_P}{(1 - \gamma)\sqrt{d_2}} \right) \epsilon_\mu. \tag{41}$$

Plugging (41) back into (34) and then (33) gives the desired result. $\square$

**Lemma 14** (Population measure approximation error)**.**

*Proof.* For uniformly ergodic MDPs, $\mathcal{P}_*^\infty := \lim_{t\to\infty} \mathcal{P}_*^t$ exists, which maps any distribution to $\mu_*$. The uniform ergodicity is equivalent to strong ergodicity (Meyn & Tweedie, 2012), which implies following relation about the geometric convergence rate (Isaacson & Luecke, 1978):

$$\rho(\mathcal{P}_* - \mathcal{P}_*^\infty) \leq \rho < 1,$$

where $\rho(\mathcal{P})$ returns the spectral radius of $\mathcal{P}$. Without loss of generality, we assume $\rho > \rho(\mathcal{P}_* - \mathcal{P}_*^\infty)$. Then, by Isaacson & Luecke (1978, Corollary 3.9), for any $\rho > \rho(\mathcal{P}_* - \mathcal{P}_*^\infty)$, there exists $k \in \mathbb{N}$ such that

$$\|\mathcal{P}_*^k - \mathcal{P}_*^\infty\|_{\mathrm{TV}} \leq \rho^k,$$

where the norm is the operator norm induced by the total variation norm. Now we apply the decomposition:

$$\epsilon_\mu \leq \underbrace{\left\|\langle\psi, \eta_\diamond\rangle - \mathcal{P}_*^k \langle\psi, \eta_\diamond\rangle\right\|_{\mathrm{TV}}}_{E_1} + \underbrace{\left\|\mathcal{P}_*^k \langle\psi, \eta_\diamond\rangle - \mu_*\right\|_{\mathrm{TV}}}_{E_2}. \tag{42}$$

Since $\xi_\diamond$ is a projected MFE, by Definition 3, we have

$$E_1 = \left\|\left((\Pi_\psi \mathcal{P}_\diamond)^k - \mathcal{P}_*^k\right)\langle\psi, \eta_\diamond\rangle\right\|_{\mathrm{TV}} \leq \left\|(\Pi_\psi \mathcal{P}_\diamond)^k - \mathcal{P}_*^k\right\|_{\mathrm{TV}}.$$

A further decomposition gives

$$E_1 \leq \left\|\Pi_\psi \mathcal{P}_\diamond\left((\Pi_\psi \mathcal{P}_\diamond)^{k-1} - \mathcal{P}_*^{k-1}\right) + \Pi_\psi (\mathcal{P}_\diamond - \mathcal{P}_*) \mathcal{P}_*^{k-1} + (\Pi_\psi - \mathrm{Id})\mathcal{P}_*^k\right\|_{\mathrm{TV}}.$$

Note that both $\Pi_\psi$ and $\mathcal{P}$ are non-expansive operators. By the sub-multiplicativity of operator norms, we have

$$E_1 \leq \left\|(\Pi_\psi \mathcal{P}_\diamond)^{k-1} - \mathcal{P}_*^{k-1}\right\|_{\mathrm{TV}} + \underbrace{\|\mathcal{P}_\diamond - \mathcal{P}_*\|_{\mathrm{TV}}}_{E_3} + \underbrace{\|(\Pi_\psi - \mathrm{Id})\mathcal{P}_*\|_{\mathrm{TV}}}_{\epsilon_\psi}. \tag{43}$$

We denote $\epsilon_\psi := \|\mathcal{P}_* - \Pi_\psi \mathcal{P}_*\|_{\mathrm{TV}}$; $\epsilon_\psi$ is the inherent approximation error induced by $\psi$. For $E_3$, by definition, we have

$$E_3 = \sup_{\|\mu\|_{\mathrm{TV}} \leq 1} \left\|\int_{\mathcal{S}\times\mathcal{A}} (P(\cdot \,|\, s, a, \langle\psi, \eta_\diamond\rangle)\pi_{\theta_\diamond}(a \,|\, s) - P(\cdot \,|\, s, a, \mu_*)\pi_{q_*}(a \,|\, s))\,\mu(s)\mathrm{d}s\mathrm{d}a\right\|_{\mathrm{TV}}$$

$$\leq L_\pi \epsilon_q + \frac{L_P}{\sqrt{d_2}}\epsilon_\mu.$$

Plugging the above bound of $E_3$ back into (43) gives the following recursion:

$$E_1 \leq \left\|(\Pi_\psi \mathcal{P}_\diamond)^{k-1} - \mathcal{P}_*^{k-1}\right\|_{\mathrm{TV}} + L_\pi \epsilon_q + \frac{L_P}{\sqrt{d_2}}\epsilon_\mu + \epsilon_\psi$$

$$\leq k\left(L_\pi \epsilon_q + \frac{L_P}{\sqrt{d_2}}\epsilon_\mu + \epsilon_\psi\right). \tag{44}$$

For $E_2$, since $\mu_*$ is the equilibrium population measure, and $\mathcal{P}_*^\infty$ maps any distribution to $\mu_*$, we have

$$E_2 = \left\|\mathcal{P}_*^k \langle\psi, \eta_\diamond\rangle - \mu_* + \mu_* - \mu_*\right\|_{\mathrm{TV}}$$

$$= \left\|\mathcal{P}_*^k \langle\psi, \eta_\diamond\rangle - \mathcal{P}_*^k \mu_* + \mathcal{P}_*^\infty \langle\psi, \eta_\diamond\rangle - \mathcal{P}_*^\infty \mu_*\right\|_{\mathrm{TV}}$$

$$= \left\|(\mathcal{P}_*^k - \mathcal{P}_*^\infty)(\langle\psi, \eta_\diamond\rangle - \mu_*)\right\|_{\mathrm{TV}}$$

$$\leq \left\|\mathcal{P}_*^k - \mathcal{P}_*^\infty\right\|_{\mathrm{TV}} \epsilon_\mu$$

$$\leq \rho^k \epsilon_\mu. \tag{45}$$

Plugging (44) and (45) back into (42) gives

$$\epsilon_\mu \leq \left(\rho^k + \frac{kL_P}{\sqrt{d_2}}\right)\epsilon_\mu + kL_\pi \epsilon_q + k\epsilon_\psi.$$

Furthermore, if $\|\mathcal{P}_* - \mathcal{P}_*^\infty\|_{\mathrm{TV}} \leq \rho$, for example, $\mathcal{P}_*$ corresponds to a reversible Markov chain, then $k = 1$. $\qquad\square$

## K    MEAN FIELD GAMES WITH FINITE STATE-ACTION SPACE

Recall that SemiSGD with PA-LFA (Algorithm 1) reduces to tabular SemiSGD (Algorithm 1) for finite state-action spaces, when the feature map and measure map are chosen as

$$\phi(s, a) = e_{(s,a)} \in \mathbb{R}^{|\mathcal{S}||\mathcal{A}|}, \quad \psi(s') = e_{s'} \in \Delta^{|\mathcal{S}|}.$$

Then, $Q = \theta \in \mathbb{R}^{|\mathcal{S}||\mathcal{A}|}$ and $M = \eta \in \Delta^{|\mathcal{S}|}$ are the parameters themselves.

### K.1    IMPLICIT REGULARIZATION

We first show that tabular SemiSGD does not need the projection step for regularizing the parameters (see (6)). That is, tabular SemiSGD enjoys implicit regularization. For $\|M\|$, we first have $\|\psi(s')\|_1 \leq 1 =: F$. Recall the stochastic update rule of $M$:

$$M_{t+1} = M_t - \alpha \left( M_t - e_{s_{t+1}} \right) = (1 - \alpha)M_t + \alpha e_{s_{t+1}}.$$

Suppose $M_t \in \Delta^{|\mathcal{S}|}$ and the step-size is smaller than one. Then $M_{t+1} \geq 0$. Furthermore, we have

$$\|M_{t+1}\|_1 = \sum_{s \in \mathcal{S}} |(1-\alpha)M_t(s) + \alpha e_{s_{t+1}}(s)| = (1-\alpha)\sum_{s \in \mathcal{S}} M_t(s) + \alpha \sum_{s \in \mathcal{S}} e_{s_{t+1}}(s) = (1-\alpha) + \alpha = 1,$$

indicating that $M_{t+1} \in \Delta^{|\mathcal{S}|}$, without any projection.

For $\|Q\|$, notice that the true action-value function induced by any policy $\pi$ is bounded by $\|q_\pi\|_\infty \leq \sum_{t=0}^{\infty} \gamma^t R = R/(1-\gamma) =: D_\infty$ Suppose current estimated value function satisfies that $\|Q_t\|_\infty \leq D_\infty$, then we have

$$\begin{aligned}
|Q_{t+1}(s, a)| &= |Q_t(s_t, a_t) + \alpha \left( r_t + \gamma Q_t\left( s_{t+1}, a_{t+1} \right) - Q_t(s_t, a_t) \right)| \\
&= |(1-\alpha)Q_t(s_t, a_t) + \alpha\gamma Q_t\left( s_{t+1}, a_{t+1} \right) + \alpha r_t| \\
&\leq (1-\alpha)D_\infty + \alpha\gamma D_\infty + \alpha R \\
&= (1 - \alpha + \alpha\gamma)\frac{R}{1-\gamma} + \alpha R \\
&= \frac{R}{1-\gamma} = D_\infty.
\end{aligned}$$

Therefore, if the bound holds for the initial estimated value function, it holds for all sequential estimated value functions. Then, the following $\ell_2$ norm bound holds for all value functions:

$$\|Q\|_2 \leq \sqrt{|\mathcal{S}||\mathcal{A}|}\|Q\|_\infty \leq \frac{\sqrt{|\mathcal{S}||\mathcal{A}|}R}{1-\gamma} =: D$$

Consequently,

$$H = \frac{((1+\gamma)\sqrt{|\mathcal{S}||\mathcal{A}|} + 1 - \gamma)R}{1-\gamma} + 2 = O\left( \frac{\sqrt{|\mathcal{S}||\mathcal{A}|}R}{1-\gamma} \right).$$

### K.2    CONVERGENCE RATE

We now figure out the scale of the descent parameter $w$ for finite MFGs. First, for tabular SemiSGD, $G_\psi = I$. According to Lemma 9, $\lambda_{\min}(G_\psi) = 1$. Second, $\hat{G}_* = \text{diag}(\mu_*^\dagger(s, a))$, where $\mu_*^\dagger$ is the steady state-action distribution induced by $\xi_*$. Thus, $\lambda_{\min}(\hat{G}_*) = \min_{s,a} \mu_*^\dagger(s, a) \leq \frac{1}{|\mathcal{S}||\mathcal{A}|}$. We define $\lambda := \min_{s,a} \mu_*^\dagger(s, a) > 0$, the probability of visiting the least probable state-action pair in the MFE. Then, we have

$$w = \frac{1}{2}\min\{(1-\gamma)\lambda_{\min}(\hat{G}_{\xi_*}), \lambda_{\min}(G_\psi)\} \geq \frac{1}{2}(1-\gamma)\lambda.$$

We are now ready to state the sample complexity of tabular SemiSGD.

**Corollary 4.** *With either a constant step size $\alpha_t \equiv \alpha_0 = \log T/(wT)$ or a linearly decaying step size $\alpha_t = 1/(w(1+t))$, there exists a convex combination $\widetilde{\xi}_T$ of the iterates $\{\xi_t\}_{t=1}^T$ such that*

$$\mathbb{E}\left\|\widetilde{\xi}_T - \xi_*\right\|^2 = \widetilde{O}\left(\frac{|\mathcal{S}||\mathcal{A}|R^2}{\lambda^2(1-\gamma)^4 T}\right),$$

*where $\widetilde{O}$ suppresses logarithmic factors.*

Notably, tabular SemiSGD has the same sample complexity as TD Learning methods (Bhandari et al., 2018; Zou et al., 2019).

