# OpenReview forum: "Stochastic Semi-Gradient Descent for Learning Mean Field Games with Population-Aware Function Approximation"
_ICLR.cc/2025/Conference — ICLR 2025 Poster_

### Official Review · Reviewer_B4eq · 2024-10-21

**Soundness:** 2
**Presentation:** 3
**Contribution:** 3
**Rating:** 6
**Confidence:** 3

**Summary:**

This paper studies learning equilibrium policy in mean-field game. The authors propose semiSGD, a SGD style online algorithm, where they treat the policy and population as a unified parameter. For linear MFGs, under small Lipschitz factor case, they establish convergence analysis. Besides, when the Lipschitz factor of the problem is relative large, they show the algorithm converges to the neighborhood of Nash, whose radius scales with the Lipschitz factors.

**Strengths:**

It seems interesting to me the idea that considers the policy and population parameters as a unified parameter to optimize.

The assumptions are clearly stated, and some explanations are provided.

Numerical experiments are provided.

The comparison in Table 2 provides a good summary.

**Weaknesses:**

1. **About assumptions**: I think the authors should provide some justification about why it is reasonable to consider Assumption 2. In the more standard MFG setting, the equilibrium policy is defined to be the policy that no agent can deviate and increase its value, which correspond to the setting $\Gamma_\pi(Q)(\cdot|s) := \arg\max_a Q(s,a)$. However, Assumption 2 can not cover this setting with finite $L_\pi$, because of the discountinuity of the argmax operator.

    As implied by Assumption 2, the paper seems to consider a "smooth" version of standard Nash equilibrium. It is unclear that why this objective is of interests to study, and also what's the relationship between it and the standard Nash equilibrium.

2. **Comparison with Assumptions in previous work**: this paper consider contraction style assumptions to establish the convergence. Although in remark 2, the authors provide some discussion to justify the assumptions, it is still unclear to me how they compare with previous works. For example, does the contractivity assumption in [1] can be recovered as a special case by choosing some specific parameters in Assumption 2 and 4? Or maybe comparing with [2], which ensures contractivity by introducing the regularization, would Assumption 2 or 4 be satisfied by introducing large enough regularization?

3. **When Theorem 2 is meaningful?** Theorem 2 suggests the convergence with bounded bias. I think it is necessary to have some discussion about the magnitude of the bias term comparing with another algorithm randomly compute $\xi_T$. For instance, can you provide some examples when the bias term can be (much) lower than $\|\xi^*\|$, which suggests the algorithm is better than a random guess by directly assigning $\xi_T = 0$.

    It is also unclear how large $\bar{w}$ is. Given that $\bar{w}$ is a problem-dependent constant, can you explain when $\bar{w}$ can be large or small? That would be helpful to understand when Theorem 2 is meaningful and when it is vacuous.

[1] Guo et al., Learning mean-field games.

[2] Yardim et al., Policy mirror ascent for efficient and independent learning in mean field games

**Questions:**

1. In Assumption 5, $w$ is introduced to denote the upper bound of $\bar{w}$. However, it does not appears in Theorem 2. So what's the reason to mention such an upper bound?

---

> ### Author Response · Authors · 2024-11-19
> **Response to Reviewer B4eq**
>
> Dear Reviewer B4eq,
>
> Thank you for your detailed review and insightful questions. Regarding the weaknesses and questions, we provide the following detailed responses:
>
> > **W1**: The authors should provide some justification about why it is reasonable to consider Assumption 2.
>
> The reviewer is correct the argmax policy operator does not satisfy Assumption 2 and the assumption implies a "smooth" version of the standard Nash equilibrium.
> To address reviewer's concerns, we have added a new section (Appendix B.1) in the revised manuscript, providing a detailed discussion on the definitions of the mean field equilibrium (MFE) and policy operators considered in this work.
> We summarize the discussion below, justifying the feasibility of Assumption 2 and consistency of our definition of the MFE with the literature.
>
> To the best of our knowledge, all *regularized* policy operator in the literature are Lipschitz continuous and thus satisfy Assumption 2; these include
>
> - softmax/Boltzmann policy operator (Cui and Koeppl, 2021; Zaman et al., 2023);
> - softmin policy operator (Angiuli et al., 2023);
> - mirror descent policy operator (Perolet et al., 2021; Laurière et al. 2022b);
> - mirror ascent policy operator (Yardim et al., 2023).
>
> Additionally, works that apply regularization to the reward function rather than the policy operator (e.g., Xie et al. (2021) and Anahtarci et al. (2023)) also **imply** the Lipschitz continuity of the policy operator w.r.t. the population measure.
>
> Therefore, Assumption 2 can be satisfied by introducing regularization (not necessarily large). And with regularization, the "smooth" version of the MFE corresponds to the *regularized MFE*, which is widely studied in the literature. This is also discussed in Remark 2, and we have added the above examples of policy operators satisfying Assumption 2 to Remark 2 in the revised manuscript.
>
> We summarize some discussions in the paper and in the above references to answer why this objective, which encompasses regularized MFE, is of interests to study.
> First, when the regularization is small, the regularized MFE **approximates the standard Nash equilibrium** (Cui and Koeppl, 2021; Zaman et al., 2023).
> This is the case in Sections 5.1 and 7, where nearly no regularization is applied (see e.g., Equation (7)).
> The second reason why regularization is widely used in learning MFGs is that it serves as a stabilization mechanism, addressing the instability issue widely observed in learning MFGs with non-regularized policies (Cui and Koeppl, 2021; Laurière et al., 2022a).
>
> However, a central argument from the work is that, **without regularization**, SemiSGD can stably learn near-greedy policies (close to argmax) in the reward-maximization setting (standard MFE). As part of the endeavor, our numerical experiments compare SemiSGD **without regularization** against entropy-regularized methods to substantiate this claim.
>
> We hope these clarifications and the expanded discussion in Appendix B.1 address the reviewer’s concerns about our assumptions. We are happy to provide further clarifications if needed.

---

> ### Author Response · Authors · 2024-11-19
> **Response to Reviewer B4eq [Continued]**
>
> > **W2**: Comparison with Assumptions in previous work.
>
> Due to space constraints, we moved detailed discussions comparing our assumptions with those in prior works to Appendix A, particularly in the last paragraph. As noted by the reviewer, Table 2 in Appendix A provides a comprehensive comparison of our work with prior studies. The "Theoretical Properties" section of Table 2 highlights differences in assumptions, inspired by and extending the discussion in Yardim et al. (2023, Appendix C.1 and Table 2).
>
> As discussed in Appendix A and Yardim et al. (2023), Guo et al. (2019) uses a blanket contractivity assumption without examining the underlying MFG/MDP structure.
> Thus, we now provide a detailed comparison of our assumptions with those in Yardim et al. (2023), Anahtarci et al. (2023), and Angiuli et al. (2023), summarized in Table 1.
>
> Table 1. Comparison of Assumptions in different works.
>
> | Reference                   | Lip. const. of kernel | Lip. const. of reward | Regularization strength | Asmp. on combination/composition          | Remark         |
> | --------------------------- | --------------------- | --------------------- | ----------------------- | ----------------------------------------- | -------------- |
> | (Yardim et al., 2023, Assumptions 1 and 2)    | $K_\mu \lesssim 1$    | $L_\mu$               | $\rho^{-1}$             | $(K\_{\mu}+L\_{\mu}) \rho^{-1} \lesssim 1$  | Multiplicative |
> | (Anahtarci et al., 2023, Assumptions 1 and 2)    | $K_1\lesssim 1$       | $L_1$                 | $\rho^{-1}$             | $K\_{1}+(K\_{1}+L\_{1})\rho^{-1} \lesssim 1$ | Multiplicative |
> | (Angiuli et al., 2023, Assumptions 3.2, 3.6, and 3.7)    | $L_{p}\lesssim 1$       | $L_f$                 | $\phi$             | $(L_{p}+L_{f})\phi \lesssim 1$ | Multiplicative |
> | Our Assumptions 1, 2, and 4 | $L\_{P}$               | $L\_{r}$               | $L_\pi$                 | $L\_{P}+L\_{r}+L\_{\pi} \lesssim 1$          | Additive       |
>
> Several remarks on Table 1:
>
> 1. Yardim et al. (2023) and Angiuli et al. (2023) apply regularization to the policy update; Anahtarci et al. (2023) applies regularization to the reward function; and our Assumption 2 can be satisfied by applying regularization to the policy operator, e.g., using entropy regularization with a temperature $\rho = L\_{\pi}^{-1}$ (this is also discussed in our response to previous point);
> 2. All three works have an additional assumption requiring the Lipschitz constant of the kernel to be smaller than 1. Although we do not explicitly assume this, it is implicitly satisfied due to the additive nature of our assumptions.
>
> We now discuss the main difference between our assumptions and those in Yardim et al. (2023), Anahtarci et al. (2023), and Angiuli et al. (2023), i.e., the assumption on the combination/composition of Lipschitz constants.
> Using the notation in this work:
>
> - The other studies require the **multiplicative composition** of $(L_{P} + L_{r})$ and $L_{\pi}$ to be sufficiently small.
> - In contrast, our analysis behind Theorem 1 requires the **additive combination** of $L_{P}$, $L_{r}$, and $L_{\pi}$ to be sufficiently small.
>
> Aside from using different analysis frameworks, this difference potentially stems from the difference in the algorithm schemes:
>
> - Similar to FPI, the other studies iteratively calculate the policy and population updates in a forward-backward manner, resulting in *compositional* error dynamics, where the error scales after each inner loop or outer iteration.
> - In contrast, our algorithm updates the policy and population fully asynchronously in an SGD manner, resulting in *additive* error dynamics, where the noise at each step is from the combined contributions of policy update noise (controlled by $L\_{\pi}$) and population update noise (controlled by $L\_{P}$ and $L\_{r}$).
>
> We would be happy to include the above discussion in the revised manuscript if the reviewer finds it helpful.

---

> ### Author Response · Authors · 2024-11-19
> **Response to Reviewer B4eq [Continued]**
>
> > **W3**: When Theorem 2 is meaningful?
>
> This is a good question! The reviewer's reasoning is correct that if the bias is large, the bound in Theorem 2 becomes less informative.
> Below, we outline the conditions under which the bound in Theorem 2 is *informative*, i.e., when the following inequality holds:
> $$\tag{$\star$}
> \frac{D (L\_{r} + \bar{L}(R+F)\bar{w}^{-1} )}{\bar{w}^{2}} \lesssim \\|\xi_{\*}\\|
> .$$
> Note that the only requirement on the projection radius $D$ is that it should be no smaller than the magnitude of the optimal policy parameter $\theta\_{*}$, i.e., $D\ge \|\theta_{\*}\|$.
> We first consider the case where we have a good estimate of $\theta_{\*}$. Let $D \asymp  \|\xi_{\*}\|$. Then, $(\star)$ holds if
> $$
> L\_{r} \lesssim \bar{w}^{2} \text{ and } R+F \lesssim \bar{w}^{3}.
> $$
> Recall from Assumption 5 that:
> $$
> L\_{r} \lesssim \bar{w} \text{ and } R+F \lesssim \bar{w}^{2}.
> $$
> Therefore, for $(\star)$ to hold, bounds on $L\_{r}$, $R$, and $F$ need to be further scaled down by $\bar{w}$.
>
> Now, we consider a general $D$. Denote $c\coloneqq (L\_{r} + \bar{L}(R+F)\bar{w}^{-1}) / \bar{w}^{2}$, which is independent of $D$. Theorem 2 implies that, in expectation, $\xi\_{T}$ will be bounded by $\|\xi\_{T}\| \lesssim \|\xi_{\*}\| + cD$. Thus, the projection will be inactive if $D \gtrsim \|\xi_{\*}\| + cD$. Let $D' \asymp \|\xi_{\*}\|/ (1-c)$. We know that running the algorithm with projection radius $D$ or $D'$ is equivalent for large $T$, as the projection will be inactive in both cases. Consequently, the bound in Theorem 2 is informative if $(\star)$ holds with $D'$, which is equivalent to
> $$
>  \frac{c}{1-c} \lesssim 1 \impliedby L\_{r}\lesssim \bar{w}^{2} /4 \text{ and } R+F \lesssim \bar{w}^{3} /4.
> $$
> Therefore, we conclude that the bound in Theorem 2 is informative if $L\_{r}\lesssim \bar{w}^{2}$ and $R+F \lesssim \bar{w}^{3}$.
>
> Beyond condition $(\star)$, the above disccussion also suggests that the convergence region in Theorem 2 can be much smaller than the projection radius $D$. This property is particularly desirable for algorithms using linear function approximation. For instance, Q-learning with linear function approximation does not enjoy this property, as it asymptotically visits every part of the projection ball.
>
> We would be happy to include the above discussion in the revised manuscript if the reviewer finds it helpful.

---

> ### Author Response · Authors · 2024-11-19
> **Response to Reviewer B4eq [Continued]**
>
> > **W3+Q1**: Can you explain when $\bar{w}$ can be large or small? What's the reason to mention the upper bound $\overline{w}\le w$?
>
> We appreciate the reviewer's careful reading. $w$ and $\bar{w}$ are determined by the smallest eigenvalues of the Gram matrices of the feature map and basis measure.
> Specifically, consistent with Lemma 9 and Section I, we define
> $$
> w\_{\xi} \coloneqq \frac{1}{2} \min \left\\{ (1-\gamma)\lambda\_{\min}(\hat{G}\_{\xi}), \lambda\_{\min}(G\_{\psi}) \right\\}
> ,$$
> where $\hat{G}\_{\xi}$ is the Gram matrix of the feature map $\phi\_{\xi}$ under the steady distribution induced by $\xi$ and $G\_{\psi}$ is the Gram matrix of the basis measure $\psi$.
> $w\_{\xi}$ corresponds to the convergence rate when using iid data from the steady distribution associated with $\xi$.
> Then, we have $w = w\_{\xi\_{*}}$ (Lemma 9) and $\bar{w} = \inf\_{\xi\in\Xi}w\_{\xi}$ (Section I), and naturally $\bar{w}\le w$.
>
> From the definition, we can see that $w\_{\xi}$ is influenced by the orthogonality and coverage of the feature map and basis measure:
>
> - $w_{\xi}$ is large if the feature map and measure basis are *well-designed*, with components that are (nearly) orthogonal;
> - $w\_{\xi}$ can be small if the feature map and measure basis have components with a large degree of linear dependence.
>
> $w\_{\xi}$ also depends on the steady distribution induced by $\xi$, as the inner product in the Gram matrix is taken with respect to the steady distribution.Specifically, $w\_{\xi}$ can be small if some state-action pairs are visited with a small probability under the steady distribution $\mu\_{\xi}^\dagger$.
> Consequently, $\bar{w}$ can be much smaller than $w$ if certain parameters in $\Xi$ induce steady distributions with poor coverage of the state-action space.
>
> In essence, $w\_{\xi}$ reflects the *ease of exploring the state-action space*. Large $w\_{\xi}$ occurs when all state-action pairs are visited with reasonable probability and the observed features are diverse; and $w\_{\xi}$ occurs when certain state-action pairs are rarely visited or the features are redundant.

---

> > ### Comment · Reviewer_B4eq · 2024-11-19
> >
> > Thanks for the detailed response. My questions are addressed and I rasied my score. I would suggest to integrate all the above explanations into the paper revision, since it makes the arguments in the paper more clear.

---

> > > ### Author Response · Authors · 2024-11-22
> > >
> > > Thank you so much for your timely response and raising your score! We are glad that our explanations were helpful. We have updated our revised manuscript to include the discussions as you suggested.

---

### Official Review · Reviewer_b6VH · 2024-11-01

**Soundness:** 3
**Presentation:** 3
**Contribution:** 3
**Rating:** 8
**Confidence:** 3

**Summary:**

The manuscript goes beyond existing fixed-point iteration (FPI) methods for solving mean field games, by treating the mean field and agent policy jointly instead of performing forward and backward updates. Here, FPI-type methods are understood as those computing the full forward and backward equations for mean field and optimal policies. The advantage is allowing asynchronous updates to both, which can lead to improved results. An analysis of the resulting semi-gradient descent method is performed under linear models and varying sets of assumptions. Empirically, the designed algorithms are further demonstrated on a variety of problems.

**Strengths:**

- The paper is well written and relatively clear despite the subject matter.
- The analysis is novel, interesting and extensive, e.g., the combination of linear approximations with MFGs and producing finite-time error bounds, or the convergence results without (directly) assuming contractivity or monotonicity.
- Empirically, the approach appears to be able to outperform existing approaches in some of the demonstrated problems in terms of exploitability. Further, the method is practical, as it is applicable in an online manner to unknown models.
- The introduction of synchronous updates as opposed to forward-backward computations seems somewhat significant to the study of MFG learning algorithms, as it allows for improved sample complexity and better empirical results.

**Weaknesses:**

- The approach seems to be limited to stationary mean field games, i.e. ones with time-stationary mean fields. This does not match with the compared / referenced literature, of which most are for non-stationary cases.
- For the theoretical results, there remain limits in terms of significance, as also discussed by the authors. The significance of theoretical results is limited due to the requirement of strong assumptions such as linear models and regularized solutions.
- Some minor issues (incomplete / TeX errors) in references.
- Some points remained unclear to me, see questions below.

**Questions:**

- Uniqueness of solutions seems to not be required. How it is possible that we obtain convergence in the presence of non-unique MFE? Does the theory imply convergence to multiple MFE, or is uniqueness implicitly assumed?
- What is the reasoning behind replacing $M_*$ (the non-unique? unknown MFE) by bootstrap estimates under the current policy (before Eq. (3))? Why can the current policy produce estimates for the mean field $M_*$ of the MFE?
- The methodology does not directly optimize exploitability, what is the difficulty in instead using "true" gradients on the unified parameters to minimize exploitability?
- Can you extend similar techniques to non-time-stationary MFGs?

---

> ### Author Response · Authors · 2024-11-19
> **Response to Reviewer b6VH**
>
> Dear Reviewer b6VH,
>
> We sincerely appreciate your detailed and positive feedback and insightful questions. Below, we provide responses to your questions and address the points raised in your review.
>
> > **W1+Q4**: The approach seems to be limited to stationary mean field games. Can you extend similar techniques to non-time-stationary MFGs?
>
> As a simple SGD-type algorithm, our method is not limited to stationary MFGs and is readily applicable to non-stationary MFGs. We focus on stationary MFGs for simplicity and to highlight the core ideas. The theoretical analysis for non-stationary MFGs is an interesting avenue for future work.
>
> While non-stationary MFGs indeed are an important class of MFGs encompassing time-dependent policies and distributions, we respectfully disagree with the comment that most of the referenced literature focuses on non-stationary MFGs. To clarify, among the papers cited in Table 2, **9 out of 14** consider stationary MFGs (Guo et al., 2019; Xie et al., 2021; Mao et al., 2022; Angiuli et al., 2022, 2023; Anahtarci et al., 2023; Zaman et al., 2023; Yardim et al., 2023; Zhang et al., 2024a).
>
> To address the reviewer's question more directly, we now demonstrate how our method can be extended to non-stationary MFGs with minor modifications.
> Consider a finite-horizon MDP $(\mu\_{1},\mathcal{S},\mathcal{A},\mathcal{H},\\{ P\_{h} \\}\_{h=1}^{\mathcal{H}}, \\{ r\_{h} \\}\_{h=1}^{\mathcal{H}} )$, where $\mu\_{1}$ is the initial population measure, $\mathcal{H}$ is the time horizon, and $P\_{h}$ and $r\_{h}$ are the transition kernel and reward function at time $t$. Our goal is to learn the time-dependent equilibrium policies $\\{\pi^\* _{h}\\}\_{h\in \mathcal{H}}$ and population measures $\left\\{ \mu^\* _{h} \right\\}\_{h\in \mathcal{H}}$.
> Thus, instead of maintaining a vector parameter $\xi = (\eta;\theta)\in\mathbb{R}^{d\_1+d\_2}$ for estimating the value function and population measure as in the infinite-horizon case, we maintain a sequence of vectors $\left\\{ \xi\_{h} = (\eta\_{h};\theta\_{h}) \right\\}\_{h\in \mathcal{H}}$.
> Additionally, this framework can also accommodate time-dependent feature maps $\left\\{ \phi\_{h} \right\\}\_{h\in \mathcal{H}}$ and basis measures $\left\\{ \psi\_{h} \right\\}\_{h\in \mathcal{H}}$.
> The resulting algorithm is presented below:
>
> 1. input: initial parameters $\xi^{(0)} = \\{ \xi\_{h}^{(0)}   \\}\_{h\in \mathcal{H}}$.
> 2. for $t = 0,1,\dots,T$ do
>     1. sample initial state $s\_{1} \sim \mu\_{1}$.
>     2. for $h=1,\dots, \mathcal{H}$ do
>         1. observe $a_h \sim \Gamma\_{\pi}(\theta^{(t)} _{h})[s_h]$, $r\_{h}=r\_{h}(s_h,a_h,\eta_h^{(t)})$, $s\_{h+1} \sim P\_{h}(\cdot \mid s\_{h},a\_{h}, \eta\_{h}^{(t)})$, and $a\_{h+1} \sim \Gamma\_{\pi}(\theta\_{h+1}^{(t)})[s\_{h+1}]$.
>         2. update the parameter $\theta\_{h}^{(t+1)} = \Pi (\theta\_{h}^{(t)} - \alpha _{t} \mathfrak{g}\_{h} (\theta^{(t)}) )$ and $\eta\_{h+1}^{(t+1)} = \Pi (\eta\_{h+1}^{(t)} - \alpha _{t} \mathfrak{g}\_{h} (\eta^{(t)}) )$.
> 3. return $\xi^{(T)}$.
>
> The semi-gradient $\mathfrak{g}\_{h}$ in this case is
> $$
> \begin{cases}
>   \mathfrak{g}\_{h}(\theta^{(t)}) =& \phi\_{h}(s\_{h},a\_{h}) \left( \phi\_{h}(s\_{h},a\_{h})\theta\_{h}^{(t)} - \phi\_{h+1}(s\_{h+1},a\_{h+1}) \theta\_{h+1}^{(t)} - r\_{h}  \right),\\\\
>   \mathfrak{g}\_{h}(\eta^{(t)}) =& \int\_{\mathcal{S}}\psi\_{h+1}(s)\psi\_{h+1}(s)^{T} \eta\_{h+1}^{(t)}\mathrm{d} s -  \int\_{\mathcal{S}} \psi\_{h+1}(s) \delta\_{s\_{h+1}} (s)\mathrm{d} s
> .\end{cases}
> $$

---

> ### Author Response · Authors · 2024-11-19
> **Response to Reviewer b6VH [Continued]**
>
> > **W2**: The significance of theoretical results is limited due to the requirement of strong assumptions such as linear models and regularized solutions.
>
> We appreciate the reviewer's acknowledgment of our discussion on the limitations of our theoretical results.
> However, we would like to respectfully point out that a significant portion of our work is dedicated to addressing these limitations, and the assumptions of linear models and regularized solutions are **not required** in many sections. Specifically:
>
> - Section 5.1 removes the regularization condition, allowing the method to learn **non-regularized solutions**.
> - Section 6 extends the analysis to **non-linear models**, characterizing the approximation error introduced by linear function approximation.
>
> In summary, our method is flexible and practical, as it does not rely on strong assumptions such as linear models or regularized solutions, as evidenced by our numerical experiments. Additionally, we provide extensive theoretical results that cover both non-linear models and non-regularized MFGs.

---

> ### Author Response · Authors · 2024-11-19
> **Response to Reviewer b6VH [Continued]**
>
> > **W3**: Minor issues (incomplete / TeX errors) in references.
>
> Thank you for the careful reading! The issues in references have been fixed in the revised version. Please let us know if you find any other issues.

---

> ### Author Response · Authors · 2024-11-19
> **Response to Reviewer b6VH [Continued]**
>
> > **Q1**: Uniqueness of solutions seems to not be required. How it is possible that we obtain convergence in the presence of non-unique MFE? Does the theory imply convergence to multiple MFE, or is uniqueness implicitly assumed?
>
> This is a good question! The uniqueness of the MFE is implied by Assumption 4 or 5.
> Lemma 9 shows how Assumption 4 leads to the global contractivity of the MFE, which in turn ensures its uniqueness. This aligns with previous work on contractive MFGs, where the uniqueness of the MFE is established.
>
> In the revised manuscript, we have added the discussion on how Assumption 5 implies the uniqueness of the MFE in Theorem 2 and Section I.
> In short, let $\Lambda$ be the operator that maps a parameter $\xi$ to the learned parameter based on iid data from the steady distribution w.r.t. $\xi$. By the definition of the MFE, we have $\Lambda(\xi\_{\*}) = \xi\_{\*}$ for any MFE. Assumption 5 implies that $\Lambda$ is a contraction operator, and thus the MFE is unique.

---

> ### Author Response · Authors · 2024-11-19
> **Response to Reviewer b6VH [Continued]**
>
> > **Q2**: What is the reasoning behind replacing $M\_\*$ by bootstrap estimates under the current policy (before Eq. (3))? Why can the current policy produce estimates for the mean field $M\_\*$ of the MFE?
>
> This is also good question! Developing the rule for online population measure update with linear function approximation is a main novelty of this work, and it can be derived either by mirroring the **TD learning** or generalizing the **MC sampling**, highlighting its well-motivated and rigorously justified nature.
>
> Intuitively, due to the contraction property of the transition operator, the empirical population measure $\delta\_{s'}$ produced by the current policy estimates of the steady population measure $\mu\_{\pi}$ induced by the current policy; and the current policy $\pi$ is driven toward fixed-point policy $\pi\_{*}$ due to the contraction property of the Bellman operator. Together, $\delta\_{s'}$ points toward a *good direction* (or descent direction, as discussed in Section 2.2) approaching the MFE.
> In analogy to a two-timescale scheme, the population estimate follows the policy estimate, which converges to the equilibrium policy, thereby driving the population estimate toward the equilibrium population measure.
>
> We now recall the derivation of TD learning [Tsitsiklis and Van Roy, 1997].
> Define the loss function $\mathcal{L}\coloneqq \frac{1}{2}\\|Q-Q_*\\|\_{\mu_{\*}}^{2}$, whose gradient is
> $$
> \nabla\_{\theta} \mathcal{L} = \left<\nabla\_{\theta}Q, Q-Q_\*  \right>\_{\mu\_\*}
> = \left<\phi, Q-\mathcal{T}Q_{\*}  \right>\_{\mu_{\*}}
> ,$$
> where the inner product and norm are induced by the $L_2$ inner product with state-action distribution $\mu_{\*}^\dagger$. Replacing the target $Q_{\*}$ with the current value function and using stochastic estimates of the expectation, we obtain the TD update rule
> $$
> 	\mathfrak{g}(\theta; O) = \phi(s,a) \left(\left< \phi(s,a) - \gamma\phi(s',a'), \theta \right> - r \right)
> 	\eqqcolon G\_{\phi}(O)\theta - \varphi(O).
> $$
> TD(0) works because of the positive definiteness of $G\_{\phi}$.
> Mirroring the above derivation, we arrive at the population update rule presented in Section 4. Similarly, our method relies on the positive definiteness of $G\_{\psi}$, the Gram matrix of the basis measure.
>
> Next, we recall MCMC sampling for estimating the distribution:
> $$
> \eta\_{t+1} = \frac{1}{t+1} \sum\_{l=1}^{t+1}\delta\_{s\_{l}}
> = \left(1 - \frac{1}{t+1}\right)\eta\_{t} + \frac{1}{t+1}\delta\_{s\_{t+1}}.
> $$
> Note that we also use a step size of $\alpha\_{t}\asymp 1 /t$. Thus, our population measure update generalizes the MCMC sampling to the setting of linear function approximation. As discussed in Section 2.2, our method is a stochastic approximation method on a non-stationary Markov chain, making it natural to treat MCMC sampling as a heuristic for the population measure update. Additional assumptions are necessary to ensure that this non-stationary Markov chain mixes to the MFE.

---

> ### Author Response · Authors · 2024-11-19
> **Response to Reviewer b6VH [Continued]**
>
> > **Q3**: The methodology does not directly optimize exploitability, what is the difficulty in instead using "true" gradients on the unified parameters to minimize exploitability?
>
> We thank the reviewer for this thought-provoking question!
> We assume the reviewer is referring to the true gradient of the exploitability function.
> Recall the definition and analysis of exploitability in Perrin et al. (2020), the true gradient of the exploitability function involves the best response policy w.r.t. the current population, which typically requires an additional iterative process to calculate/estimate.
> This is also the case for gradient-based temporal difference learning (Sutton et al., 2009): to obtain an unbiased estimate of the true gradient w.r.t. the loss $\mathcal{L}$ defined above, one would need to solve another fixed-point equation, resulting in a multi-loop/timescale algorithm.
> Since one of the main motivations of our work is to avoid such forward-backward or multi-loop processes, we consider obtaining an unbiased estimate of the true gradient of the exploitability function as the main difficulty.
>
> However, the reviewer’s question points to an interesting direction: can we develop a (Semi)SGD-type method for directly optimizing the exploitability function? Here is a tentative roadmap for this approach inspired by the development of SemiSGD in this work:
>
> 1. Introduce an auxiliary policy parameter $\theta^{\mathrm{BR}}$ to keep track of the best response policies.
> 2. At each iteration, update the objective policy parameter $\theta$ using the semi-gradient of the exploitability function, which is calculated using the current best response policy parameter $\theta^{\mathrm{BR}}$.
> 3. Within the same iteration, update the population measure accordingly (e.g., by MCMC), and then update the best response parameter $\theta^{\mathrm{BR}}$ accordingly (e.g., by Q-learning).
>
> We would be happy to discuss this direction further if the reviewer is interested.
>
> [Sutton et al., 2009]: Richard S Sutton, Hamid Reza Maei, Doina Precup, Shalabh Bhatnagar, David Silver, Csaba Szepesvári, and Eric Wiewiora. Fast gradient-descent methods for temporal-difference learning with linear function approximation.

---

> ### Author Response · Authors · 2024-11-22
> **Follow up**
>
> Thank you again for your valuable time in reviewing our work. We hope our responses have been helpful. If you have any further questions, we would be happy to provide more discussions!

---

> > ### Comment · Reviewer_b6VH · 2024-11-25
> >
> > Thank you for the detailed response! My questions have been addressed. Given the substantial additional discussions and hopefully future incorporation thereof in the manuscript, I have raised my score.

---

> > > ### Author Response · Authors · 2024-11-27
> > > **Thank you!**
> > >
> > > We thank the reviewer once again for their detailed review, valuable suggestions, insightful questions, and thoughtful appreciation of the work.

---

### Official Review · Reviewer_VqQ9 · 2024-11-04

**Soundness:** 3
**Presentation:** 3
**Contribution:** 3
**Rating:** 6
**Confidence:** 4

**Summary:**

In this paper, a numerical method for mean field games is proposed. Mean field games games approximate Nash equilibria for games with many players. Here, the proposed method relies on stochastic semi-gradient updates. Theoretical convergence is proved and numerical illustrations are also given.

**Strengths:**

The paper is relatively clear and the algorithm is new, to the best of my knowledge. The convergence analysis includes sample complexity. The experiments cover different examples and contain a few baselines.

**Weaknesses:**

I understand that empirical convergence is observed beyond the assumptions, but still: it is not clear to me how to check these assumptions in practice.
In the numerical examples, the baselines could be explained more clearly.

**Questions:**

Q1: Theorem 2: Please provide an example satisfying the assumptions of this theorem.

Q2: Section 7: Can you please explain what is FPI+MD? Is it the same as the online mirror descent of (Perolat et al., 2021)? If not, please compare with this algorithm, which is known to empirically converge much faster than fictitious play-type iterations.

---

> ### Author Response · Authors · 2024-11-19
> **Response to Reviewer VqQ9**
>
> Dear Reviewer VqQ9,
>
> We sincerely appreciate your positive feedback and the opportunity to clarify some points. Below, we provide detailed responses to your questions.
>
> > **W1**: How to check these assumptions in practice?
>
> Thank you for raising this important question. For a concrete example that satisfies the assumptions, please refer to our response to Q1. Here, we outline how these assumptions can be verified in practical settings.
>
> Assumptions 1-3 are standard regularity conditions that are satisfied by many MFGs:
>
> 1. Assumption 1 requires the underlying MDP to be Lipschitz continuous in the population measure, meaning that small changes in the population measure should result in small, controlled changes in the transition dynamics and reward function. For example, MFGs with population-independent transitions satisfy this condition.
> 2. Assumption 2 requires the policy operator to be Lipschitz continuous in the value function, meaning that small changes in the value function should not change the policy drastically.
> For instance, softmax function satisfies this assumption.
> 3. Assumption 3 requires that the state-action space can be easily explored by the agent for any policy and population.
> As mentioned in the paper, this is a standard assumption for online learning algorithms and holds for many ergodic MDPs.
>
> Assumptions 4 and 5 are structural assumptions enabling theoretical convergence guarantees. They are generally more challenging to verify in practice but can still be interpreted and satisfied under specific conditions.
>
> 4. Assumption 4 requires the Lipschitz constants of the transition dynamics, reward function, and policy operator to be sufficiently small, ensuring smooth changes throughout the learning process.
>     - Evaluating $w$. $w$ is a constant determined by the underlying MDP and feature maps used in the function approximation, which may not be known a priori. However, for MFGs with a finite state-action space, as calculated in Section K.2, we have a lower bound $w>(1-\gamma)\lambda /2$, where $\lambda$ is the probability of visiting the least probable state-action pair in the MFE. Generally, $w$ reflects the *ease of exploring the state-action space*, which is large when all state-action pairs are visited with reasonable probability and the feature maps are close to orthogonal.
>     - Practical considerations. Assumption 4 can be satisfied for MFGs with transition dynamics sufficiently smooth in population measure (e.g., population independent transition kernel), a smooth reward function, and regularized policies (e.g., softmax policy operator with a large temperature).
> 5. Assumption 5, on the other hand, imposes no requirement on the policy operator or the transition kernel. However, it requires a small and smooth reward function and a small basis measure for function approximation.
>     - Practical considerations. By appropriately scaling the reward function and the basis measure, this assumption can always be satisfied in practice. See the response to Q1 for an example.

---

> ### Author Response · Authors · 2024-11-19
> **Response to Reviewer VqQ9 [Continued]**
>
> > **W2+Q2**: The baselines could be explained more clearly. Can you please explain what is FPI+MD in Section 7?
>
> In the revised manuscript, we have added a section "Reference methods" in Appendix C.1 to provide more detailed explanations of the baselines used in our numerical experiments, including their references and pseudocode.
>
> As cited in the paper, MD refers exactly to the online mirror descent algorithm developed by Perolat et al. (2021) and later considered in Laurière et al. (2022b).
> Since online mirror descent for learning MFGs also follows a forward-backward process (see Perolat et al. (2021)), we denote it as FPI+MD in our experiments.
> And demonstrated in our experiments, online mirror descent converges much faster than fictitious play-type iterations; see e.g., Figure 2(a).

---

> ### Author Response · Authors · 2024-11-19
> **Response to Reviewer VqQ9 [Continued]**
>
> > **Q1**: Please provide an example satisfying the assumptions of Theorem 2.
>
> We now show that the first example (speed control on a ring road) from our numerical experiments satisfies the assumptions of Theorem 2 under certain conditions.
> Recall that in Appendix C.2, we define the reward function as
> $$
> 	r(s,a,\mu) = - \frac{1}{2}\left(b(s) + \frac{1}{2}\left(1-\frac{\mu(s)}{\mu\_{\mathrm{jam}}}\right) - \frac{a}{a\_{\mathrm{max}}}\right)^2  \Delta s
> .$$
> First, this reward function is Lipschitz continuous in the population measure $\mu$, and since the transition dynamics of this game are independent of the population measure, Assumption 1 is satisfied.
>
> Assumption 2 can be satisfied by any Lipschitz continuous policy operator, such as a softmax policy.
>
> The transition dynamics are equivalent to the stochastic transition dynamics: the agent moves to the next state with probability $a$, and stays in the same state with probability $1-a$, where $a$ is the speed chosen by the agent.
> Since $a\in[0,1]$ and we consider soft policies where each action is chosen with a non-zero probability, the underlying Markov chain is uniformly ergodic, thus satisfying Assumption 3.
>
> Next, we examine Assumption 5. Recall that in Appendix I, $\bar{w}$ is determined by the smallest eigenvalues of Gram matrices of the feature maps, which are constant diagonal matrices independent of the reward function or the basis measure magnitude for a discretized state space (see Example 2).
> $\bar{L}$ in Assumption 5 is also determined by the underlying MDP but independent of the reward function or the population measure basis.
> Therefore, if the two inequalities in Assumption 5, namely $L\_{r}\le \bar{w} /2$ and $R+F \le \bar{w}^{2} /(4\bar{L})$, are not met, we can scale the reward function and the basis measure to satisfy these inequalities without affecting $\bar{w}$ and $\bar{L}$.
> Specifically, if we embed the ring road $\mathbb{S}^{1}$ into the real line as an interval $[0,s]$ and partition it into $|\mathcal{S}|$ states, the basis measure $\psi$ for this discretization satisfies
> $$
> F = \sup\_{s}\|\psi(s)\|_{1} = \frac{1}{\Delta s} = \frac{1}{s/|\mathcal{S}|} = \frac{|\mathcal{S}|}{s}
> .$$
> Thus, for a fixed number of states, we can choose $s > 8|\mathcal{S}|\bar{L}\bar{w}^{-2}$ so that $F \le \bar{w}^{2} /(8\bar{L})$.
> Similarly, suppose the original reward function has a Lipschitz constant $L\_{r}$ and supremum norm $R$. Then we can scale the reward function by $r \leftarrow cr$ so that the resultant Lipschitz constant $cL\_{r}\le \bar{w} /2$ and supremum norm $cR \le \bar{w}^{2} /(8\bar{L})$.
>
> In conclusion, MFGs with small reward functions and basis measures can satisfy Assumption 5. If the original MFG does not satisfy Assumption 5, we reformulate the game by scaling the reward function and the basis measure, resulting in a scaled MFG that satisfies Assumption 5.

---

> ### Author Response · Authors · 2024-11-22
> **Follow up**
>
> Thank you again for your valuable time in reviewing our work. We hope our responses have been helpful. If you have any further questions, we would be happy to provide more discussions!

---

> > ### Comment · Reviewer_VqQ9 · 2024-12-01
> > **Response**
> >
> > Thank you for the detailed answers; considering all the reviewers' questions and all the answers, I will keep my score as it is.

---

### Official Review · Reviewer_bQAJ · 2024-11-07

**Soundness:** 2
**Presentation:** 3
**Contribution:** 2
**Rating:** 5
**Confidence:** 4

**Summary:**

This paper primarily concentrates on the development of a stochastic gradient method for learning in mean-field games.
The proposed algorithm significantly reduces the computational cost associated with calculating the population distribution.
Moreover, the author demonstrates that under certain standard assumptions, the proposed algorithm converges to a stationary point.
Additionally, the algorithm's performance is validated experimentally in various mean-field games.

**Strengths:**

* The problem is well-motivated. Eliminating the forward-backward process of learning algorithms for mean-field games is significantly important.
* The proposed algorithm has a strong convergence guarantee with a rate.

**Weaknesses:**

My primary concern is that the definition of the mean-field equilibrium in this study differs from that in existing studies.
Specifically, this study considers the Bellman operator, as opposed to the commonly used Bellman optimality operator.
Hence, I believe that there is no guarantee for the agent to maximize the expected cumulative discounted reward in the mean field equilibria.
In the context of learning in multi-agent systems, it seems more natural to consider maximizing the expected cumulative discounted reward.
Consequently, I'm wondering why the current definition of equilibrium was considered in this study.
Is it possible to extend the provided theoretical results to the reward-maximizing setting?
If not, I don't think we can fairly compare this study with existing studies on mean-field games.

Additionally, I believe that a more detailed explanation of the numerical experiments is necessary.
For example, I'm curious why exploitability was reported for the flocking game, despite the fact that reward maximization is not the main focus of this study.

**Questions:**

My main concerns and questions are outlined in Weaknesses.
Additionally, I have the following question:
* I am not sure how Assumption 5 is relatively weaker than Assumption 4. Could you provide a more intuitive explanation of Assumption 5?

---

> ### Author Response · Authors · 2024-11-19
> **Response to Reviewer bQAJ**
>
> Dear Reviewer bQAJ,
>
> Thank you for your thoughtful review and for giving us the opportunity to address your concerns. We believe that most of your concerns stem from some confusion surrounding our notations, which aim to generalize existing definitions in the literature. We would like to offer the following clarifications.
>
> > **W1**: The definition of the mean-field equilibrium in this study differs from that in existing studies. Specifically, this study considers the Bellman operator, as opposed to the commonly used Bellman optimality operator. Hence, I believe that there is no guarantee for the agent to maximize the expected cumulative discounted reward in the mean field equilibria. ... I'm wondering why the current definition of equilibrium was considered in this study.
>
> Our definition of the mean-field equilibrium (MFE) encompasses the reward-maximizing setting by including the Bellman optimality operator through the use of an **argmax** policy operator.
> Specifically, with an argmax policy operator, the Bellman operator is defined as
> $$
> 	\mathcal{T}\_{(Q,M)} Q'(s,a) \coloneqq \mathbb{E}\_{(Q,M)}\left[r(s,a,M) + \gamma \max\_{a'} Q'(s',a')\right], \text{ with } a' = \operatorname{argmax}\_{a'}Q(s',a'), s' \sim P(\cdot\mid s,a,M).
> $$
> Then, the fixed point of this operator satisfies
> $$
>     Q^\*(s,a) = \mathbb{E}\_{(Q^\*,M)}\left[r(s,a,M) + \gamma \max\_{a'} Q^\*(s',a')\right],
> $$
> which is exactly the Bellman optimality equation. Therefore, when using an argmax policy operator, our definition is exactly the same as definitions in the literature, e.g., Laurière et al. (2022b), Zaman et al. (2023), and Angiuli et al. (2023), as noted in Section 2.1.
>
> Our definition is more general as it accommodates general policy operators, including regularized policies, thereby covering the **regularized MFE**.
> For example, using a softmax policy operator, our definition is exactly the same as Definition 4 (Boltzmann MFE) in Cui and Koeppl (2021), which is also adopted by Zaman et al. (2023, Definition 3).
> Regularized MFE is a widely used definition in the literature; see also, Mao et al. (2022) and Anahtarci et al. (2023).
> We have added a remark after Definition 1 and a dedicated section (Appendix B.1) elaborating on the above points.
> We hope this resolves the concern regarding the definition of the MFE.
>
> We now clarify the scope of MFE definitions considered in our theoretical results and numerical experiments. As stated in Remark 2, assumptions for Theorem 1 are typically satisfied by regularization. Hence, Theorem 1 provides convergence guarantees for regularized MFE.
> Note that although many works define MFE using the argmax policy operator, they often use regularized policies in algorithms; these works include Laurière et al. (2022b), Xie et al. (2021), Zaman et al. (2023), and all works listed in Table 2 with an (R) mark in the column "Dyna. assump." More detailed comparisons of theoretical properties are provided in Appendix A.
> We hope this clarifies the concern regarding our comparison with existing studies.
>
> Moreover, in Section 5.1, we explicitly remove Assumption 4, which is typically satisfied by regularization. As a result, Theorem 2 applies to the reward-maximizing setting and the standard definition of MFE.
> Regularization is not used in our numerical experiments, as stated in Remark 3 and throughout the paper. Thus, SemiSGD learns near-greedy policies that approximate the reward-maximizing equilibrium policy to arbitrary accuracy (see the near-greediness condition in Equation (7)).
>
> In summary, our definition of MFE, theoretical analysis, and numerical experiments all encompass the standard definition of MFE and the reward-maximizing setting.
> Our definition and Theorem 1 also cover regularized MFE, which is widely studied as regularization is an important stabilization mechanism in practice (see e.g., Laurière et al. (2022a); also discussed in the paper), and stability is one of the main themes of the paper.
> We aim to demonstrate that SemiSGD stabilizes without regularization, and effectively learn standard MFE and reward-maximizing equilibrium policies. Thus, removing regularization is another focus of the paper (Sections 5.1 and 7).

---

> ### Author Response · Authors · 2024-11-19
> **Response to Reviewer bQAJ [Continued]**
>
> > **W2**: Why exploitability was reported for the flocking game, despite the fact that reward maximization is not the main focus of this study. A more detailed explanation of the numerical experiments is necessary.
>
> We thank the reviewer for bringing up this point.
> To answer why exploitability was reported for the flocking game: as explained in our response to the previous point, learning reward-maximizing policies falls within the scope of our method, and the flocking game serves as a compelling example to demonstrate this capability. As highlighted throughout the paper (e.g., Remark 3), our numerical experiments involve learning near-greedy policies without regularization, which approximate reward-maximizing equilibrium policies to arbitrary accuracy. Consequently, we report exploitability across all experiments as it is an important measure for learned policies.
>
> Due to space constraints, the detailed explanation of the numerical experiments was moved to **Appendix C**, with clear pointers provided in the main text. Appendix C includes the experimental setups, analyses, implications of the results, and connections to the theoretical findings. In the revised manuscript, we have also added detailed descriptions of the baseline methods, including their references and pseudocode, in Appendix C.1.
> We now summarize the numerical experiments detailed in Appendix C and their takeaways:
>
> - Appendix C.2 considers a speed control game on a ring road, characterized by a continuous state-action space and a smooth kernel and reward function. SemiSGD performs the best in this game in terms of minimizing the mean squared error (MSE) of the population measure, showcasing its stability, sample efficiency, and accuracy.
> - Appendix C.3 evaluates the impact of the number of inner loop iterations for online FPI. Notably, when the number of inner loop iterations is set to 1, online FPI becomes equivalent to SemiSGD. The results indicate that reducing the number of inner loop iterations almost monotonically improves performance in both MSE and exploitability, strongly supporting the use of a fully asynchronous update scheme over the forward-backward process.
> - Appendix C.4 considers a one-dimensional flocking game. This game is special in that the reward function is small and the dynamics are highly sensitive to the policy, requiring algorithms capable of learning near-greedy policies and exploiting subtle reward signals. All methods except SemiSGD return near-constant policies that have high MSE and exploitability, as they either regularize the policy too much or cannot stably learn near-greedy policies. SemiSGD, however, achieves the best exploitability, demonstrating its ability to effectively detect and exploit small reward signals.
> - Appendix C.5 further explores the impact of regularization in the flocking game. The results verify the findings in Appendix C.4 that large regularization obscures reward signals in the flocking game and leads to poor performance, highlighting the importance of learning near-greedy policies in this game.
> - Appendix C.6 considers a routing game on a network, characterized by an discrete state-action space and a highly non-smooth sparse reward function. SemiSGD exhibits the most stable performance and achieves the lowest MSE, effectively learning reasonable policies under these challenging conditions.
> - Appendix C.7 compares linear function approximation and grid discretization in the speed control game which has a continuous state-action space. Results show that linear function approximation with a proper measure basis can significantly reduce the MSE compared to grid discretization, demonstrating the effectiveness of linear function approximation in handling continuous state-action spaces.

---

> ### Author Response · Authors · 2024-11-19
> **Response to Reviewer bQAJ [Continued]**
>
> > **Q1**: I am not sure how Assumption 5 is relatively weaker than Assumption 4. Could you provide a more intuitive explanation of Assumption 5?
>
> We thank the reviewer for giving us an opportunity to further explain this point.
> First, Assumption 5 is not claimed to be *weaker* than Assumption 4; as discussed in Section 5.1, we argue that Assumption 5 is more *practical*.
> In words, both assumptions require the reward function to be sufficiently smooth ($L\_{r} \lesssim 1$), but they impose conditions on different elements:
>
> - Assumption 4 requires the kernel to be sufficiently smooth ($L\_{P} \lesssim 1$), and the policy operator to be sufficiently smooth ($L\_{\pi}\lesssim 1$), which is typically satisfied by using regularized policies, such as softmax policies with a high temperature;
> - Assumption 5 requires the reward function to be sufficiently small ($R \lesssim 1$) and the basis measure to be sufficiently small ($F \lesssim 1$), which can be satisfied by scaling the reward function and the basis measure.
>
> Since these assumptions impose conditions on different components, neither is strictly weaker than the other. However, Assumption 5 is more practical in certain settings:
>
> 1. **Flexibility of policy operators.**
> Assumption 4, satisfied by regularization, restricts the types of policies that can be learned.
> Echoing your question in Weaknesses, if the reward-maximizing equilibrium policy is greedy (argmax policies), it may not be learnable under Assumption 4, as greedy or near-greedy policies typically do not satisfy the small $L_{\pi }$ condition.
> In contrast, Assumption 5 imposes no restriction on the policy operator, allowing for more flexible policy operators, including near-greedy policies that can approximate the reward-maximizing equilibrium policy with arbitrary accuracy.
> 2. **Practicality of scaling.**
> If the original MFG does not satisfy the small reward function or basis measure conditions in Assumption 5, the MFG can be reformulated through scaling of the reward function and the state space. Without loss of generality, we assume the original state space can be embedded in the interval $[0,|\mathcal{S}|]$.
> Then scaling by a factor $c$ gives
> $$
> \begin{cases}
> r \leftarrow  c r \implies R \leftarrow c R, L\_{r} \leftarrow c L\_{r},\\\\
> |\mathcal{S}| \leftarrow  c^{-1} |\mathcal{S}| \implies F \leftarrow c F.
> \end{cases}
> $$
> 3. **Invariance of dynamics.**
> The reformulation/scaling does not affect the game dynamics; specifically, it does not change the transition kernel and high-reward state-action pairs remain high-reward after scaling.

---

> ### Author Response · Authors · 2024-11-22
> **Follow up**
>
> Thank you again for your valuable time in reviewing our work. We hope our responses have been helpful and are happy to provide further clarifications if needed. If our discussion has addressed your concerns, we would greatly appreciate it if you could consider revising your evaluation of the manuscript. We look forward to hearing your feedback.

---

> > ### Comment · Reviewer_bQAJ · 2024-11-27
> >
> > Thank you for your detailed response and thorough explanations.
> >
> > I comprehend that the MFE under the Bellman optimality operator is also the MFE in this study.
> > However, I continue to believe that the inverse is not always true, i.e., the MFE in this study does not consistently satisfy the Bellman optimality condition.
> > Therefore, I suggest that the proposed algorithm is not necessarily guaranteed to converge to the MFE under the Bellman optimality operator.
> > At the very least, a rigorous proof is required to show that Theorem 1 is also applicable to MFE under the Bellman optimality operator.

---

> ### Author Response · Authors · 2024-11-27
>
> Thank you for your reply! We understand your concern and appreciate the opportunity to provide further clarification.
> Your understanding is correct that a limitation of Theorem 1 is that it does not generally apply to the MFE defined under the Bellman optimality operator. We would like to reiterate a portion of our initial response to address this point:
>
> > We now clarify the scope of MFE definitions considered in our theoretical results and numerical experiments. As stated in Remark 2, assumptions for Theorem 1 are typically satisfied by regularization. Hence, Theorem 1 provides convergence guarantees for regularized MFE. Note that although many works define MFE using the argmax policy operator, they often use regularized policies in algorithms; these works include Laurière et al. (2022b), Xie et al. (2021), Zaman et al. (2023), and all works listed in Table 2 with an (R) mark in the column "Dyna. assump." More detailed comparisons of theoretical properties are provided in Appendix A. We hope this clarifies the concern regarding our comparison with existing studies.
>
> We are fully transparent that Theorem 1 provides convergence guarantees for *regularized MFE*, with its assumptions ensuring the necessary contractivity.
> The assumptions in Theorem 1 ensure the required contractivity.
> While it would be desirable to extend these guarantees to the Bellman optimality based MFE, achieving this without imposing contractivity or monotonicity assumptions remains an open challenge.
> To the best of our knowledge, no prior work has established theoretical convergence guarantees for MFE in this setting without such assumptions. We would appreciate any references or approaches that have addressed this challenge to further inform our work.
>
> Additionally, Section 5.1 and Theorem 2 represent a step toward addressing the Bellman optimality based MFE. In this section, we remove the regularization and contractivity condition. A trade-off is that we only show convergence to a neighborhood of the MFE.
>
> We hope this clarifies the scope of our theoretical results and the challenges in extending them to the Bellman optimality based MFE. We appreciate your feedback and are open to further discussions on this topic.

---

### Author Response · Authors · 2024-11-19
**General Response**

We would like to thank the reviewers for their valuable and insightful comments on our work. We are happy with the overall positive feedback regarding
clarity of presentation (Reviewers `VqQ9` and `b6VH`),
significance of the problem considered in eliminating the forward-backward process (Reviewers `bQAJ` and `b6VH`),
novelty of the method in unifying the policy and population parameters and applying population-aware linear function approximation (Reviewers `VqQ9`, `b6VH`, and `B4eq`),
extensive analysis and strong finite-time convergence guarantees (Reviewers `bQAJ`, `VqQ9`, and `b6VH`),
and effectiveness of the proposed algorithm in practice (Reviewers `VqQ9`, `b6VH`, and `B4eq`).

We are also grateful for the suggestions for improvement. In what follows, we summarize our responses to each of the main comments. We also point out the revisions we have made to the paper in response to these comments.
The modifications are highlighted in **orange** in the revised manuscript.

- Reviewer `bQAJ`: We clarified that our definition of mean-field equilibrium (MFE) does indeed encompass the standard definition of MFE, and our theoretical results and numerical experiments do extend to the reward-maximizing setting. We have added a dedicated section, Appendix B.1, to elaborate on the definition of MFE and its relationship to existing works. We have also added a section in Appendix C.1 to provide detailed explanations of the baseline methods used in our numerical experiments.
- Reviewer `VqQ9`: We provided a detailed explanation of how to verify the assumptions in practice and offered an example satisfying the assumptions of Theorem 2. We have added a section in Appendix C.1 to provide detailed explanations of the baseline methods used in our numerical experiments.
- Reviewer `b6VH`: We clarified that our method, analysis, and experiments go beyond linear models and regularized solutions. We also discussed how our method can be generalized to non-stationary MFGs and how we might directly optimize exploitability. In the revised manuscript, we have added the analysis of MFE uniqueness implied by Assumption 4 or 5.
- Reviewer `B4eq`: We provided detailed discussions about Assumption 2, Theorem 2, and comparisons with assumptions in previous work. We have added a dedicated section (Appendix B.1) that justifies our definitions of MFE and policy operators, as well as their relationship to existing works.

We hope our responses and revisions address the comments raised. We would be happy to engage in further discussions if any queries remain.

---

### Meta-Review · Area_Chair_rbPD · 2024-12-20

**Metareview:**

This paper treats the problem of learning in mean-field games (MFGs) and, in particular, analyzes a stochastic approximation method dubbed SemiSGD (stochastic semi-gradient descent) in which agents simultaneously update their policy and population estimates. The authors provide a finite-time analysis for SemiSGD, and they establish its convergence to regularized mean-field equilibria (or neighobrhoods of mean-field equilibria).

The reviewers were mostly positive regarding the paper, and found it relatively well-written and topical. Two main concerns that were raised concern the following points:
- The authors' convergence analysis in Theorem 1 concerns the game's regularized MFE, with the degree of regularization being sufficient to ensure contraction (and, a fortiori, a unique regularized MFE). This reduces the relevance of this result (the authors stated in their rebuttal that they were transparent about this, but the writing is not sufficiently clear in this regard).
- It was also unclear how the authors' assumptions can be verified in practice - especially the delicate conditions involving $w$ and the eigenvalues of the various operators in Appendix H (which, incidentally, make the paper very hard to read at that point).

The authors provided very extensive replies to the reviewers' other questions, but the above concerns remained. While I remain concerned that these points impose considerable limitations on the authors' results (which are not adequately discussed in the paper), I believe the paper's merits outweigh its flaws, so I recommend acceptance.

**Additional Comments On Reviewer Discussion:**

Except for the concerns raised above, the authors provided very extensive replies to the reviewers' questions. Reviewer bQAJ (and myself) remained skeptical concerning the meaningfulness of high amounts of regularizations but, in the end, it was decided that this does not outweigh the more positive aspects of the paper.

---

### Decision · Program_Chairs · 2025-01-22

Accept (Poster)